# Equal Improvability: A New Fairness Notion Considering the Long-term Impact

**Ozgur Guldogan**[*‡] **Yuchen Zeng**[*§] **Jy-yong Sohn**[†¶] **Ramtin Pedarsani**[‡] **Kangwook Lee**[§]
[‡] University of California, Santa Barbara  [¶] Yonsei University  [§] University of Wisconsin-Madison

## Abstract

Devising a fair classifier that does not discriminate against different groups is an important problem in machine learning. Recently, effort-based fairness notions are getting attention, which considers the scenarios of each individual making effort to improve its feature over time. Such scenarios happen in the real world, e.g., college admission and credit loaning, where each rejected sample makes effort to change its features to get accepted afterward. In this paper, we propose a new effort-based fairness notion called Equal Improvability (EI), which equalizes the potential acceptance rate of the rejected samples across different groups assuming a bounded level of effort will be spent by each rejected sample. We also propose and study three different approaches for finding a classifier that satisfies the EI requirement. Through experiments on both synthetic and real datasets, we demonstrate that the proposed EI-regularized algorithms encourage us to find a fair classifier in terms of EI. Additionally, we ran experiments on dynamic scenarios which highlight the advantages of our EI metric in equalizing the distribution of features across different groups, after the rejected samples make some effort to improve. Finally, we provide mathematical analyses of several aspects of EI: the relationship between EI and existing fairness notions, and the effect of EI in dynamic scenarios. Codes are available in a GitHub repository [1].

## 1 Introduction

Over the past decade, machine learning has been used in a wide variety of applications. However, these machine learning approaches are observed to be unfair to individuals having different ethnicity, race, and gender. As the implicit bias in artificial intelligence tools raised concerns over potential discrimination and equity issues, various researchers suggested defining fairness notions and developing classifiers that achieve fairness. One popular fairness notion is *demographic parity* (DP), which requires the decision-making system to provide output such that the groups are equally likely to be assigned to the desired prediction classes, *e.g.*, acceptance in the admission procedure. DP and related fairness notions are largely employed to mitigate the bias in many realistic problems such as recruitment, credit lending, and university admissions (Zafar et al., 2017b; Hardt et al., 2016; Dwork et al., 2012; Zafar et al., 2017a).

However, most of the existing fairness notions only focus on immediate fairness, without taking potential follow-up inequity risk into consideration. In Fig. 1, we provide an example scenario when using DP fairness has a long-term fairness issue, in a simple loan approval problem setting. Consider two groups (group 0 and group 1) with different distributions, where each individual has one label (approve the loan or not) and two features (credit score, income) that can be improved over time. Suppose each group consists of two clusters (with three samples each), and the distance between the clusters is different for two groups. Fig. 1 visualizes the distributions of two groups and the decision boundary of a classifier $f$ which achieves DP among the groups. We observe that the rejected samples (left-hand-side of the decision boundary) in group 1 are located further away from the decision boundary than the rejected samples in group 0. As a result, the rejected applicants in group 1 need to make more effort to cross the decision boundary and get approval. This improvability gap between the two groups can make the rejected applicants in group 1 less motivated to improve their features, which may increase the gap between different groups in the future.

---

[*]Equal Contribution. Emails: `ozgurguldogan@ucsb.edu`, `yzeng58@wisc.edu`.
[†]Work done at the University of Wisconsin-Madison.

[1]https://github.com/guldoganozgur/ei_fairness

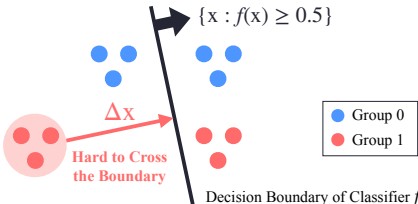

This motivated the advent of fairness notions that consider dynamic scenarios when each rejected sample makes effort to improve its feature, and measure the group fairness after such effort is made Gupta et al. (2019); Heidari et al. (2019); Von Kügelgen et al. (2022). However, as shown in Table 1, they have various limitations *e.g.*, vulnerable to imbalanced group negative rates or outliers.

In this paper, we introduce another fairness notion designed for dynamic scenarios, dubbed as *Equal Improvability* (EI), which does not suffer from these limitations. Let $\mathbf{x}$ be the feature of a sample and $f$ be a score-based classifier, *e.g.*, predicting a sample as accepted if $f(\mathbf{x}) \geq 0.5$ holds and as rejected otherwise. We assume each rejected individual wants to get accepted in the future, thus

Figure 1: **Toy example showing the insufficiency of fairness notion that does not consider improvability.** We consider the binary classification (accept/reject) on 12 samples (dots), where $\mathbf{x}$ is the feature of the sample and the color of the dot represents the group. The given classifier $f$ is fair in terms of a popular notion called demographic parity (DP), but does not have equal improvability of rejected samples ($f(\mathbf{x}) < 0.5$) in two groups; the rejected samples in group 1 needs more effort $\Delta \mathbf{x}$ to be accepted, *i.e.*, $f(\mathbf{x} + \Delta \mathbf{x}) \geq 0.5$, compared with the rejected samples in group 0.

improving its feature within a certain effort budget towards the direction that maximizes its score $f(\mathbf{x})$. Under this setting, we define EI fairness as the equity of the potential acceptance rate of the different rejected groups, once each individual makes the best effort within the predefined budget. This prevents the risk of exacerbating the gap between different groups in the long run.

Our key contributions are as follows:

- We propose a new group fairness notion called *Equal Improvability* (EI), which aims to equalize the probability of rejected samples being qualified after a certain amount of feature improvement, for different groups. EI encourages rejected individuals in different groups to have an equal amount of motivation to improve their feature to get accepted in the future. We analyze the properties of EI and the connections of EI with other existing fairness notions.

- We provide three methods to find a classifier that is fair in terms of EI, each of which uses a unique way of measuring the inequity in the improvability. Each method is solving a min-max problem where the inner maximization problem is finding the best effort to measure the EI unfairness, and the outer minimization problem is finding the classifier that has the smallest fairness-regularized loss. Experiments on synthetic/real datasets demonstrate that our algorithms find classifiers having low EI unfairness.

- We run experiments on dynamic scenarios where the data and the classifier evolve over multiple rounds, and show that training a classifier with EI constraints is beneficial for making the feature distributions of different groups identical in the long run.

## 2 EQUAL IMPROVABILITY

Before defining our new fairness notion called *Equal Improvability* (EI), we first introduce necessary notations. For an integer $n$, let $[n] = \{0, \ldots, n-1\}$. We consider a binary classification setting where each data sample has an input feature vector $\mathbf{x} \in \mathcal{X} \subseteq \mathbb{R}^d$ and a label $\mathbf{y} \in \mathcal{Y} = \{0, 1\}$. In particular, we have a sensitive attribute $\mathbf{z} \in \mathcal{Z} = [Z]$, where $Z$ is the number of sensitive groups. As suggested by Chen et al. (2021), we sort $d$ features $\mathbf{x} \in \mathbb{R}^d$ into three categories: improvable features $\mathbf{x}_{\mathrm{I}} \in \mathbb{R}^{d_{\mathrm{I}}}$, manipulable features $\mathbf{x}_{\mathrm{M}} \in \mathbb{R}^{d_{\mathrm{M}}}$, and immutable features $\mathbf{x}_{\mathrm{IM}} \in \mathbb{R}^{d_{\mathrm{IM}}}$, where $d_{\mathrm{I}} + d_{\mathrm{M}} + d_{\mathrm{IM}} = d$ holds. Here, improvable features $\mathbf{x}_{\mathrm{I}}$ refer to the features that can be improved and can directly affect the outcome, *e.g.*, salary in the credit lending problem, and GPA in the school's admission problem. In contrast, manipulable features $\mathbf{x}_{\mathrm{M}}$ can be altered, but are not directly related to the outcome, *e.g.*, marital status in the admission problem, and communication type in the credit lending problem. Although individuals may manipulate these manipulable features to get the desired outcome, we do not consider it as a way to make efforts as it does not affect the individual's true qualification status. Immutable features $\mathbf{x}_{\mathrm{IM}}$ are features that cannot be altered, such as race, age, or date of birth. Note that if sensitive attribute $\mathbf{z}$ is included in the feature vector, then it belongs to immutable features. For ease of notation, we write $\mathbf{x} = (\mathbf{x}_{\mathrm{I}}, \mathbf{x}_{\mathrm{M}}, \mathbf{x}_{\mathrm{IM}})$. Let $\mathcal{F} = \{f : \mathcal{X} \to [0, 1]\}$ be the set of classifiers, where each classifier is parameterized by $\boldsymbol{w}$, *i.e.*, $f = f_{\boldsymbol{w}}$. Given $f \in \mathcal{F}$, we consider the following deterministic prediction: $\hat{\mathbf{y}} \mid \mathbf{x} = \mathbf{1}\{f(\mathbf{x}) \geq 0.5\}$ where $\mathbf{1}\{A\} = 1$ if condition A holds, and $\mathbf{1}\{A\} = 0$ otherwise. We now introduce our new fairness notion.

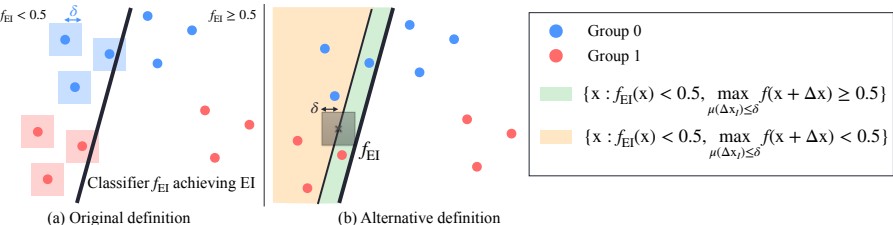

Figure 2: **Visualization of EI fairness.** For binary classification on 12 samples (dots) in two groups (red/blue), we visualize the fairness notion defined in this paper: (a) shows the original definition in Def. 2.1, and (b) shows an equivalent definition in Prop. 2.2. Here we assume two-dimensional features (both are improvable) and $L_\infty$ norm for $\mu(\mathbf{x}) = \|\mathbf{x}\|_\infty$. The classifier $f_{\mathrm{EI}}$ achieves equal improvability (EI) since the same portion (1 out of 3) of unqualified samples in each group can be improved to qualified samples.

**Definition 2.1** (Equal Improvability). Define a norm $\mu : \mathbb{R}^{d_\mathrm{I}} \to [0, \infty)$. For a given constant $\delta > 0$, a classifier $f$ is said to achieve *equal improvability with $\delta$-effort* if

$$\mathbb{P}\left(\max_{\mu(\Delta\mathbf{x}_\mathrm{I})\leq\delta} f(\mathbf{x} + \Delta\mathbf{x}) \geq 0.5 \mid f(\mathbf{x}) < 0.5, \mathbf{z} = z\right) = \mathbb{P}\left(\max_{\mu(\Delta\mathbf{x}_\mathrm{I})\leq\delta} f(\mathbf{x} + \Delta\mathbf{x}) \geq 0.5 \mid f(\mathbf{x}) < 0.5\right)$$

holds for all $z \in \mathcal{Z}$, where $\Delta\mathbf{x}_\mathrm{I}$ is the effort for improvable features and $\Delta\mathbf{x} = (\Delta\mathbf{x}_\mathrm{I}, \mathbf{0}, \mathbf{0})$.

Note that the condition $f(\mathbf{x}) < 0.5$ represents that an individual is unqualified, and $f(\mathbf{x} + \Delta\mathbf{x}) \geq 0.5$ implies that the effort $\Delta\mathbf{x}$ allows the individual to become qualified. The above definition of fairness in *equal improvability* requires that unqualified individuals from different groups $z \in [Z]$ are equally likely to become qualified if appropriate effort is made. Note that $\mu$ can be defined on a case-by-case basis. For example, we can use $\|\mathbf{x}_\mathrm{I}\| = \sqrt{\mathbf{x}_\mathrm{I}^\top C \mathbf{x}_\mathrm{I}}$, where $C \in \mathbb{R}^{d_\mathrm{I} \times d_\mathrm{I}}$ is a cost matrix that is diagonal and positive definite. Here, the diagonal terms of $C$ characterize how difficult to improve each feature. For instance, consider the graduate school admission problem where $\mathbf{x}_\mathrm{I}$ contains features "number of publications" and "GPA". Since publishing more papers is harder than raising the GPA, the corresponding diagonal term for the number of publications feature in $C$ should be greater than that for the GPA feature. The constant $\delta$ in Definition 2.1 can be selected depending on the classification task and the features. Appendix B.1 contains the interpretation of each term in Def. 2.1. We introduce an equivalent definition of EI fairness below.

**Proposition 2.2.** *The EI fairness notion defined in Def. 2.1 has an equivalent format: a classifier $f$ achieves equal improvability with $\delta$-effort if and only if*

$$\mathbb{P}(\mathbf{x} \in \mathcal{X}_-^{\mathrm{imp}} \mid \mathbf{x} \in \mathcal{X}_-, \mathbf{z} = z) = \mathbb{P}(\mathbf{x} \in \mathcal{X}_-^{\mathrm{imp}} \mid \mathbf{x} \in \mathcal{X}_-)$$

*holds for all $z \in \mathcal{Z}$ where $\mathcal{X}_- = \{\mathbf{x} : f(\mathbf{x}) < 0.5\}$ is the set of features $\mathbf{x}$ for unqualified samples, and $\mathcal{X}_-^{\mathrm{imp}} = \{\mathbf{x} : f(\mathbf{x}) < 0.5, \max_{\mu(\Delta\mathbf{x}_I)\leq\delta} f(\mathbf{x} + \Delta\mathbf{x}) \geq 0.5\}$ is the set of features $\mathbf{x}$ for unqualified samples that can be improved to qualified samples by adding $\Delta\mathbf{x}$ satisfying $\mu(\Delta\mathbf{x}_I) \leq \delta$.*

The proof of this proposition is trivial from the definition of $\mathcal{X}_-$ and $\mathcal{X}_-^{\mathrm{imp}}$. Note that $\mathbb{P}(\mathbf{x} \in \mathcal{X}_-^{\mathrm{imp}} \mid \mathbf{x} \in \mathcal{X}_-)$ in the above equation indicates the probability that unqualified samples can be improved to qualified samples by changing the features within budget $\delta$. This is how we define the "improvability" of unqualified samples, and the EI fairness notion is equalizing this improvability for all groups.

**Visualization of EI.** Fig. 2 shows the geometric interpretation of EI fairness notion in Def. 2.1 and Prop. 2.2, for a simple two-dimensional dataset having 12 samples in two groups $z \in \{\mathrm{red}, \mathrm{blue}\}$. Consider a linear classifier $f_{\mathrm{EI}}$ shown in the figure, where the samples at the right-hand-side of the decision boundary is classified as qualified samples ($f_{\mathrm{EI}}(\mathbf{x}) \geq 0.5$). In Fig. 2a, we have $L_\infty$ ball at each unqualified sample, representing that these samples have a chance to improve their feature in a way that the improved feature $\mathbf{x} + \Delta\mathbf{x}$ allows the sample to be classified as qualified, *i.e.*, $f_{\mathrm{EI}}(\mathbf{x} + \Delta\mathbf{x}) \geq 0.5$. One can confirm that $\mathbb{P}\left(\max_{\mu(\Delta\mathbf{x}_\mathrm{I})\leq\delta} f(\mathbf{x} + \Delta\mathbf{x}) \geq 0.5 \mid f(\mathbf{x}) < 0.5, \mathbf{z} = z\right) = \frac{1}{3}$ holds for each group $z \in \{\mathrm{red}, \mathrm{blue}\}$, thus satisfying equal improvability according to Def. 2.1. In Fig. 2b, we check this in an alternative way by using the EI fairness definition in Prop. 2.2. Here, instead of making a set of improved features at each sample, we partition the feature domain $\mathcal{X}$ into three parts: (i) the features for qualified samples $\mathcal{X}_+ = \{\mathbf{x} : f_{\mathrm{EI}}(\mathbf{x}) \geq 0.5\}$, (ii) the features for unqualified samples that can be improved $\mathcal{X}_-^{\mathrm{imp}} = \{\mathbf{x} : f_{\mathrm{EI}}(\mathbf{x}) < 0.5, \max_{\mu(\Delta\mathbf{x}_\mathrm{I})\leq\delta} f(\mathbf{x} + \Delta\mathbf{x}) \geq 0.5\}$, and (iii) the features for unqualified samples that cannot be improved $\{\mathbf{x} : f_{\mathrm{EI}}(\mathbf{x}) < 0.5, \max_{\mu(\Delta\mathbf{x}_\mathrm{I})\leq\delta} f(\mathbf{x} + \Delta\mathbf{x}) < 0.5\}$. In the figure, (ii) is represented as the green region and (iii) is shown as the yellow region. From Prop. 2.2, EI fairness means that $\frac{\text{\# samples in (ii)}}{\text{\# samples in (ii) + \# samples in (iii)}}$ is identical at each group $z \in \{\mathrm{red}, \mathrm{blue}\}$, which is true for the example in Fig. 2b.

Table 1: Comparison of our EI fairness with existing fairness notions.

| Name of fairness | Definition | Consider efforts? | Limitations |
|---|---|---|---|
| **Equal Improvability (Ours)** | $\mathbb{P}\left(\max_{\mu(\Delta\mathbf{x}) \leq \delta} f(\mathbf{x} + \Delta\mathbf{x}) \geq 0.5 \mid f(\mathbf{x}) < 0.5, \mathbf{z} = z\right) = \mathbb{P}\left(\max_{\mu(\Delta\mathbf{x}) \leq \delta} f(\mathbf{x} + \Delta\mathbf{x}) \geq 0.5 \mid f(\mathbf{x}) < 0.5\right)$ | Yes | - |
| Demographic Parity | $\mathbb{P}\left(f(\mathbf{x}) \geq 0.5 \mid \mathbf{z} = z\right) = \mathbb{P}\left(f(\mathbf{x}) \geq 0.5\right)$ | No | - |
| Equal Opportunity (Hardt et al., 2016) | $\mathbb{P}\left(f(\mathbf{x}) \geq 0.5 \mid \mathbf{y} = 1, \mathbf{z} = z\right) = \mathbb{P}\left(f(\mathbf{x}) \geq 0.5 \mid \mathbf{y} = 1\right)$ | No | - |
| Equalized Odds (Hardt et al., 2016) | $\mathbb{P}\left(f(\mathbf{x}) \geq 0.5 \mid \mathbf{y} = y, \mathbf{z} = z\right) = \mathbb{P}\left(f(\mathbf{x}) \geq 0.5 \mid \mathbf{y} = y\right)$ | No | - |
| Bounded Effort (Heidari et al., 2019) | $\mathbb{P}\left(\max_{\mu(\Delta\mathbf{x}) \leq \delta} f(\mathbf{x} + \Delta\mathbf{x}) \geq 0.5, f(\mathbf{x}) < 0.5 \mid \mathbf{z} = z\right) = \mathbb{P}\left(\max_{\mu(\Delta\mathbf{x}) \leq \delta} f(\mathbf{x} + \Delta\mathbf{x}) \geq 0.5, f(\mathbf{x}) < 0.5\right)$ | Yes | Cannot handle imbalanced group negative rates (Appendix C.4.2) |
| Equal Recourse (Gupta et al., 2019) | $\mathbb{E}\left[\min_{f(\mathbf{x}+\Delta\mathbf{x}) \geq 0.5} \mu(\Delta\mathbf{x}) \mid f(\mathbf{x}) < 0.5, \mathbf{z} = z\right] = \mathbb{E}\left[\min_{f(\mathbf{x}+\Delta\mathbf{x}) \geq 0.5} \mu(\Delta\mathbf{x}) \mid f(\mathbf{x}) < 0.5\right]$ | Yes | Vulnerable to outliers (Appendix B.3 and C.4.1) |
| Individual-Level Fair Causal Recourse (Von Kügelgen et al., 2022) | $\min_{f(\mathbf{x}') \geq 0.5} \mu(\mathbf{x}' - \mathbf{x}(z, \mathbf{u})) = \min_{f(\mathbf{x}') \geq 0.5} \mu(\mathbf{x}' - \mathbf{x}(z', \mathbf{u}))$ for all latent variable $\mathbf{u}$, all groups $z, z'$ | Yes | Limitations of counterfactual fairness |

**Comparison of EI with other fairness notions.** The suggested fairness notion *equal improvability* (EI) is in stark difference with existing popular fairness notions that do not consider dynamics, *i.e.*, demographic parity (DP), equal opportunity (EO) (Hardt et al., 2016), and equalized odds (EOD) (Hardt et al., 2016), which can be "myopic" and focuses only on achieving classification fairness in the current status. Our notion instead, aims at achieving classification fairness in the long run when each sample improves its feature over time.

On the other hand, EI also has differences with existing fairness notions that capture the dynamics of samples (Heidari et al., 2019; Huang et al., 2019; Gupta et al., 2019; Von Kügelgen et al., 2022). Table 1 compares our fairness notion with the related existing notions. In particular, Bounded Effort (BE) fairness proposed by Heidari et al. (2019) equalizes "the available reward after each individual making a bounded effort" for different groups, which is very similar to EI when we set a proper reward function (see Appendix B.2). To be more specific, the BE fairness can be represented as in Table 1. Comparing this BE expression with EI in Definition 2.1, one can confirm the difference: the inequality $f(\mathbf{x}) < 0.5$ is located at the conditional part for EI, which is not true for BE. EI and BE are identical if the negative prediction rates are equal across the groups, but in general, they are different. The condition $f(\mathbf{x}) < 0.5$ here is very important since only looking into the unqualified members makes more sense when we consider *improvability*. More importantly, the BE definition is based on reward functions and we are presenting BE in a form that is closest to our EI fairness expression. Besides, Equal Recourse (ER) fairness proposed by Gupta et al. (2019) suggests equalizing the average effort of different groups without limiting the amount of effort that each sample can make. Note that ER is vulnerable to outliers. For example, when we have an unqualified outlier sample that is located far way from the decision boundary, ER disparity will be dominated by this outlier and fail to reflect the overall unfairness. Von Kügelgen et al. (2022) suggested Individual-Level Fair Causal Recourse (ILFCR), a fairness notion that considers a more general setting that allows causal influence between the features. This notion aims to equalize the cost of recourse required for a rejected individual to obtain an improved outcome if the individual is from a different group. Individual-level equal recourse shares a similar spirit with EI since both of them are taking care of equalizing the potential to improve the decision outcome for the rejected samples. However, introducing individual-level fairness with respect to different groups inherently requires counterfactual fairness, which has its own limitation, as described in Wu et al. (2019), and it is also vulnerable to outliers. Huang et al. (2019) proposed a causal-based fairness notion to equalize the minimum level of effort such that the expected prediction score of the groups is equal to each other. Note that, their definition is specific to causal settings and it considers the whole sensitive groups instead of the rejected samples of the sensitive groups. In addition to fairness notions, we also discuss other related works such as fairness-aware algorithms in Sec. 5.

**Compatibility of EI with other fairness notions.** Here we prove the compatibility of three fairness notions (EI, DP, and BE), under two mild assumptions. Assumption 2.3 ensures that EI is well-defined, while Assumption 2.4 implies that the norm $\mu$ and the effort budget $\delta$ are chosen such that we have nonzero probability that unqualified individuals can become qualified after making efforts.

**Assumption 2.3.** For any classifier $f$, the probability of unqualified samples for each demographic group is not equal to 0, *i.e.*, $\mathbb{P}\left(f(\mathbf{x}) < 0.5, \mathbf{z} = z\right) \neq 0$ for all $z \in \mathcal{Z}$.

**Assumption 2.4.** For any classifier $f$, the probability of being qualified after the effort for unqualified samples is not equal to 0, *i.e.*, $\mathbb{P}\left(\max_{\mu(\Delta\mathbf{x}_1) \leq \delta} f(\mathbf{x} + \Delta\mathbf{x}) \geq 0.5, f(\mathbf{x}) < 0.5\right) \neq 0$.

Under these assumptions, the following theorem reveals the relationship between DP, EI and BE.

**Theorem 2.5.** *If a classifier $f$ achieves two of the following three fairness notions, DP, EI, and BE; then it has to achieve the remaining fairness notion as well.*

The proof of the Theorem 2.5 is provided in Appendix A.1. This theorem immediately implies the following corollary, which provides a condition such that EI and BE conflict with each other.

**Corollary 2.6.** *The above theorem says that if a classifier $f$ achieves EI and BE, it has to achieve DP. Thus, by contraposition, if $f$ does not achieve DP, then it cannot achieve EI and BE simultaneously.*

Besides, we also investigate the connections between EI and ER in Appendix A.2.

## 3 ACHIEVING EQUAL IMPROVABILITY

In this section, we discuss methods for finding a classifier that achieves EI fairness. Following existing in-processing techniques (Zafar et al., 2017c; Donini et al., 2018; Zafar et al., 2017a; Cho et al., 2020), we focus on finding a fair classifier by solving a fairness-regularized optimization problem. To be specific, we first derive a differentiable penalty term $U_\delta$ that approximates the unfairness with respect to EI, and then solve a regularized empirical minimization problem having the unfairness as the regularization term. This optimization problem can be represented as

$$\min_{f \in \mathcal{F}} \left\{ \frac{(1-\lambda)}{N} \sum_{i=1}^{N} \ell(\mathrm{y}_i, f(\mathbf{x}_i)) + \lambda U_\delta \right\}, \tag{1}$$

where $\{(\mathbf{x}_i, \mathrm{y}_i)\}_{i=1}^{N}$ is the given dataset, $\ell : \{0,1\} \times [0,1] \to \mathbb{R}$ is the loss function, $\mathcal{F}$ is the set of classifiers we are searching over, and $\lambda \in [0,1)$ is a hyperparameter that balances fairness and prediction loss. Here we consider three different ways of defining the penalty term $U_\delta$, which are (a) covariance-based, (b) kernel density estimator (KDE)-based, and (c) loss-based methods. We first introduce how we define $U_\delta$ in each method, and then discuss how we solve (1).

**Covariance-based EI penalty.** Our first method is inspired by Zafar et al. (2017c), which measures the unfairness of a score-based classifier $f$ by the covariance of the sensitive attribute z and the score $f(\mathbf{x})$, when the demographic parity (DP) fairness condition ($\mathbb{P}(f(\mathbf{x}) > 0.5 | \mathrm{z} = z) = \mathbb{P}(f(\mathbf{x}) > 0.5)$ holds for all $z$) is considered. The intuition behind this idea of measuring the covariance is that a perfect fair DP classifier should have zero correlation between z and $f(\mathbf{x})$. By applying similar approach to our fairness notion in Def. 2.1, the EI unfairness is measured by the covariance between the sensitive attribute z and the maximally improved score of rejected samples within the effort budget. In other words, $(\mathrm{Cov}(\mathrm{z}, \max_{\|\Delta\mathbf{x}_i\| \le \delta} f(\mathbf{x} + \Delta\mathbf{x}) \mid f(\mathbf{x}) < 0.5))^2$ represents the EI unfairness of a classifier $f$ where we took the square to penalize negative correlation case as well. Let $I_- = \{i : f(\mathbf{x}_i) < 0.5\}$ be the set of indices of unqualified samples, and $\bar{\mathrm{z}} = \sum_{i \in I_-} \mathrm{z}_i / |I_-|$. Then, EI unfairness can be approximated by the square of the empirical covariance, *i.e.*,

$$U_\delta \triangleq \left( \frac{1}{|I_-|} \sum_{i \in I_-} (\mathrm{z}_i - \bar{\mathrm{z}}) \left( \max_{\|\Delta\mathbf{x}_{I_i}\| \le \delta} f(\mathbf{x}_i + \Delta\mathbf{x}_i) - \sum_{j \in I_-} \max_{\|\Delta\mathbf{x}_{I_j}\| \le \delta} f(\mathbf{x}_j + \Delta\mathbf{x}_j)/|I_-| \right) \right)^2.$$

Since $\sum_{i \in I_-} (\mathrm{z}_i - \bar{\mathrm{z}}) \left( \sum_{j \in I_-} \max_{\|\Delta\mathbf{x}_{I_j}\| \le \delta} f(\mathbf{x}_j + \Delta\mathbf{x}_j)/|I_-| \right) = 0$ from $\sum_{i \in I_-} (\mathrm{z}_i - \bar{\mathrm{z}}) = 0$, we have $U_\delta = \left( \frac{1}{|I_-|} \sum_{i \in I_-} (\mathrm{z}_i - \bar{\mathrm{z}}) \max_{\|\Delta\mathbf{x}_{I_i}\| \le \delta} f(\mathbf{x}_i + \Delta\mathbf{x}_i) \right)^2$.

**KDE-based EI penalty.** The second method is inspired by Cho et al. (2020), which suggests to first approximate the probability density function of the score $f(\boldsymbol{x})$ via kernel density estimator (KDE) and then put the estimated density formula into the probability term in the unfairness penalty. Recall that given $m$ samples $\mathrm{y}_1, \dots, \mathrm{y}_m$, the true density $g_{\mathrm{y}}$ on y is estimated by KDE as $\hat{g}_{\mathrm{y}}(\hat{\mathrm{y}}) \triangleq \frac{1}{mh} \sum_{i=1}^{m} g_k \left( \frac{\hat{\mathrm{y}} - \mathrm{y}_i}{h} \right)$, where $g_k$ is a kernel function and $h$ is a smoothing parameter.

Here we apply this KDE-based method for estimating the EI penalty term in Def. 2.1. Let $\mathrm{y}_i^{\max} = \max_{\|\Delta\mathbf{x}_{I_i}\| \le \delta} f(\mathbf{x}_i + \Delta\mathbf{x}_i)$ be the maximum score achievable by improving feature $\mathbf{x}_i$ within budget $\delta$, and $I_{-,z} = \{i : f(\mathbf{x}_i) < 0.5, \mathrm{z}_i = z\}$ be the set of indices of unqualified samples of group $z$. Then, the density of $\mathrm{y}_i^{\max}$ for the unqualified samples in group $z$ can be approximated as[2]

$$\hat{g}_{\mathrm{y}^{\max}|f(\mathbf{x})<0.5,z}(\hat{\mathrm{y}}^{\max}) \triangleq \frac{1}{|I_{-,z}|h} \sum_{i \in I_{-,z}} g_k \left( \frac{\hat{\mathrm{y}}^{\max} - \mathrm{y}_i^{\max}}{h} \right).$$

Then, the estimate on the left-hand-side (LHS) probability term in Def. 2.1 is represented as $\hat{\mathbb{P}} \left( \max_{\mu(\Delta\mathbf{x}_I) \le \delta} f(\mathbf{x} + \Delta\mathbf{x}) \ge 0.5 \mid f(\mathbf{x}) < 0.5, \mathrm{z} = z \right) = \int_{0.5}^{\infty} \hat{g}_{\mathrm{y}^{\max}|f(\mathbf{x})<0.5,z}(\hat{\mathrm{y}}^{\max}) \mathrm{d}\hat{\mathrm{y}}^{\max} =$

---

[2]This term is differentiable with respect to model parameters, since $g_k$ is differentiable w.r.t $\mathrm{y}_i^{\max}$, and $\mathrm{y}_i^{\max} = \max_{\|\Delta\mathbf{x}_{I_i}\| \le \delta} f(\mathbf{x}_i + \Delta\mathbf{x}_i)$ is differentiable w.r.t. model parameters from (Danskin, 1967).

$\frac{1}{|I_{-,z}|h} \sum_{i \in I_{-,z}} G_k \left( \frac{0.5 - \mathrm{y}_i^{\max}}{h} \right)$ where $G_k(\tau) \triangleq \int_\tau^\infty g_k(y) \mathrm{d}y$. Similarly, we can estimate the right-hand-side (RHS) probability term in Def. 2.1, and the EI-penalty $U_\delta$ is computed as the summation of the absolute difference of the two probability values (LHS and RHS) among all groups $z$.

**Loss-based EI penalty.** Another common way of approximating the fairness violation as a differentiable term is to compute the absolute difference of group-specific losses (Roh et al., 2021; Shen et al., 2022). Following the spirit of EI notion in Def. 2.1, we define EI loss of group $z$ as $\tilde{L}_z \triangleq \frac{1}{|I_{-,z}|} \sum_{i \in I_{-,z}} \ell \left( 1, \max_{\|\Delta \mathbf{x}_{\mathrm{I}_i}\| \le \delta} f(\mathbf{x}_i + \Delta \mathbf{x}_i) \right)$. Here, $\tilde{L}_z$ measures how far the rejected samples in group $z$ are away from being accepted after the feature improvement within budget $\delta$. Similarly, EI loss for all groups is written as $\tilde{L} \triangleq \sum_{z \in \mathcal{Z}} \frac{I_{-,z}}{I_-} \tilde{L}_z$. Finally, the EI penalty term is defined as $U_\delta \triangleq \sum_{z \in \mathcal{Z}} \left| \tilde{L}_z - \tilde{L} \right|$.

**Solving (1).** For each approach defined above (covariance-based, KDE-based and loss-based), the penalty term $U_\delta$ is defined uniquely. Note that in all cases, we need to solve a maximization problem $\max_{\|\Delta \mathbf{x}_{\mathrm{I}}\| \le \delta} f(\mathbf{x} + \Delta \mathbf{x})$ in order to get $U_\delta$. Since (1) is a minimization problem containing $U_\delta$ in the cost function, it is essentially a minimax problem. We leverage adversarial training techniques to solve (1). The inner maximization problem is solved using one of two methods: (i) derive the closed-form solution for generalized linear models, (ii) use projected gradient descent for general settings. The details can be found in Appendix B.4.

## 4 EXPERIMENTS

This section presents the empirical results of our EI fairness notion. To measure the fairness violation, we use EI disparity$= \max_{z \in [Z]} |\mathbb{P}(\max_{\mu(\Delta \mathbf{x}_{\mathrm{I}}) < \delta} f(\mathbf{x} + \Delta \mathbf{x}) \ge 0.5 \mid f(\mathbf{x}) < 0.5, \mathrm{z} = z) - \mathbb{P}(\max_{\mu(\Delta \mathbf{x}_{\mathrm{I}}) < \delta} f(\mathbf{x} + \Delta \mathbf{x}) \ge 0.5 \mid f(\mathbf{x}) < 0.5)|$. First, in Sec. 4.1, we show that our methods suggested in Sec. 3 achieve EI fairness in various real/synthetic datasets. Second, in Sec. 4.2, focusing on the dynamic scenario where each individual can make effort to improve its outcome, we demonstrate that training an EI classifier at each time step promotes achieving the long-term fairness, *i.e.*, the feature distribution of two groups become identical in the long run. In Appendix C.4, we put additional experimental results comparing the robustness of EI and related notions including ER and BE. Specifically, we compare (i) the outlier-robustness of EI and ER, and (ii) the robustness of EI and BE to imbalanced group negative rates.

### 4.1 SUGGESTED METHODS ACHIEVE EI FAIRNESS

Recall that Sec. 3 provided three approaches for achieving EI fairness. Here we check whether such methods successfully find a classifier with a small EI disparity, compared with ERM which does not have fairness constraints. We train all algorithms using logistic regression (LR) in this experiment. Due to the page limit, we defer the presentation of results for (1) a two-layer ReLU neural network with four hidden neurons and (2) a five-layer ReLU network with 200 hidden neurons per layer to Appendix C.3, and provide more details of hyperparameter selection in Appendix C.2.

**Experiment setting.** For all experiments, we use the Adam optimizer and cross-entropy loss. We perform cross-validation on the training set to find the best hyperparameter. We provide statistics for five trials having different random seeds. For the KDE-based approach, we use the Gaussian kernel.

**Datasets.** We perform the experiments on one synthetic dataset, and two real datasets: German Statlog Credit (Dua & Graff, 2017), and ACSIncome-CA (Ding et al., 2021). The synthetic dataset has two non-sensitive attributes $\mathbf{x} = (\mathrm{x}_1, \mathrm{x}_2)$, one binary sensitive attribute z, and a binary label y. Both features $\mathrm{x}_1$ and $\mathrm{x}_2$ are assumed to be improvable. We generate 20,000 samples where $(\mathbf{x}, \mathrm{y}, \mathrm{z})$ pair of each sample is generated independently as below. We define z and $(\mathrm{y}|\mathrm{z} = z)$ as Bernoulli random variables for all $z \in \{0, 1\}$, and define $(\mathbf{x}|\mathrm{y} = y, \mathrm{z} = z)$ as multivariate Gaussian random variables for all $y, z \in \{0, 1\}$. The numerical details are in Appendix C.1 The maximum effort $\delta$ for this dataset is set to 0.5. The ratio of the training versus test data is 4:1.

German Statlog Credit dataset contains 1,000 samples and the ratio of the training versus test data is 4:1. The task is to predict the credit risk of an individual given its financial status. Following Jiang & Nachum (2020), we divide the samples into two groups using the age of thirty as the boundary, *i.e.*, $z = 1$ for samples with age above thirty. Four features $\mathbf{x}$ are considered as improvable: `checking account`, `saving account`, `housing` and `occupation`, all of which are ordered categorical features. For example, the `occupation` feature has four levels: (1) unemployed, (2) unskilled, (3)

Table 2: **Error rate and EI disparities of ERM and three proposed EI-regularized methods on logistic regression (LR).** For each dataset, the lowest EI disparity (disp.) value is in boldface. Classifiers obtained by our three methods have much smaller EI disparity values than the ERM solution, without having much additional error.

| DATASET | METRIC | ERM | METHODS | | |
| --- | --- | --- | --- | --- | --- |
| | | | COVARIANCE-BASED | KDE-BASED | LOSS-BASED |
| SYNTHETIC | ERROR RATE($\downarrow$) | $.221 \pm .001$ | $.253 \pm .003$ | $.250 \pm .001$ | $.246 \pm .001$ |
| | EI DISP.($\downarrow$) | $.117 \pm .007$ | $.003 \pm .001$ | $.005 \pm .003$ | $\mathbf{.002 \pm .001}$ |
| GERMAN STAT. | ERROR RATE($\downarrow$) | $.220 \pm .009$ | $.262 \pm .009$ | $.243 \pm .024$ | $.237 \pm .008$ |
| | EI DISP.($\downarrow$) | $.041 \pm .008$ | $.021 \pm .019$ | $.035 \pm .026$ | $\mathbf{.015 \pm .009}$ |
| ACSINCOME-CA | ERROR RATE($\downarrow$) | $.184 \pm .000$ | $.200 \pm .000$ | $.196 \pm .000$ | $.193 \pm .000$ |
| | EI DISP.($\downarrow$) | $.031 \pm .001$ | $.008 \pm .001$ | $\mathbf{.005 \pm .001}$ | $.006 \pm .001$ |

skilled, and (4) highly qualified. We set the maximum effort $\delta = 1$, meaning that an unskilled man (with level 2) can become a skilled man, but cannot be a highly qualified man.

ACSIncome-CA dataset consists of data for 195,665 people and is split into training/test sets in the ratio of 4:1. The task is predicting whether a person's income would exceed 50K USD per year. We use sex as the sensitive attribute; we have two sensitive groups, male and female. We select education level (ordered categorical feature) as the improvable feature. We set the maximum effort $\delta = 3$.

**Results.** Table 2 shows the test error rate and test EI disparity (disp.) for ERM and our three EI-regularized methods (covariance-based, KDE-based, and loss-based) suggested in Sec. 3. For all three datasets, our EI-regularized methods successfully reduce the EI disparity without increasing the error rate too much, compared with ERM. Figure 3 shows the tradeoff between the error rate and EI disparity of our EI-regularized methods. We marked the dots after running each method multiple times with different penalty coefficients $\lambda$, and plotted the

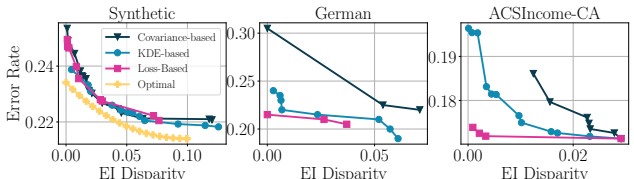

Figure 3: **Tradeoff between EI disparity and error rate.** We run three EI-regularized methods suggested in Sec. 3 for different regularizer coefficient $\lambda$ and plot the frontier lines. For the synthetic dataset, the tradeoff curve for the ideal classifier is located at the bottom left corner, which is similar to the curves of proposed EI-regularized methods. This shows that our methods successfully find classifiers balancing EI disparity and error rate.

frontier line. For the synthetic dataset with the Gaussian feature, we numerically obtained the performance of the optimal EI classifier, which is added in the yellow line at the bottom left corner of the first column plot. The details of finding the optimal EI classifier is in Appendix B.5. One can confirm that our methods of regularizing EI are having similar tradeoff curves for the synthetic dataset. Especially, for the synthetic dataset, the tradeoff curve of our methods nearly achieves that of the optimal EI classifier. For German and ACSIncome-CA datasets, the loss-based method is having a slightly better tradeoff curve than other methods.

### 4.2 EI PROMOTES LONG-TERM FAIRNESS IN DYNAMIC SCENARIOS

Here we provide simulation results on the dynamic setting, showing that EI classifier encourages the *long-term fairness*, *i.e.*, equalizes the feature distribution of different groups in the long run [3].

#### 4.2.1 DYNAMIC SYSTEM DESCRIPTION

We consider a binary classification problem under the dynamic scenario with $T$ rounds, where the improvable feature $\mathbf{x}_t \in \mathbb{R}$ and the label $\mathbf{y}_t \in \{0, 1\}$ of each sample as well as the classifier $f_t$ evolve at each round $t \in \{0, \cdots, T-1\}$. We denote the sensitive attribute as $\mathbf{z} \in \{0, 1\}$, and the estimated label as $\hat{\mathbf{y}}_t$. We assume $\mathbf{z} \sim \text{Bern}(0.5)$ and $(\mathbf{x}_t \mid \mathbf{z} = z) \sim \mathcal{P}_t^{(z)} = \mathcal{N}(\mu_t^{(z)}, \{\sigma_t^{(z)}\}^2)$. To mimic the admission problem, we only accept a fraction $\alpha \in (0, 1)$ of the population, *i.e.*, the true label is modeled as $\mathbf{y}_t = \mathbf{1}\{\mathbf{x}_t \geq \chi_\alpha^{(t)}\}$, where $\chi_\alpha^{(t)}$ is the $(1-\alpha)$ percentile of the feature distribution at round $t$. We consider z-aware linear classifier outputting $\hat{\mathbf{y}}_t = \mathbf{1}\{\mathbf{x}_t \geq \tau_t^{(z)}\}$, which is parameterized by the thresholds $(\tau_t^{(0)}, \tau_t^{(1)})$ for two groups. Note that this classification rule is equivalent to defining score function $f_t(x, z) = 1/(\exp(\tau_t^{(z)} - x) + 1)$ and $\hat{\mathbf{y}}_t = \mathbf{1}\{f(\mathbf{x}_t, \mathbf{z}) \geq 0.5\}$.

**Updating data parameters** $(\mu_t^{(z)}, \sigma_t^{(z)})$. At each round $t$, we allow each sample can improve its feature from $x$ to $x + \epsilon(x)$. Here we model $\epsilon(x) = \nu(x; z) = \frac{1}{(\tau_t^{(z)} - x + \beta)^2}\mathbf{1}\{x < \tau_t^{(z)}\}$

---

[3]Appendix D provides analysis on the effect of fairness notions on the long-term fairness when a single step of feature improvement is applied, for toy examples. Our results show that EI better equalizes the feature distribution (compared with other fairness notions), which coincides with our empirical results in Sec. 4.2.

Figure 4: **Long-term unfairness** $d_{\text{TV}}(\mathcal{P}_t^{(0)}, \mathcal{P}_t^{(1)})$ **at each round** $t$ **for various algorithms.** Consider the binary classification problem over two groups, under the dynamic scenario where the data distribution and the classifier for each group evolve over multiple rounds. We plot how the long-term unfairness (measured by the total variation distance between two groups) changes as round $t$ increases. Here, each column shows the result for different initial feature distributions, details of which are given in Sec. 4.2.2. The long-term unfairness of EI classifier reduces faster than other existing fairness notions, showing that EI proposed in this paper is helpful for achieving long-term fairness.

for a constant $\beta > 0$. In this model, the rejected samples with larger gap $(\tau_t^{(z)} - x)$ with the decision boundary are making less effort $\Delta x$, which is inspired by the intuition that a rejected sample is less motivated to improve its feature if it needs to take a large amount of effort to get accepted in one scoop. More detailed justification of the $\epsilon(\cdot)$ selection is provided in Appendix C.5. After such effort is made, we compute the mean and standard deviation of each group: $\mu_{t+1}^{(z)} = \int_{-\infty}^{\infty}(x + \nu(x;z))\phi(x;\mu_t^{(z)}, \sigma_t^{(z)})\mathrm{d}x$ and $\sigma_{t+1}^{(z)} = \sqrt{\int_{-\infty}^{\infty}(x + \nu(x;z) - \mu_{t+1}^{(z)})^2\phi(x;\mu_t^{(z)}, \sigma_t^{(z)})\mathrm{d}x}$, where $\phi(\cdot;\mu,\sigma)$ is the pdf of $\mathcal{N}(\mu, \sigma^2)$. We assume that the feature $\mathbf{x}_{t+1}$ in the next round follows a Gaussian distribution parameterized by $(\mu_{t+1}^{(z)}, \sigma_{t+1}^{(z)})$ for ease of simulation.

**Updating classifier parameters** $(\tau_t^{(0)}, \tau_t^{(1)})$**.** At each round $t$, we update the classifier depending on the current feature distribution $\mathbf{x}_t$. The EI classifier considered in this paper updates $(\tau_t^{(0)}, \tau_t^{(1)})$ as below. Note that the maximized score $\max_{\|\Delta\mathbf{x}_\mathbf{I}\|\leq\delta} f(\mathbf{x} + \Delta\mathbf{x})$ in Def. 2.1 can be written as $f_t(\mathbf{x}_t + \delta_t, z)$, and the equation $\max_{\|\Delta\mathbf{x}_\mathbf{I}\|\leq\delta} f(\mathbf{x} + \Delta\mathbf{x}) \geq 0.5$ is equivalent to $\mathbf{x}_t + \delta_t > \tau_t^{(z)}$. Consequently, EI classifier obtains $(\tau_t^{(0)}, \tau_t^{(1)})$ by solving

$$\min_{\tau_t^{(0)}, \tau_t^{(1)}} \left|\mathbb{P}(\mathbf{x}_t + \delta_t > \tau_t^{(0)} \,|\, \mathbf{z} = 0, \mathbf{x}_t < \tau_t^{(0)}) - \mathbb{P}(\mathbf{x}_t + \delta_t > \tau_t^{(1)} \,|\, \mathbf{z} = 1, \mathbf{x}_t < \tau_t^{(1)})\right| \text{ s.t. } \mathbb{P}(\hat{\mathbf{y}}_t \neq \mathbf{y}_t) \leq c,$$

where $c \in [0, 1)$ is the maximum classification error rate we allow, and $\delta_t$ is the effort level at iteration $t$. In our experiments, $\delta_t$ is chosen as the mean efforts the population makes, *i.e.*, $\delta_t = 0.5\sum_{z=0}^1 \int_{-\infty}^{\infty} \nu(x;z)\phi(x;\mu_t^{(z)}, \sigma_t^{(z)})\mathrm{d}x$. Meanwhile, we can similarly obtain the classifier for DP, BE, ER, and ILFCR constraints, details of which are in Appendix C.6. In the experiments, we numerically obtain the solution of this optimization problem.

### 4.2.2 EXPERIMENTS ON LONG-TERM FAIRNESS

We first initialize the feature distribution in a way that both sensitive groups have either different mean (*i.e.*, $\mu_0^{(0)} \neq \mu_0^{(1)}$) or different variance (*i.e.*, $\sigma_0^{(0)} \neq \sigma_0^{(1)}$). At each round $t \in \{1, \cdots, T\}$, we update the data parameter $(\mu_t^{(z)}, \sigma_t^{(z)})$ for group $z \in \{0, 1\}$ and the classifier parameter $(\tau_t^{(0)}, \tau_t^{(1)})$, following the rule described in Sec. 4.2.1. At each round $t \in \{1, \cdots, T\}$, we measure the *long-term unfairness* defined as the total variation distance between the two group distributions: $d_{\text{TV}}(\mathcal{P}^{(0)}, \mathcal{P}^{(1)}) = \frac{1}{2}\int_{-\infty}^{\infty} |\phi(x;\mu_t^{(0)}, \sigma_t^{(0)}) - \phi(x;\mu_t^{(1)}, \sigma_t^{(1)})|\mathrm{d}x$. We run experiments on four different initial feature distributions: (i) $(\mu_0^{(0)}, \sigma_0^{(0)}, \mu_0^{(1)}, \sigma_0^{(1)}) = (0, 1, 1, 0.5)$, (ii) $(\mu_0^{(0)}, \sigma_0^{(0)}, \mu_0^{(1)}, \sigma_0^{(1)}) = (0, 0.5, 1, 1)$, (iii) $(\mu_0^{(0)}, \sigma_0^{(0)}, \mu_0^{(1)}, \sigma_0^{(1)}) = (0, 2, 0, 1)$, and (iv) $(\mu_0^{(0)}, \sigma_0^{(0)}, \mu_0^{(1)}, \sigma_0^{(1)}) = (0, 0.5, 1, 0.5)$, respectively. We set $\alpha = 0.2, c = 0.1, \beta = 0.25$.

**Baselines.** We compare our EI classifier with multiple baselines, including the empirical risk minimization (ERM) and algorithms with fairness constraints: demographic parity (DP), bounded effort (BE) (Heidari et al., 2019), equal recourse (ER) (Gupta et al., 2019), and individual-level fair causal recourse (ILFCR) (Von Kügelgen et al., 2022). In particular, we assume a causal model for solving the ILFCR classifier for this specific data distribution, which is described in detail in Appendix C.6.

**Results.** Fig. 4 shows how the long-term unfairness $d_{\text{TV}}(\mathcal{P}_t^{(0)}, \mathcal{P}_t^{(1)})$ changes as a function of round $t$, for cases (i) – (iv) having different initial feature distribution. Note that except ILFCR, other fairness notions (DP, BE, ER, and EI) yield lower long-term unfairness compared with ERM, for cases (i), (ii), and (iv). More importantly, EI accelerates the process of mitigating long-term unfairness,

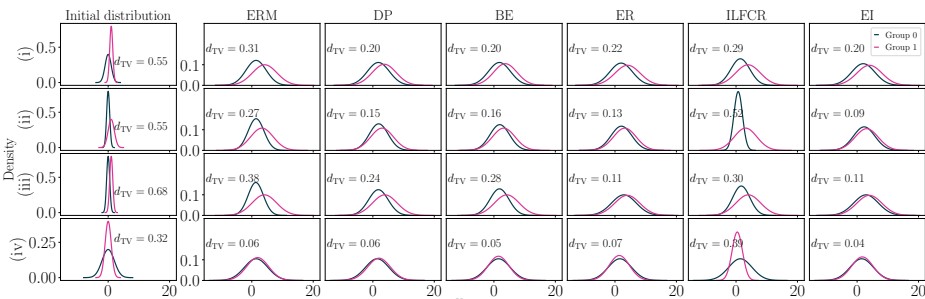

Figure 5: **Evolution of the feature distribution, when we apply each algorithm for $t = 3$ rounds.** At each row, the leftmost column shows the initial distribution and the rest of the columns show the evolved distribution for each algorithm, under the dynamic setting. Compared with existing fairness notions (DP, BE, ER, and ILFCR), EI achieves a smaller feature distribution gap between groups.

compared to other fairness notions. This observation highlights the benefit of EI in promoting true equity of groups in the long run. Fig. 5 visualizes the initial distribution (at the leftmost column) and the evolved distribution at round $t = 3$ for multiple algorithms (at the rest of the columns). Each row represents different initial feature distribution, for cases (i) – (iv). One can confirm that EI brings the distribution of the two groups closer, compared with baselines. Moreover, in Appendix C.7, we explore the long-term impact of fairness notions on a different dynamic model, where most existing methods have an adverse effect on long-term fairness, while EI continues to enhance it.

## 5 RELATED WORKS

**Fairness-aware algorithms.** Most of the existing fair learning techniques fall into three categories: i) pre-processing approaches (Kamiran & Calders, 2012; 2010; Gordaliza et al., 2019; Jiang & Nachum, 2020), which primarily involves massaging the dataset to remove the bias; ii) in-processing approaches (Fukuchi et al., 2013; Kamishima et al., 2012; Calders & Verwer, 2010; Zafar et al., 2017c;a; Zhang et al., 2018; Cho et al., 2020; Roh et al., 2020; 2021; Shen et al., 2022), adjusting the model training for fairness; iii) post-processing approaches (Calders & Verwer, 2010; Alghamdi et al., 2020; Wei et al., 2020; Hardt et al., 2016) which achieve fairness by modifying a given unfair classifier. Prior work (Woodworth et al., 2017) showed that the in-processing approach generally outperforms other approaches due to its flexibility. Hence, we focus on the in-processing approach and propose three methods to achieve EI. These methods achieve EI by solving fairness-regularized optimization problems. In particular, our proposed fairness regularization terms are inspired by Zafar et al. (2017c); Cho et al. (2020); Roh et al. (2021); Shen et al. (2022).

**Fairness notions related with EI** See Table 1 and the corresponding explanation given in Sec. 2.

**Fairness dynamics.** There are also a few attempts to study the long-term impact of different fairness policies (Zhang et al., 2020; Heidari et al., 2019; Hashimoto et al., 2018). In particular, Hashimoto et al. (2018) studies how ERM amplifies the unfairness of a classifier in the long run. The key idea is that if the classifier of the previous iteration favors a certain candidate group, then the candidate groups will be more unbalanced since fewer individuals from the unfavored group will apply for this position. Thus, the negative feedback leads to a more unfair classifier. In contrast, Heidari et al. (2019) and Zhang et al. (2020) focus more on long-term impact instead of classification fairness. To be specific, Heidari et al. (2019) studies how fairness intervention affects the different groups in terms of evenness, centralization, and clustering by simulating the population's response through effort. Zhang et al. (2020) investigates how different fairness policies affect the gap between the qualification rates of different groups under a partially observed Markov decision process. Besides, there are a few works which study how individuals may take strategic actions to improve their outcomes given a classifier (Chen et al., 2021). However, Chen et al. (2021) aims to address this problem by designing an optimization problem that is robust to strategic manipulation, which is orthogonal to our focus.

## 6 CONCLUSION

In this paper, we introduce Equal Improvability (EI), a group fairness notion that equalizes the potential acceptance of rejected samples in different groups when appropriate effort is made by the rejected samples. We analyze the properties of EI and provide three approaches to finding a classifier that achieves EI. While our experiments demonstrate the effectiveness of the proposed approaches in reducing EI disparity, the theoretical analysis of the approximated EI penalties remains open. Moreover, we formulate a simplified dynamic model with one-dimensional features and a binary sensitive attribute to showcase the benefits of EI in promoting equity in feature distribution across different groups. We identify extending this model to settings with multiple sensitive attributes and high-dimensional features as an interesting future direction.

ACKNOWLEDGEMENT

Yuchen Zeng was supported in part by NSF Award DMS-2023239. Ozgur Guldogan and Ramtin Pedarsani were supported by NSF Award CNS-2003035, and NSF Award CCF-1909320. Kangwook Lee was supported by NSF/Intel Partnership on Machine Learning for Wireless Networking Program under Grant No. CNS-2003129 and by the Understanding and Reducing Inequalities Initiative of the University of Wisconsin-Madison, Office of the Vice Chancellor for Research and Graduate Education with funding from the Wisconsin Alumni Research Foundation. The authors would also like to thank the anonymous reviewers and AC for their valuable suggestions.

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

# A    THEORETICAL RESULTS

## A.1    CONNECTIONS BETWEEN EI, DP, AND BE

In this section, we provide the proof of Theorem 2.5 and Corollary 2.6.

*Proof of Theorem 2.5.* All we need to prove are three statements:

1. Prove that EI and BE imply DP

2. Prove that DP and EI imply BE

3. Prove that BE and DP imply EI

Below we prove each statement.

**1. EI, BE $\Rightarrow$ DP**    Suppose a classifier $f$ achieves EI and BE. Recall that a classifier achieves EI if

$$\mathbb{P}\left(\max_{\mu(\Delta\mathbf{x}_{\mathrm{I}})\leq\delta} f(\mathbf{x}+\Delta\mathbf{x})\geq 0.5 \mid f(\mathbf{x})<0.5, \mathbf{z}=z\right)=\mathbb{P}\left(\max_{\mu(\Delta\mathbf{x}_{\mathrm{I}})\leq\delta} f(\mathbf{x}+\Delta\mathbf{x})\geq 0.5 \mid f(\mathbf{x})<0.5\right) \tag{2}$$

and a classifier achieves BE if

$$\mathbb{P}\left(\max_{\mu(\Delta\mathbf{x}_{\mathrm{I}})\leq\delta} f(\mathbf{x}+\Delta\mathbf{x})\geq 0.5, f(\mathbf{x})<0.5 \mid \mathbf{z}=z\right)=\mathbb{P}\left(\max_{\mu(\Delta\mathbf{x}_{\mathrm{I}})\leq\delta} f(\mathbf{x}+\Delta\mathbf{x})\geq 0.5, f(\mathbf{x})<0.5\right) \tag{3}$$

By dividing both sides of 3 by the both sides of 2, we have

$$\frac{\mathbb{P}\left(\max_{\mu(\Delta\mathbf{x}_{\mathrm{I}})\leq\delta} f(\mathbf{x}+\Delta\mathbf{x})\geq 0.5, f(\mathbf{x})<0.5 \mid \mathbf{z}=z\right)}{\mathbb{P}\left(\max_{\mu(\Delta\mathbf{x}_{\mathrm{I}})\leq\delta} f(\mathbf{x}+\Delta\mathbf{x})\geq 0.5 \mid f(\mathbf{x})<0.5, \mathbf{z}=z\right)} = \frac{\mathbb{P}\left(\max_{\mu(\Delta\mathbf{x}_{\mathrm{I}})\leq\delta} f(\mathbf{x}+\Delta\mathbf{x})\geq 0.5, f(\mathbf{x})<0.5\right)}{\mathbb{P}\left(\max_{\mu(\Delta\mathbf{x}_{\mathrm{I}})\leq\delta} f(\mathbf{x}+\Delta\mathbf{x})\geq 0.5 \mid f(\mathbf{x})<0.5\right)}$$

Then, it can be simplified as

$$\mathbb{P}\left(f(\mathbf{x})<0.5 \mid \mathbf{z}=z\right)=\mathbb{P}\left(f(\mathbf{x})<0.5\right),$$

which implies that the classifier achieves demographic parity,

$$\mathbb{P}\left(f(\mathbf{x})\geq 0.5 \mid \mathbf{z}=z\right)=\mathbb{P}\left(f(\mathbf{x})\geq 0.5\right)$$

**2. DP, EI $\Rightarrow$ BE**    Suppose a classifier $f$ achieves DP and EI. Recall that a classifier achieves DP if

$$\mathbb{P}\left(f(\mathbf{x})\geq 0.5 \mid \mathbf{z}=z\right)=\mathbb{P}\left(f(\mathbf{x})\geq 0.5\right),$$

which implies

$$\mathbb{P}\left(f(\mathbf{x})<0.5 \mid \mathbf{z}=z\right)=\mathbb{P}\left(f(\mathbf{x})<0.5\right). \tag{4}$$

Recall that a classifier achieves EI if

$$\mathbb{P}\left(\max_{\mu(\Delta\mathbf{x}_{\mathrm{I}})\leq\delta} f(\mathbf{x}+\Delta\mathbf{x})\geq 0.5 \mid f(\mathbf{x})<0.5, \mathbf{z}=z\right)=\mathbb{P}\left(\max_{\mu(\Delta\mathbf{x}_{\mathrm{I}})\leq\delta} f(\mathbf{x}+\Delta\mathbf{x})\geq 0.5 \mid f(\mathbf{x})<0.5\right) \tag{5}$$

By multiplying both sides of 4 and 5, we have

$$\mathbb{P}\left(\max_{\mu(\Delta\mathbf{x}_{\mathrm{I}})\leq\delta} f(\mathbf{x}+\Delta\mathbf{x})\geq 0.5 \mid f(\mathbf{x})<0.5, \mathbf{z}=z\right)\mathbb{P}\left(f(\mathbf{x})<0.5 \mid \mathbf{z}=z\right)$$

$$=\mathbb{P}\left(\max_{\mu(\Delta\mathbf{x}_{\mathrm{I}})\leq\delta} f(\mathbf{x}+\Delta\mathbf{x})\geq 0.5 \mid f(\mathbf{x})<0.5\right)\mathbb{P}\left(f(\mathbf{x})<0.5\right)$$

Then, it can be simplified as

$$\mathbb{P}\left(\max_{\mu(\Delta\mathbf{x}_{\mathrm{I}})\leq\delta} f(\mathbf{x}+\Delta\mathbf{x})\geq 0.5, f(\mathbf{x})<0.5 \mid \mathbf{z}=z\right)=\mathbb{P}\left(\max_{\mu(\Delta\mathbf{x}_{\mathrm{I}})\leq\delta} f(\mathbf{x}+\Delta\mathbf{x})\geq 0.5, f(\mathbf{x})<0.5\right),$$

which implies that the classifier $f$ achieves BE.

**3. BE, DP $\Rightarrow$ EI**  Suppose a classifier $f$ achieves BE and DP. Recall that a classifier achieves DP if

$$\mathbb{P}\left(f(\mathbf{x}) \geq 0.5 \mid \mathbf{z} = z\right) = \mathbb{P}\left(f(\mathbf{x}) \geq 0.5\right),$$

which implies

$$\mathbb{P}\left(f(\mathbf{x}) < 0.5 \mid \mathbf{z} = z\right) = \mathbb{P}\left(f(\mathbf{x}) < 0.5\right) \tag{6}$$

Recall that a classifier achieves BE if

$$\mathbb{P}\left(\max_{\mu(\Delta\mathbf{x}_\mathrm{I}) \leq \delta} f(\mathbf{x} + \Delta\mathbf{x}) \geq 0.5, f(\mathbf{x}) < 0.5 \mid \mathbf{z} = z\right) = \mathbb{P}\left(\max_{\mu(\Delta\mathbf{x}_\mathrm{I}) \leq \delta} f(\mathbf{x} + \Delta\mathbf{x}) \geq 0.5, f(\mathbf{x}) < 0.5\right). \tag{7}$$

By dividing both sides of 7 by the both sides of 6, we have

$$\frac{\mathbb{P}\left(\max_{\mu(\Delta\mathbf{x}_\mathrm{I}) \leq \delta} f(\mathbf{x} + \Delta\mathbf{x}) \geq 0.5, f(\mathbf{x}) < 0.5 \mid \mathbf{z} = z\right)}{\mathbb{P}\left(f(\mathbf{x}) < 0.5 \mid \mathbf{z} = z\right)} = \frac{\mathbb{P}\left(\max_{\mu(\Delta\mathbf{x}_\mathrm{I}) \leq \delta} f(\mathbf{x} + \Delta\mathbf{x}) \geq 0.5, f(\mathbf{x}) < 0.5\right)}{\mathbb{P}\left(f(\mathbf{x}) < 0.5\right)}$$

Then, it can be simplified as

$$\mathbb{P}\left(\max_{\mu(\Delta\mathbf{x}_\mathrm{I}) \leq \delta} f(\mathbf{x} + \Delta\mathbf{x}) \geq 0.5 \mid f(\mathbf{x}) < 0.5, \mathbf{z} = z\right) = \mathbb{P}\left(\max_{\mu(\Delta\mathbf{x}_\mathrm{I}) \leq \delta} f(\mathbf{x} + \Delta\mathbf{x}) \geq 0.5 \mid f(\mathbf{x}) < 0.5\right),$$

which implies that the classifier $f$ achieves EI. $\qquad\square$

*Proof of Corollary 2.6.*  The Corollary 2.6 can be proved directly from Theorem 2.5. $\qquad\square$

### A.2  CONNECTIONS BETWEEN EI AND ER

In this part, we investigate the connections between EI and ER.

**Lemma A.1.** *Consider* $\mathbf{x} \mid \mathbf{z} = z \sim \mathcal{N}(\mu_z, \sigma^2)$ *for* $z \in \{0, 1\}$, $\mu_z, \sigma \in \mathbb{R}$, *and classifiers characterized by two accepting thresholds* $(\tau_0, \tau_1)$, *where* $\tau_0, \tau_1 \in \mathbb{R}$. *If a classifier satisfies EI, then it satisfies ER.*

*Proof.* Here we use $\Phi = 1 - Q$ and $\phi$ to denote the CDF and PDF of standard Gaussian distribution, respectively. We consider the cost function $\mu = |\cdot|$.

Recall the definition of EI disparity and ER disparity

$$\text{EI Disparity} = \left| \mathbb{P}\left( \underbrace{\max_{\mu(\Delta\mathbf{x}) < \delta} f(\mathbf{x} + \Delta\mathbf{x}) > 0.5}_{\mathbf{x} > \tau_0 - \delta} \mid \underbrace{f(\mathbf{x}) < 0.5}_{\mathbf{x} \leq \tau_0}, \mathbf{z} = 0 \right), \right.$$

$$\left. - \mathbb{P}\left( \max_{\mu(\Delta\mathbf{x}) < \delta} f(\mathbf{x} + \Delta\mathbf{x}) > 0.5 \mid f(\mathbf{x}) < 0.5, \mathbf{z} = 1 \right) \right|$$

$$\text{ER Disparity} = \left| \mathbb{E}\left[ \underbrace{\min_{f(\mathbf{x}+\Delta\mathbf{x}) \geq 0.5} \mu(\Delta\mathbf{x})}_{\tau_0 - \mathbf{x}} \mid \underbrace{f(\mathbf{x}) < 0.5}_{\mathbf{x} \leq \tau_0}, \mathbf{z} = 0 \right] \right.$$

$$\left. - \mathbb{E}\left[ \min_{f(\mathbf{x}+\Delta\mathbf{x}) \geq 0.5} \mu(\Delta\mathbf{x}) \mid f(\mathbf{x}) < 0.5, \mathbf{z} = 1 \right] \right|.$$

Consequently, the EI constraint and ER constraint can be written as

$$\text{EI Disparity}\,(\tau_0, \tau_1) = \left| \Phi\left(\frac{\tau_0 - \delta - \mu_0}{\sigma}\right) \Big/ \Phi\left(\frac{\tau_0 - \mu_0}{\sigma}\right) - \right.$$

$$\left. \Phi\left(\frac{\tau_1 - \delta - \mu_1}{\sigma}\right) \Big/ \Phi\left(\frac{\tau_1 - \mu_1}{\sigma}\right) \right| = 0, \tag{8}$$

$$\text{ER Disparity}\,(\tau_0, \tau_1) = \left| \frac{1}{\Phi((\tau_0 - \mu_0)/\sigma)} \int_{-\infty}^{\tau_0} (\tau_0 - t)\phi\left(\frac{t - \mu_0}{\sigma}\right) \mathrm{d}t - \right.$$

$$\left. \frac{1}{\Phi((\tau_1 - \mu_1)/\sigma)} \int_{-\infty}^{\tau_1} (\tau_1 - t)\phi\left(\frac{t - \mu_1}{\sigma}\right) \mathrm{d}t \right| = 0 \tag{9}$$

In this proof, we will first show that achieving EI is equivalent to $\tau_0 - \mu_0 = \tau_1 - \mu_1$, and then show that the classifier with $\tau_0 - \mu_0 = \tau_1 - \mu_1$ satisfies the ER constraint.

1. EI constraint.

   Let $\varphi(x) = \Phi(\frac{x-\delta}{\sigma})/\Phi(\frac{x}{\sigma})$. First, we show that $\varphi$ is a strictly increasing function. Note that

   $$\varphi'(x) = \frac{1}{\sigma\Phi\left(\frac{x}{\sigma}\right)^2}\left(\phi\left(\frac{x-\delta}{\sigma}\right)\Phi\left(\frac{x}{\sigma}\right) - \Phi\left(\frac{x-\delta}{\sigma}\right)\phi\left(\frac{x}{\sigma}\right)\right).$$

   Therefore, to show that $\varphi$ is strictly increasing, it is sufficient to show that

   $$\phi\left(\frac{x-\delta}{\sigma}\right)\Phi\left(\frac{x}{\sigma}\right) > \Phi\left(\frac{x-\delta}{\sigma}\right)\phi\left(\frac{x}{\sigma}\right). \tag{10}$$

   We show that (10) is equivalent as the following inequality by dividing both the left-hand side and right-hand side by $\phi(\frac{x-\delta}{\sigma})\phi(\frac{x}{\sigma})$:

   $$\Phi\left(\frac{x}{\sigma}\right)/\phi\left(\frac{x}{\sigma}\right) > \Phi\left(\frac{x-\delta}{\sigma}\right)/\phi\left(\frac{x-\delta}{\sigma}\right). \tag{11}$$

   Note that $\frac{1-\Phi(\cdot)}{\phi(\cdot)}$ is known in literatures as Mills' ratio (Mitrinovic & Vasic, 1970), which is strictly decreasing on $\mathbb{R}$. Therefore, $\frac{\Phi(\cdot)}{\phi(\cdot)}$ is strictly increasing on $\mathbb{R}$. Since $\frac{x}{\sigma} > \frac{x-\delta}{\sigma}$, (11) holds, thereby (10) holds and $\varphi$ is strictly increasing.

   Given that $\varphi(x) = \Phi(\frac{x-\delta}{\sigma})/\Phi(\frac{x}{\sigma})$ is a strictly increasing function on $\mathbb{R}$,

   $$(8) = |\varphi(\tau_0 - \mu_0) - \varphi(\tau_1 - \mu_1)| = 0$$

   yields that

   $$\tau_0 - \mu_0 = \tau_1 - \mu_1.$$

2. ER constraint.

   We first note that

   $$\int_{-\infty}^{\tau_0}(\tau_0 - t)\phi\left(\frac{t-\mu_0}{\sigma}\right)\mathrm{d}t \xrightarrow{t'=t-\mu_0} \int_{-\infty}^{\tau_0-\mu_0}(\tau_0 - \mu_0 - t')\phi\left(\frac{t'}{\sigma}\right)\mathrm{d}t'$$

   Let $\psi(x) = \frac{1}{\Phi(x/\sigma)}\int_{-\infty}^{x}(x-t)\phi(\frac{t}{\sigma})\mathrm{d}t$. It is clear that ER constraint is equivalent to

   $$(9) = |\psi(\tau_0 - \mu_0) - \psi(\tau_1 - \mu_1)|.$$

   Therefore, the classifier with $\tau_0 - \mu_0 = \tau_1 - \mu_1$ clearly satisfies the ER constraint.

Combining all the discussion above completes the proof. □

## B  SUPPLEMENTARY MATERIALS ON THE FAIRNESS NOTIONS

Recall that in Sec. 2 and Sec. 3, this paper proposes a new fairness notion called equal improvability (EI), compares it with other existing effort-based fairness notions, and finds a classifier by solving a EI-constrained optimization which is formulated as a minimax problem. In Sec. B.1, we first explain what each term in the definition of EI means. Then, we provide more details of reward function selection of bounded effort (BE) (Heidari et al., 2019) and discuss the vulnerability of equal recourse (ER) (Gupta et al., 2019) in Sec. B.2 and B.3, respectively Then in Sec. B.4, we provide how we solved the inner maximization problem in the EI-constrained optimization. Finally, in Sec. B.5, we provide numerical methods for finding the optimal solution for the EI-constrained problem, under simple synthetic dataset setting.

### B.1 MEANING OF EACH TERM IN THE DEFINITION OF EI

To help readers better understand our EI definition, here we explain what each term means in EI definition means. Let the data sample $(\mathbf{x}, \mathrm{y}, \mathrm{z})$ follows the distribution $\mathcal{P}_{(\mathbf{x},\mathrm{y},\mathrm{z})}$. The meaning of each term of EI defined in Def. 2.1 is detailed below.

$$\underbrace{\mathbb{P}_{\mathbf{x},\,\mathrm{y},\,\mathrm{z}\,\sim\,\mathcal{P}_{\mathbf{x},\mathrm{y},\mathrm{z}}}_{\substack{\text{Randomness is over}\\\text{the data distribution}}}\Bigg(\underbrace{\overbrace{\max_{\mu(\Delta\mathbf{x}_I)\leq\delta}f(\mathbf{x}+\Delta\mathbf{x})\geq 0.5}^{\substack{\text{Maximum score}\\\text{after improvement}}}\Bigg|\underbrace{f(\mathbf{x})<0.5,\mathrm{z}=z}_{\substack{\text{Event in which the sample}\\\text{comes from group }z\\\text{and gets rejected}}}\Bigg)}_{\substack{\text{Event in which the sample can}\\\text{be accepted after improvement}}}$$

$$=\underbrace{\mathbb{P}_{\mathbf{x},\,\mathrm{y},\,\mathrm{z}\,\sim\,\mathcal{P}_{\mathbf{x},\mathrm{y},\mathrm{z}}}_{\substack{\text{Randomness is over}\\\text{the data distribution}}}\Bigg(\underbrace{\overbrace{\max_{\mu(\Delta\mathbf{x}_I)\leq\delta}f(\mathbf{x}+\Delta\mathbf{x})\geq 0.5}^{\substack{\text{Maximum score}\\\text{after improvement}}}\Bigg|\underbrace{f(\mathbf{x})<0.5}_{\substack{\text{Event in which the}\\\text{sample gets rejected}}}\Bigg)}_{\substack{\text{Event in which the sample can}\\\text{be accepted after improvement}}},$$

### B.2 SPECIFYING THE REWARD FUNCTION FOR BE

In this part, we introduce how we set the reward function so that we can connect BE with EI. Recall that the reward function in Heidari et al. (2019) is defined as the benefit gained by changing an individual's characteristics from $w = (\mathbf{x}, y)$ to $w' = (\mathbf{x}', y')$:

$$\mathcal{R}(w, w') = b(f(\mathbf{x}'), y') - b(f(\mathbf{x}), y)$$

where $b$ is the benefit function, and $\mathbf{x}' = \mathbf{x} + \Delta\mathbf{x}$ is the updated feature. If we set the benefit function as $b(f(\mathbf{x}), y) = \mathbf{1}\{f(\mathbf{x}) \geq 0.5\}$, then the reward function becomes:

$$\mathcal{R}(w, w') = \mathbf{1}\{f(\mathbf{x} + \Delta\mathbf{x}) \geq 0.5\} - \mathbf{1}\{f(\mathbf{x}) \geq 0.5\}$$

Then, the bounded effort fairness is defined as:

$$\mathbb{E}\left[\max_{\Delta\mathbf{x}}\mathcal{R}(w, w') \text{ s.t. } \mu(\Delta\mathbf{x}) < \delta|\mathrm{z} = z\right] = \mathbb{E}\left[\max_{\Delta\mathbf{x}}\mathcal{R}(w, w') \text{ s.t. } \mu(\Delta\mathbf{x}) < \delta\right] \text{ for all } z$$

Consequently,

$$\left(\max_{\Delta\mathbf{x}}\mathcal{R}(w, w') \text{ s.t. } \mu(\Delta\mathbf{x}) < \delta\right) = \begin{cases} 1 \text{ if } \max_{\mu(\Delta\mathbf{x})<\delta} f(\mathbf{x} + \Delta\mathbf{x}) \geq 0.5 \text{ and } f(\mathbf{x}) < 0.5 \\ 0 \text{ otherwise} \end{cases}$$

Therefore, we can write expectation as probability:

$$P\left(\max_{\mu(\Delta\mathbf{x})\leq\delta} f(\mathbf{x} + \Delta\mathbf{x}) \geq 0.5, f(\mathbf{x}) < 0.5 \mid \mathrm{z} = z\right) = P\left(\max_{\mu(\Delta\mathbf{x})\leq\delta} f(\mathbf{x} + \Delta\mathbf{x}) \geq 0.5, f(\mathbf{x}) < 0.5\right),$$

which is in Table 1.

### B.3 VULNERABILITY OF ER TO OUTLIERS

As we mentioned in Table 1, ER is vulnerable to outliers. Here we provide a simple analysis. Suppose an outlier, having feature-attribute pair $(\mathbf{x}, \mathrm{z})$, is added to the dataset with $n$ samples. Let the outlier is misqualified $f(\mathbf{x}) < 0.5$, and requires effort $\mu(\Delta\mathbf{x}) = M$ to achieve $f(\mathbf{x} + \Delta\mathbf{x}) \geq 0.5$. In such case, the EI disparity increases at most $\frac{1}{n}$ since EI measures the *portion* of samples with improved outcomes after making efforts. On the other hand, the ER disparity increases by $\frac{M}{n}$, since ER measures the minimum required efforts *averaged* over all samples. Note that a single outlier with a large $M$ can significantly increase the ER disparity, which does not happen for EI. Thus, one can observe that EI is much more robust to outliers, compared with ER. We conduct a numerical experiment in Sec. C.4.1 to further highlight the robustness of EI to outliers.

### B.4 SOLVING THE INNER MAXIMIZATION PROBLEM FOR EI

As explained in Sec.3, finding a EI classifier can be formulated as a minimax problem (1), where solving the inner maximization problem $\max_{\|\Delta \mathbf{x_I}\| \leq \delta} f(\mathbf{x} + \Delta \mathbf{x})$ is required to compute $U_\delta$ in the regularization term, and the outer problem is the regularized-loss minimization finding the optimal model parameter $\boldsymbol{w}$ for the classifier $f = f_{\boldsymbol{w}}$. In this section, we provide two ways of solving the inner maximization problem. In particular, in Sec. B.4.1 we give the explicit expression of the optimizer $\Delta \mathbf{x_I}$ when generalized linear model is considered. In Sec. B.4.2, we solve the problem under a more general setting via adversarial training.

#### B.4.1 CLOSED-FORM SOLUTION FOR GENERALIZED LINEAR MODEL

Consider a Generalized Linear Model (GLM) written as $f(\mathbf{x}) = g^{-1}(\boldsymbol{w}^\top \mathbf{x})$, where $g : [0, 1] \to \mathbb{R}$ is a strictly increasing link function, and $\boldsymbol{w}$ is the model parameter. Denote the weights corresponding to $\mathbf{x_I}$ as $\boldsymbol{w_I}$. Then, the inner maximization problem can be written as:

$$
\begin{aligned}
\max_{\|\Delta \mathbf{x_I}\| \leq \delta} f(\mathbf{x} + \Delta \mathbf{x}) &= \max_{\|\Delta \mathbf{x_I}\| \leq \delta} g^{-1}(\boldsymbol{w}^\top (\mathbf{x} + \Delta \mathbf{x})) \\
&= \max_{\|\Delta \mathbf{x_I}\| \leq \delta} g^{-1}(\boldsymbol{w}^\top \mathbf{x} + \boldsymbol{w}_I^\top \Delta \mathbf{x_I}) \quad (\because \Delta \mathbf{x} = (\Delta \mathbf{x}_I, 0, 0)) \\
&= g^{-1}\left( \boldsymbol{w}^\top \mathbf{x} + \max_{\|\Delta \mathbf{x_I}\| \leq \delta} \boldsymbol{w}_I^\top \Delta \mathbf{x_I} \right) \quad (\because g \text{ is strictly increasing})
\end{aligned}
$$

When $\|\cdot\| = \|\cdot\|_\infty$, the maximum is achieved by letting $\Delta \mathbf{x_I} = \delta \operatorname{sign}(\boldsymbol{w_I}), \boldsymbol{w}^\top \Delta \mathbf{x_I} = \delta \|\boldsymbol{w_I}\|_1$. When $\|\cdot\| = \|\cdot\|_2$, the maximum can be achieved by letting $\Delta \mathbf{x_I} = \delta \boldsymbol{w_I} / \|\boldsymbol{w_I}\|_2, \boldsymbol{w}^\top \Delta \mathbf{x_I} = \delta \|\boldsymbol{w_I}\|_2$.

#### B.4.2 ADVERSARIAL TRAINING BASED APPROACH FOR GENERAL SETUP

Here we discuss how we solve the inner maximization problem under a more general setting. Following popular adversarial training methods, we apply projected gradient descent (PGD) for multiple times to update $\Delta \mathbf{x_I}$, *i.e.*, set

$$
\Delta \mathbf{x_I} = \mathcal{P}(\Delta \mathbf{x_I} + \gamma \nabla_{\Delta \mathbf{x_I}} f(\mathbf{x} + \Delta \mathbf{x})), \tag{12}
$$

where $\gamma > 0$ is the step size, and $\mathcal{P}$ is the projection onto the constrained space $\|\Delta \mathbf{x_I}\| \leq \delta$. For instance, $\mathcal{P}$ is equivalent to the clipping process when we use $\ell_\infty$ norm. Denote the maximizer of the inner maximization problem as $\Delta \mathbf{x}^\star = (\Delta \mathbf{x_I}^\star, \mathbf{0}, \mathbf{0})$. Then, from Danskin's theorem Danskin (1967), we have $\nabla_{\boldsymbol{w}} \max_{\|\Delta \mathbf{x_I}\| \leq \delta} f_{\boldsymbol{w}}(\mathbf{x} + \Delta \mathbf{x}) = \nabla_{\boldsymbol{w}} f_{\boldsymbol{w}}(\mathbf{x} + \Delta \mathbf{x}^\star)$. We can use this derivative to update $\boldsymbol{w}$ in the outer loss minimization problem. The pseudocode of this adversarial training based method is shown in Algorithm 1.

---

**Algorithm 1** Pseudocode for achieving EI

**Input** :Dataset $\mathcal{D}$
**Output** :Model parameter $\boldsymbol{w}$ for the classifier $f$.
Initialize $\boldsymbol{w}$;
  **for** *each iteration* **do**
    **for** *each* $(\mathbf{x}_i, \mathbf{y}_i) \in \mathcal{D}$ **do**
      Initialize $\Delta \mathbf{x}_i^\star$;
      **for** *each PGD iteration* **do**
        Update $\Delta \mathbf{x}_i^\star$ according to (12);
    Update $\boldsymbol{w}$ according to the regularized loss function defined in (1);

---

### B.5 DERIVATION OF OPTIMAL EI CLASSIFIER FOR SYNTHETIC DATASET

This section shows how we obtain the optimal EI classifier (that minimizes the cost function in (1)) for a synthetic dataset having two features $\mathbf{x} = [\mathbf{x}_1, \mathbf{x}_2]$ sampled from a Gaussian distribution $\mathcal{N}(\boldsymbol{\mu}_{\mathbf{z},\mathbf{y}}, \boldsymbol{\Sigma}_{\mathbf{z},\mathbf{y}})$ where the mean $\boldsymbol{\mu}_{\mathbf{z},\mathbf{y}}$ and the standard deviation $\boldsymbol{\Sigma}_{\mathbf{z},\mathbf{y}}$ depends on the label $\mathbf{y} \in \{0, 1\}$ and the group attribute $\mathbf{z} \in \{0, 1\}$. Note that the performance curve of the optimal EI classifier obtained in this section is provided in the yellow line in Fig. 3.

The optimal EI classifier is obtained in the following steps: (i) define mathematical notations used for analysis (Sec. B.5.1), (ii) compute the error probability (Sec. B.5.2), (iii) compute EI disparity (Sec. B.5.3), and (iv) solve the EI-regularized optimization problem and find the optimal EI classifier (Sec. B.5.4).

### B.5.1 NOTATIONS

We consider finding a z-aware linear classifier which predicts the label $y$ from two features $x_1, x_2$ and one sensitive attribute $z$. In other words, given $\mathbf{x}$ and $z$, the output of a model is represented as $f(\mathbf{x}) = w_1 x_1 + w_2 x_2 + w_3 z + b$ [4] where $[w_1, w_2, w_3]$ is the weight vector, $b$ is the bias.

For group $z = 0$,

$$\hat{y} = \begin{cases} 1 & \text{if } w_1 x_1 + w_2 x_2 > -b \\ 0 & \text{else} \end{cases}$$

For group $z = 1$,

$$\hat{y} = \begin{cases} 1 & \text{if } w_1 x_1 + w_2 x_2 > -w_3 - b \\ 0 & \text{else} \end{cases}$$

Without the loss of generality, let $\sqrt{w_1^2 + w_2^2} = 1$, and parameterize them as $w_1 = \sin\theta, w_2 = \cos\theta$. Then, for group $z = 0$,

$$\hat{y} = \begin{cases} 1 & \text{if } (\sin\theta)x_1 + (\cos\theta)x_2 > b_0 \\ 0 & \text{else} \end{cases}$$

and for group $z = 1$,

$$\hat{y} = \begin{cases} 1 & \text{if } (\sin\theta)x_1 + (\cos\theta)x_2 > b_1 \\ 0 & \text{else} \end{cases}$$

where $b_0 = -b$ and $b_1 = -w_3 - b$. Since the linear combination of multivariate Gaussian is a univariate Gaussian, we have

$$\boldsymbol{w_\theta}^\top \boldsymbol{x} \sim \mathcal{N}(\boldsymbol{w_\theta}^\top \boldsymbol{\mu}_{z,y}, \boldsymbol{w_\theta}^\top \boldsymbol{\Sigma}_{z,y} \boldsymbol{w_\theta}) \tag{13}$$

where $\boldsymbol{w_\theta} = [\sin\theta, \cos\theta]$. The decision rules can be written in terms of the $\boldsymbol{w_\theta}$. For group $z = 0$,

$$\hat{y} = \begin{cases} 1 & \text{if } \boldsymbol{w_\theta}^\top \boldsymbol{x} > b_0 \\ 0 & \text{else} \end{cases}$$

For group $z = 1$,

$$\hat{y} = \begin{cases} 1 & \text{if } \boldsymbol{w_\theta}^\top \boldsymbol{x} > b_1 \\ 0 & \text{else} \end{cases}$$

Now, the question is, what is the optimal parameters $\theta, b_0, b_1$ that solve the optimization problem in (1). In order to answer this question, we need to understand how the equal improvability condition is represented in terms of the model parameters. Suppose we use 0-1 loss function $l$, and use $l_\infty$ norm $\mu(\mathbf{x}) = \|\mathbf{x}\|_\infty$. From the result in Sec. B.4, given the effort budget $\delta$, the maximum score improvement $(\max_{\mu(\Delta \mathbf{x}_I) \le \delta} f(\mathbf{x} + \Delta \mathbf{x}) - f(\mathbf{x})) = \delta \|\boldsymbol{w}_I\|_1 = \delta(|\sin\theta| + |\cos\theta|)$ where $\boldsymbol{w}_I$ is the weights for improvable features $x_1, x_2$. Thus, if we denote the $\hat{y}^{\max}$ as the estimated label after the improvement, we have

$$\hat{y}^{\max} = \begin{cases} 1 & \text{if } (\sin\theta)x_1 + (\cos\theta)x_2 > b_0 - \delta(|\sin\theta| + |\cos\theta|) = b_0' \\ 0 & \text{else} \end{cases}$$

for group $z = 0$ and

$$\hat{y}^{\max} = \begin{cases} 1 & \text{if } (\sin\theta)x_1 + (\cos\theta)x_2 > b_1 - \delta(|\sin\theta| + |\cos\theta|) = b_1' \\ 0 & \text{else} \end{cases}$$

for group $z = 1$.

---

[4] In this case, the decision boundary is $\{\mathbf{x} : f(\mathbf{x}) = 0\}$ instead of $\{\mathbf{x} : f(\mathbf{x}) = 0.5\}$ used in the main paper.

### B.5.2 COMPUTE ERROR PROBABILITY

The error probability can be written as,

$$\Pr(\hat{y} \neq y) = \sum_{i=0}^{1} \Pr(z = i) \Pr(\hat{y} \neq y | z = i)$$

We can derive the term $\Pr(\hat{y} \neq y | z = 0)$ as below:

$$\Pr(\hat{y} \neq y | z = 0) = \Pr(y = 0 | z = 0) \Pr(\hat{y} = 1 | y = 0, z = 0) \\ + \Pr(y = 1 | z = 0) \Pr(\hat{y} = 0 | y = 1, z = 0)$$

We can look each term $\Pr(\hat{y} = 1 | y = 0, z = 0), \Pr(\hat{y} = 0 | y = 1, z = 0)$ and write those terms in terms of Q-functions because,

$$\Pr(\hat{y} = 1 | y = 0, z = 0) = \Pr(w_\theta^\top x > b_0 | y = 0, z = 0)$$
$$\Pr(\hat{y} = 0 | y = 1, z = 0) = \Pr(w_\theta^\top x < b_0 | y = 1, z = 0)$$

From (13), we have

$$\Pr(\hat{y} = 1 | y = 0, z = 0) = \Pr(w_\theta^\top x > b_0 | y = 0, z = 0) = Q\left(\frac{b_0 - w_\theta^\top \mu_{0,0}}{\sqrt{w_\theta^\top \Sigma_{0,0} w_\theta}}\right)$$

$$\Pr(\hat{y} = 0 | y = 1, z = 0) = \Pr(w_\theta^\top x < b_0 | y = 1, z = 0) = Q\left(\frac{w_\theta^\top \mu_{1,0} - b_0}{\sqrt{w_\theta^\top \Sigma_{1,0} w_\theta}}\right)$$

One can derive the error rates for group $z = 1$ similarly. So, the total error rate can be written as

$$\Pr(\hat{y} \neq y) = \Pr(z = 0) \left[ \Pr(y = 0 | z = 0) Q\left(\frac{b_0 - w_\theta^\top \mu_{0,0}}{\sqrt{w_\theta^\top \Sigma_{0,0} w_\theta}}\right) \right.$$

$$\left. + \Pr(y = 1 | z = 0) Q\left(\frac{w_\theta^\top \mu_{1,0} - b_0}{\sqrt{w_\theta^\top \Sigma_{1,0} w_\theta}}\right) \right]$$

$$+ \Pr(z = 1) \left[ \Pr(y = 0 | z = 1) Q\left(\frac{b_1 - w_\theta^\top \mu_{1,0}}{\sqrt{w_\theta^\top \Sigma_{1,0} w_\theta}}\right) \right.$$

$$\left. + \Pr(y = 1 | z = 1) Q\left(\frac{w_\theta^\top \mu_{1,1} - b_1}{\sqrt{w_\theta^\top \Sigma_{1,1} w_\theta}}\right) \right]$$

We have three parameters to optimize the error rate $\theta, b_0, b_1$. All the other terms are known.

### B.5.3 COMPUTE EI DISPARITY

To compute EI disparity, we start with computing

$$\Pr(\hat{y}^{\max} = 1 | \hat{y} = 0, z = 0) = \frac{\Pr(\hat{y}^{\max} = 1, \hat{y} = 0 | z = 0)}{\Pr(\hat{y} = 0 | z = 0)} \tag{14}$$

The denominator of (14) can be expanded as

$$\Pr(\hat{y} = 0 | z = 0) = \Pr(y = 0 | z = 0) \Pr(\hat{y} = 0 | y = 0, z = 0) \\ + \Pr(y = 1 | z = 0) \Pr(\hat{y} = 0 | y = 1, z = 0).$$

We can look each term $\Pr(\hat{y} = 0 | y = 0, z = 0), \Pr(\hat{y} = 0 | y = 1, z = 0)$ and write those terms in terms of Q-function because,

$$\Pr(\hat{y} = 0 | y = 0, z = 0) = \Pr(w_\theta^\top x < b_0 | y = 0, z = 0)$$

$$\Pr(\hat{\mathbf{y}} = 0 | \mathbf{y} = 1, \mathbf{z} = 0) = \Pr(\boldsymbol{w}_\theta^\top \mathbf{x} < b_0 | \mathbf{y} = 1, \mathbf{z} = 0)$$

From (13), we have

$$\Pr(\hat{\mathbf{y}} = 0 | \mathbf{y} = 0, \mathbf{z} = 0) = \Pr(\boldsymbol{w}_\theta^\top \mathbf{x} < b_0 | \mathbf{y} = 0, \mathbf{z} = 0) = Q\left(\frac{\boldsymbol{w}_\theta^\top \boldsymbol{\mu}_{0,0} - b_0}{\sqrt{\boldsymbol{w}_\theta^\top \boldsymbol{\Sigma}_{0,0} \boldsymbol{w}_\theta}}\right)$$

$$\Pr(\hat{\mathbf{y}} = 0 | \mathbf{y} = 1, \mathbf{z} = 0) = \Pr(\boldsymbol{w}_\theta^\top \mathbf{x} > b_0 | \mathbf{y} = 1, \mathbf{z} = 0) = Q\left(\frac{\boldsymbol{w}_\theta^\top \boldsymbol{\mu}_{1,0} - b_0}{\sqrt{\boldsymbol{w}_\theta^\top \boldsymbol{\Sigma}_{1,0} \boldsymbol{w}_\theta}}\right)$$

Then,

$$\Pr(\hat{\mathbf{y}} = 0 | \mathbf{z} = 0) = \Pr(\mathbf{y} = 0 | \mathbf{z} = 0) Q\left(\frac{\boldsymbol{w}_\theta^\top \boldsymbol{\mu}_{0,0} - b_0}{\sqrt{\boldsymbol{w}_\theta^\top \boldsymbol{\Sigma}_{0,0} \boldsymbol{w}_\theta}}\right)$$
$$+ \Pr(\mathbf{y} = 1 | \mathbf{z} = 0) Q\left(\frac{\boldsymbol{w}_\theta^\top \boldsymbol{\mu}_{1,0} - b_0}{\sqrt{\boldsymbol{w}_\theta^\top \boldsymbol{\Sigma}_{1,0} \boldsymbol{w}_\theta}}\right)$$

The numerator of (14) can be expanded as

$$\Pr(\hat{\mathbf{y}}^{\max} = 1, \hat{\mathbf{y}} = 0 | \mathbf{z} = 0) = \Pr(\mathbf{y} = 0 | \mathbf{z} = 0) \Pr(\hat{\mathbf{y}}^{\max} = 1, \hat{\mathbf{y}} = 0 | \mathbf{y} = 0, \mathbf{z} = 0)$$
$$+ \Pr(\mathbf{y} = 1 | \mathbf{z} = 0) \Pr(\hat{\mathbf{y}}^{\max} = 1, \hat{\mathbf{y}} = 0 | \mathbf{y} = 1, \mathbf{z} = 0).$$

We can look each term $\Pr(\hat{\mathbf{y}}^{\max} = 1, \hat{\mathbf{y}} = 0 | \mathbf{y} = 0, \mathbf{z} = 0), \Pr(\hat{\mathbf{y}}^{\max} = 1, \hat{\mathbf{y}} = 0 | \mathbf{y} = 1, \mathbf{z} = 0)$ and write those terms in terms of Q-function because,

$$\Pr(\hat{\mathbf{y}}^{\max} = 1, \hat{\mathbf{y}} = 0 | \mathbf{y} = 0, \mathbf{z} = 0) = \Pr(b_0' < \boldsymbol{w}_\theta^\top \mathbf{x} < b_0 | \mathbf{y} = 0, \mathbf{z} = 0)$$

$$\Pr(\hat{\mathbf{y}}^{\max} = 1, \hat{\mathbf{y}} = 0 | \mathbf{y} = 1, \mathbf{z} = 0) = \Pr(b_0' < \boldsymbol{w}_\theta^\top \mathbf{x} < b_0 | \mathbf{y} = 1, \mathbf{z} = 0)$$

From (13), we have

$$\Pr(\hat{\mathbf{y}}^{\max} = 1, \hat{\mathbf{y}} = 0 | \mathbf{y} = 0, \mathbf{z} = 0) = \Pr(b_0' < \boldsymbol{w}_\theta^\top \mathbf{x} < b_0 | \mathbf{y} = 0, \mathbf{z} = 0) =$$
$$Q\left(\frac{\boldsymbol{w}_\theta^\top \boldsymbol{\mu}_{0,0} - b_0}{\sqrt{\boldsymbol{w}_\theta^\top \boldsymbol{\Sigma}_{0,0} \boldsymbol{w}_\theta}}\right) - Q\left(\frac{\boldsymbol{w}_\theta^\top \boldsymbol{\mu}_{0,0} - b_0'}{\sqrt{\boldsymbol{w}_\theta^\top \boldsymbol{\Sigma}_{0,0} \boldsymbol{w}_\theta}}\right)$$

$$\Pr(\hat{\mathbf{y}}^{\max} = 1, \hat{\mathbf{y}} = 0 | \mathbf{y} = 1, \mathbf{z} = 0) = \Pr(b_0' < \boldsymbol{w}_\theta^\top \mathbf{x} < b_0 | \mathbf{y} = 1, \mathbf{z} = 0) =$$
$$Q\left(\frac{\boldsymbol{w}_\theta^\top \boldsymbol{\mu}_{1,0} - b_0}{\sqrt{\boldsymbol{w}_\theta^\top \boldsymbol{\Sigma}_{1,0} \boldsymbol{w}_\theta}}\right) - Q\left(\frac{\boldsymbol{w}_\theta^\top \boldsymbol{\mu}_{1,0} - b_0'}{\sqrt{\boldsymbol{w}_\theta^\top \boldsymbol{\Sigma}_{1,0} \boldsymbol{w}_\theta}}\right)$$

Then,

$$\Pr(\hat{\mathbf{y}}^{\max} = 1, \hat{\mathbf{y}} = 0 | \mathbf{z} = 0) = \Pr(\mathbf{y} = 0 | \mathbf{z} = 0) \left[ Q\left(\frac{\boldsymbol{w}_\theta^\top \boldsymbol{\mu}_{0,0} - b_0}{\sqrt{\boldsymbol{w}_\theta^\top \boldsymbol{\Sigma}_{0,0} \boldsymbol{w}_\theta}}\right) - Q\left(\frac{\boldsymbol{w}_\theta^\top \boldsymbol{\mu}_{0,0} - b_0'}{\sqrt{\boldsymbol{w}_\theta^\top \boldsymbol{\Sigma}_{0,0} \boldsymbol{w}_\theta}}\right) \right]$$
$$+ \Pr(\mathbf{y} = 1 | \mathbf{z} = 0) \left[ Q\left(\frac{\boldsymbol{w}_\theta^\top \boldsymbol{\mu}_{1,0} - b_0}{\sqrt{\boldsymbol{w}_\theta^\top \boldsymbol{\Sigma}_{1,0} \boldsymbol{w}_\theta}}\right) - Q\left(\frac{\boldsymbol{w}_\theta^\top \boldsymbol{\mu}_{1,0} - b_0'}{\sqrt{\boldsymbol{w}_\theta^\top \boldsymbol{\Sigma}_{1,0} \boldsymbol{w}_\theta}}\right) \right]$$

It can be derived for group $\mathbf{z} = 1$ similarly. So, we derived EI disparity in terms of $\theta, b_0, b_1$.

### B.5.4 SOLVE THE OPTIMIZATION PROBLEM

In the previous two sections, we derived the error rate and EI disparity in terms of Q-functions containing parameters $\theta, b_0, b_1$. Therefore, we can construct an EI-constrained optimization problem (which is essentially same as (1)):

$$\min_{\theta, b_0, b_1} \quad \Pr(\hat{y} \neq y)$$
$$\text{s.t.} \quad \max_{i \in \{0,1\}} |\Pr(\hat{y}^{\max} = 1|\hat{y} = 0, z = i) - \Pr(\hat{y}^{\max} = 1|\hat{y} = 0)| < c$$

where $c$ is a hyperparameter we can choose to balance error rate and EI disparity.

After writing error rate and EI disparity in terms of Q-functions (using the derivations in Sec. B.5.2 and Sec. B.5.3), we numerically solve the constrained optimization problems above with a popular python module `scipy.optimize`. To get the experimental results in Fig. 3, we numerically solved the above problem for 20 different $c$ values, where the maximum $c$ is picked as the EI disparity of the unconstrained optimization problem.

## C SUPPLEMENTARY MATERIALS ON EXPERIMENTS

In this section, we provide the details of the experiment setup and additional experimental results.

### C.1 SYNTHETIC DATASET GENERATION

We define $y, z$ as $z \sim \text{Bern}(0.4), (y \mid z = 0) \sim \text{Bern}(0.3)$, and $(y \mid z = 1) \sim \text{Bern}(0.5)$. The feature $\mathbf{x}$ follows the conditional distribution $(\mathbf{x} \mid y = y, z = z) \sim \mathcal{N}(\boldsymbol{\mu}_{y,z}, \boldsymbol{\Sigma}_{y,z})$ where the mean of each cluster is

$$\boldsymbol{\mu}_{0,0} = [-0.1, -0.2], \boldsymbol{\mu}_{0,1} = [-0.2, -0.3], \boldsymbol{\mu}_{1,0} = [0.1, 0.4], \boldsymbol{\mu}_{1,1} = [0.4, 0.3]$$

and the covariance matrix of each cluster is

$$\boldsymbol{\Sigma}_{0,0} = \begin{bmatrix} 0.4 & 0.0 \\ 0.0 & 0.4 \end{bmatrix}, \boldsymbol{\Sigma}_{1,0} = \boldsymbol{\Sigma}_{0,1} = \begin{bmatrix} 0.2 & 0.0 \\ 0.0 & 0.2 \end{bmatrix}, \boldsymbol{\Sigma}_{1,1} = \begin{bmatrix} 0.1 & 0.0 \\ 0.0 & 0.1 \end{bmatrix}.$$

### C.2 HYPERPARAMETER SELECTION

The selected hyperparameter for each experiment is provided in our anonymous Github. In all our experiments, we perform cross-validation to select the learning rate from $\{0.0001, 0.001, 0.01, 0.1\}$. In addition, for each penalty term we did two-step cross-validation to choose $\lambda$. In the first step, we used a set $\lambda \in \{0, 0.2, 0.4, 0.6, 0.8, 0.9\}$. In the second step, we generate a second set around the best $\lambda^\star$ found in the first step, i.e., the second set is $\{\max\{\lambda^\star + \varepsilon, 0\} : \varepsilon \in \{-0.1, -0.05, 0, 0.05, 0.1\}\}$. For example, if $\lambda^\star = 0.4$ is the best at the first step, then at the second step we use the set $\lambda \in \{0.3, 0.35, 0.4, 0.45, 0.5\}$.

### C.3 ADDITIONAL EXPERIMENTAL RESULTS ON ALGORITHM EVALUATION

Table 3 shows the performance of ERM baseline and our three approaches (covariance-based, KDE-based, loss-based) introduced in Sec. 3, for a multi-layer perceptron (MLP) network having one hidden layer with four neurons. Similar to the result in Table 2 for the logistic regression model, our methods can reduce the EI disparity without losing the classification accuracy (i.e., increasing the error rate) much.

Meanwhile, according to a previous work (Cherepanova et al., 2021), large deep learning models are observed to overfit to fairness constraints during training and therefore produce unfair predictions on the test data. To confirm whether our method is also having such limitations, we investigate the performance of our algorithms on over-parameterized models. Specifically, we conduct experiments on a five-layer ReLU network with 200 hidden neurons per layer, which is over-parameterized for German dataset. Table 4 reports the performance of EI-constrained classifiers on such over-parameterized setting. One can confirm that our methods (covariance-based, KDE-based, loss-based) perform well in both training and test dataset, and we do not observe the overfitting problem.

Table 3: **Comparison of error rate and EI disparities of ERM baseline and proposed methods on the synthetic, German Statlog Credit and ACSIncome-CA datasets on Multi-Layer Perceptron (MLP).** For each dataset, we boldfaced the lowest EI disparity value. Compared with ERM, all three methods proposed in this paper enjoys much lower EI disparity without losing the accuracy much. All reported numbers are evaluated on the test set.

| | | | | METHODS | |
| DATASET | METRIC | ERM | COVARIANCE-BASED | KDE-BASED | LOSS-BASED |
| --- | --- | --- | --- | --- | --- |
| SYNTHETIC | ERROR RATE($\downarrow$) | $.205 \pm .003$ | $.242 \pm .006$ | $.227 \pm .008$ | $.229 \pm .012$ |
| | EI DISP.($\downarrow$) | $.141 \pm .036$ | $\mathbf{.004 \pm .002}$ | $.011 \pm .006$ | $.018 \pm .009$ |
| GERMAN STAT. | ERROR RATE($\downarrow$) | $.221 \pm .010$ | $.299 \pm .012$ | $.232 \pm .018$ | $.238 \pm .035$ |
| | EI DISP.($\downarrow$) | $.059 \pm .045$ | $\mathbf{.013 \pm .025}$ | $.041 \pm .025$ | $\mathbf{.013 \pm .019}$ |
| ACSINCOME-CA | ERROR RATE($\downarrow$) | $.181 \pm .002$ | $.202 \pm .002$ | $.182 \pm .002$ | $.185 \pm .001$ |
| | EI DISP.($\downarrow$) | $.042 \pm .002$ | $.010 \pm .006$ | $.010 \pm .002$ | $\mathbf{.006 \pm .003}$ |

Table 4: **Error rate and EI disparities for ERM baseline and proposed methods, for an over-parameterized neural network on German Statlog Credit dataset.** Performances on train/test dataset are presented. Note that we don't observe the overfitting issue in the over-parameterized setting.

| | | | | METHODS | |
| DATASET | METRIC | ERM | COVARIANCE-BASED | KDE-BASED | LOSS-BASED |
| --- | --- | --- | --- | --- | --- |
| | TRAIN ERR.($\downarrow$) | $.117 \pm .004$ | $.133 \pm .003$ | $.125 \pm .008$ | $.132 \pm .011$ |
| GERMAN | TEST ERR.($\downarrow$) | $.117 \pm .010$ | $.118 \pm .010$ | $.121 \pm .010$ | $.130 \pm .009$ |
| STAT. | TRAIN EI DISP.($\downarrow$) | $.022 \pm .017$ | $.018 \pm .011$ | $.018 \pm .009$ | $.015 \pm .013$ |
| | TEST EI DISP.($\downarrow$) | $.060 \pm .032$ | $.049 \pm .024$ | $.057 \pm .028$ | $.047 \pm .025$ |

In addition, in Table 5, we include ER and BE as baselines for the synthetic dataset experiment provided in Table 2. We leverage the algorithm suggested by Gupta et al. (2019) for mitigating ER disparity. We extend the loss-based approach to reduce BE disparity, by redefining the BE loss of group $z$ as

$$\tilde{L}_z^{\text{BE}} \triangleq \frac{1}{\text{number of samples in group } z} \sum_{i \in I_{-,z}} \ell(1, \max_{\|\Delta \mathbf{x}_{\text{I}i}\| \leq \delta} f(\mathbf{x}_i + \Delta \mathbf{x}_i)),$$

where

$$\tilde{L}_z^{\text{EI}} \triangleq \frac{1}{\text{number of rejected samples in group } z} \sum_{i \in I_{-,z}} \ell(1, \max_{\|\Delta \mathbf{x}_{\text{I}i}\| \leq \delta} f(\mathbf{x}_i + \Delta \mathbf{x}_i)),$$

and $I_{-,z}$ is the set of rejected samples in group $z$ for $z \in [Z]$.

Table 5 shows that the minimum EI disparity is achieved by our methods. Hence, if the EI fairness needs to be achieved, then it cannot be replaced with the existing other fairness notions for some datasets.

Table 5: **Comparison of error rate and EI disparities of ERM, ER, and BE baseline and proposed methods on the synthetic dataset.** We boldfaced the lowest EI disparity value. The three EI-regularized approaches achieve the lowest EI disparity while maintaining low error rates. All reported numbers are evaluated on the test set.

| | | | | | METHODS | | |
| DATASET | METRIC | ERM | ER (GUPTA ET AL. (2019)) | BE (LOSS-BASED) | COVARIANCE-BASED | KDE-BASED | LOSS-BASED |
| --- | --- | --- | --- | --- | --- | --- | --- |
| SYNTHETIC | ERROR RATE($\downarrow$) | $.221 \pm .001$ | $.235 \pm .009$ | $.252 \pm .006$ | $.253 \pm .003$ | $.250 \pm .001$ | $.246 \pm .001$ |
| | EI DISP.($\downarrow$) | $.117 \pm .007$ | $.036 \pm .018$ | $.006 \pm .004$ | $.003 \pm .001$ | $.005 \pm .003$ | $\mathbf{.002 \pm .001}$ |

## C.4 EVALUATING ROBUSTNESS OF EI

In this section, we highlight the advantages of EI over BE and ER in terms of robustness to outliers and imbalanced negative prediction rates. In doing so, we consider certain data distributions and follow the method discussed in Sec. B.5 for solving the classifiers.

### C.4.1 EI VERSUS ER: ROBUSTNESS TO OUTLIERS

As we discussed in Sec. B.3, ER is vulnerable to outliers. In this experiment, we systematically study the robustness of EI and ER to outliers.

**Data distributions** **(Clean)** Let sensitive attribute $z \sim \mathrm{Bern}(0.5)$ and label $y \sim \mathrm{Bern}(0.5)$ be independent of sensitive attribute z. Given the sensitive attribute z and label y, feature x follows the conditional distribution $\mathbf{x} \mid \mathbf{y} = y, \mathbf{z} = z \sim \mathcal{N}(\boldsymbol{\mu}_{y,z}, \boldsymbol{\Sigma}_{y,z})$, where the mean and covariance of the four Gaussian clusters are

$$\boldsymbol{\mu}_{0,0} = [1, -6], \boldsymbol{\mu}_{0,1} = [-1, -2], \boldsymbol{\mu}_{1,0} = [2, 1.5], \boldsymbol{\mu}_{1,1} = [1, 2.5],$$

and

$$\boldsymbol{\Sigma}_{0,0} = \boldsymbol{\Sigma}_{1,0} = \boldsymbol{\Sigma}_{0,1} = \boldsymbol{\Sigma}_{1,1} = \begin{bmatrix} 0.25 & 0.0 \\ 0.0 & 0.25 \end{bmatrix}.$$

**(Contaminated)** We contaminate the distribution by introducing additional 5% outliers to group $z = 0$. The outliers follow Gaussian distribution with mean and covariance matrix

$$\boldsymbol{\mu}_{\mathrm{outlier},0} = [-1, -20], \boldsymbol{\Sigma}_{\mathrm{outlier},0} = \begin{bmatrix} 0.05 & 0.0 \\ 0.0 & 0.05 \end{bmatrix}.$$

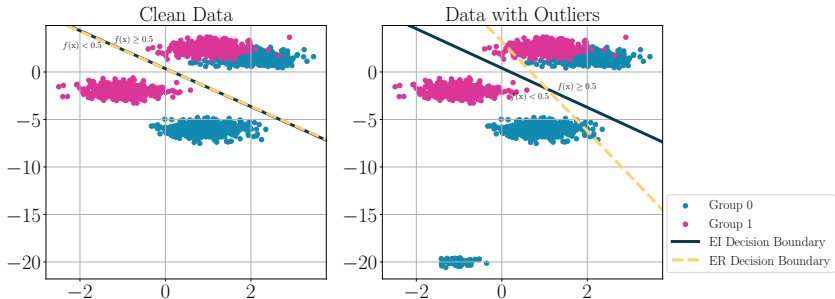

Figure 6: **Visualizations of the EI and ER decision boundaries without and with the presence of outliers.** We observe that the decision boundary of ER changes a lot in the presence of outliers, while the decision boundary of EI is not affected. This phenomenon implies the robustness of EI to outliers.

**Results** The decision boundaries of EI and ER for both the clean dataset and contaminated dataset are depicted in Fig. 6. These decision boundaries are the optimal linear decision boundaries based on the distributional information, we followed the same procedure as we mentioned in B.5. The $\delta$ for the EI classifier is picked as 1.5. We observe that the introduction of outliers makes the ER decision boundary rotate a lot, leading to a drop in classification accuracy and ER disparity *w.r.t.* the clean data distribution. Moreover, we note that the newly added outliers fail to destroy EI classifier, which implies the robustness of EI to outliers. The Table 6 also presents the accuracy, EI, and ER disparity of each classifier. These metrics are computed after excluding the outliers. It shows that the EI and ER disparities have higher values under the dataset with the outliers for the ER classifier.

Table 6: **Error rate and EI and ER disparities of linear EI and ER classifiers.**

| | | METHODS | |
|---|---|---|---|
| DATASET | METRIC | EI CLASSIFIER | ER CLASSIFIER |
| DATASET WITHOUT OUTLIERS | ERROR RATE($\downarrow$) | $.001 \pm .001$ | $.001 \pm .001$ |
| | EI DISP.($\downarrow$) | $.001 \pm .001$ | $.005 \pm .001$ |
| | ER DISP.($\downarrow$) | $.004 \pm .001$ | $.001 \pm .001$ |
| DATASET WITH OUTLIERS | ERROR RATE($\downarrow$) | $.001 \pm .001$ | $.020 \pm .001$ |
| | EI DISP.($\downarrow$) | $.017 \pm .001$ | $.348 \pm .001$ |
| | ER DISP.($\downarrow$) | $.020 \pm .001$ | $.291 \pm .001$ |

C.4.2  EI VERSUS BE: ROBUSTNESS TO IMBALANCED GROUP NEGATIVE RATE

In this experiment, we investigate the robustness of EI and BE to an imbalanced negative rate.

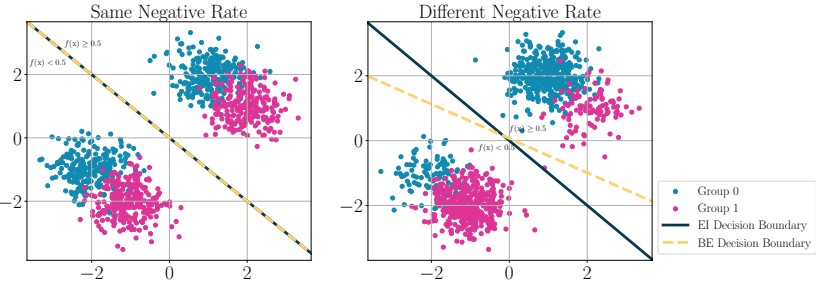

Figure 7: **Visualizations of the EI and BE decision boundaries given the data distribution with the same negative rates and different negative rates.** The decision boundary of BE rotates a lot when the negative rate of the dataset becomes different, implying the sensitivity of BE to imbalanced negative rates. In contrast, the consistency of EI decision boundaries showcases the robustness of EI *w.r.t.* imbalanced negative rates.

**Data Distributions**  **(Same Negative Rate)** We consider the distribution with balanced subgroups. In other words, let sensitive attribute $z \sim \mathrm{Bern}(0.5)$, and label $y \sim \mathrm{Bern}(0.5)$ be independent of sensitive attribute z. Given the sensitive attribute z and label y, feature, feature $\mathbf{x}$ follows the Gaussian distribution $(\mathbf{x} \mid \mathbf{y} = y, \mathbf{z} = z) \sim \mathcal{N}(\boldsymbol{\mu}_{y,z}, \boldsymbol{\Sigma}_{y,z})$ where the mean of each cluster is

$$\boldsymbol{\mu}_{0,0} = [-2, -1], \boldsymbol{\mu}_{0,1} = [-1, -2], \boldsymbol{\mu}_{1,0} = [1, 2], \boldsymbol{\mu}_{1,1} = [2, 1]$$

and the covariance matrix of each cluster is

$$\boldsymbol{\Sigma}_{0,0} = \boldsymbol{\Sigma}_{1,0} = \boldsymbol{\Sigma}_{0,1} = \boldsymbol{\Sigma}_{1,1} = \begin{bmatrix} 0.25 & 0.0 \\ 0.0 & 0.25 \end{bmatrix}.$$

**(Different Negative Rate)** We manipulate the distribution of label y for constructing data distribution with different negative rates. To be more specific, we let $y \mid z = 0 \sim \mathrm{Bern}(0.7)$, and $y \mid z = 1 \sim \mathrm{Bern}(0.3)$.

**Results**  Figure 7 shows the decision boundaries of EI and BE when (i) the dataset has the same negative rate, and (ii) the dataset has a different negative rate. These decision boundaries are the optimal linear decision boundaries based on the distributional information, we followed the same procedure as we mentioned in B.5. The $\delta$ for the EI and BE classifiers is picked as 1.5. The huge difference in BE decision boundaries under the two cases verifies our claim that BE cannot handle imbalanced negative prediction rates. In contrast, the decision boundaries of EI learned with two datasets with different group proportions are consistent.

C.5  DYNAMIC MODELING

Here we justify the dynamic model we used in our experiments in Sec. 4.2, for updating individual's features and for updating the classifier. Specifically, Sec. C.5.1 explains the choice of $\epsilon(x)$, the amount of effort a sample (having feature $x$), while Sec. C.5.2 explains the choice of $(\tau_t^{(0)}, \tau_t^{(1)})$ defining the classifier at round $t$.

C.5.1  UPDATING INDIVIDUAL'S FEATURES (CHOICE OF $\epsilon(x)$)

At first, we designed a model by assuming that each individual makes an effort, where the amount of the effort he/she makes is (1) proportional to the reward (the improvement on the outcome) it will get by making such effort, and (2) inversely proportional to the required amount of efforts to improve the

outcome, as shown in the below equation:

$$\text{Realized effort } \epsilon(x) \triangleq \frac{\text{improvement on the outcome}}{\text{required efforts for improving the outcome}^2}$$

$$= \frac{\mathbf{1}\{x < \tau_t^{(z)}\}}{(\tau_t^{(z)} - x)^2},$$

where $\tau_t^{(z)}$ is the accepting threshold for the individuals from group $z$ at round $t$, and $x$ is the feature. This equation can be interpreted as follows: an individual that is unqualified ($x < \tau_t^{(z)}$) is willing to make positive effort, but he/she is less motivated if the distance $\tau_t^{(z)} - x$ to the decision boundary is too large.

In order to upper bound the amount of reward, we added a small constant $\beta > 0$ in the numerator, thus having the final form:

$$\epsilon(x) = \frac{\mathbf{1}\{x < \tau_t^{(z)}\}}{(\tau_t^{(z)} - x + \beta)^2}.$$

### C.5.2 UPDATING THE CLASSIFIER

Note that at each round $t$, each individual's features are updated. Accordingly, we also update the classifier (having parameters $\tau_t^{(0)}$ and $\tau_t^{(1)}$) in the same round. We update the EI classifier based on the following constrained optimization problem:

$$\min (\text{EI Disparity}) \quad \text{s.t.} \quad \text{error rate} \leq \text{threshold},$$

which implies that we aim to find a classifier that guarantees good accuracy and small EI disparity. This optimization can be rewritten as

$$\min_{\tau_t^{(0)}, \tau_t^{(1)}} \overbrace{\left| \mathbb{P}(\underbrace{\mathrm{x}_t + \delta_t > \tau_t^{(0)}}_{\text{the outcome can be improved}} \mid \mathrm{z} = 0, \underbrace{\mathrm{x}_t < \tau_t^{(0)}}_{\text{rejected}}) - \mathbb{P}(\mathrm{x}_t + \delta_t > \tau_t^{(1)} \mid \mathrm{z} = 1, \mathrm{x}_t < \tau_t^{(1)}) \right|}^{\text{EI Disparity}}$$

$$\text{s.t. } \overbrace{\mathbb{P}(\hat{\mathrm{y}}_t \neq \mathrm{y}_t)}^{\text{error rate}} \leq c.$$

Given the data distribution at each round $t$, we numerically solve the constrained optimization problems using a popular python module `scipy.optimize`.

### C.6 OBTAINING BASELINE CLASSIFIERS FOR DYNAMIC SCENARIOS

Continued from Sec. 4.2, this section describes how we compute the classifiers that satisfy demographic parity (DP), bounded effort (BE) (Heidari et al., 2019) equal recourse (ER) (Gupta et al., 2019) and individual-level fairness causal recourse (ILFCR) (Von Kügelgen et al., 2022), respectively. Similar to EI classifier, we obtain the best DP classifier by considering the following constrained optimization problem:

$$\min |\mathbb{P}(\hat{\mathrm{y}}_t = 1 \mid \mathrm{z} = 0) - \mathbb{P}(\hat{\mathrm{y}}_t = 1 \mid \mathrm{z} = 1)| \quad \text{s.t. } \mathbb{P}(\hat{\mathrm{y}}_t \neq \mathrm{y}_t) \leq c.$$

The best BE classifier can be obtained by solving the following problem:

$$\min \left| \mathbb{P}(\tau_t^{(0)} - \delta_t < \mathrm{x}_t < \tau_t^{(0)} \mid \mathrm{z} = 0) - \mathbb{P}(\tau_t^{(1)} - \delta_t < \mathrm{x}_t < \tau_t^{(1)} \mid \mathrm{z} = 1) \right| \quad \text{s.t. } \mathbb{P}(\hat{\mathrm{y}}_t \neq \mathrm{y}_t) \leq c.$$

Similarly, the optimization problem for obtaining the best ER classifier is written as:

$$\min \left| \mathbb{E}\left[\tau_t^{(0)} - \mathrm{x}_t \mid \mathrm{x}_t < \tau_t^{(0)}, \mathrm{z} = 0\right] - \mathbb{E}\left[\tau_t^{(1)} - \mathrm{x}_t \mid \mathrm{x}_t < \tau_t^{(1)}, \mathrm{z} = 1\right] \right| \quad \text{s.t. } \mathbb{P}(\hat{\mathrm{y}}_t \neq \mathrm{y}_t) \leq c.$$

While the computation of ILFCR classifier is less straightforward, we first describe the setting considered in Von Kügelgen et al. (2022). This paper assumes that each observed variable $x_i$ is

determined by (i) its direct causes (causal parents) which include the sensitive attribute $z$ and other observed variables $x_j$, and (ii) an unobserved variable $u_i$. Since our dynamics experiment considers only one-dimensional variable $x$, it is determined by the sensitive attribute $z$ and latent variable $u$. Therefore, we write $x$ as a function of $z$ and $u$, i.e., $x(z, u)$.

Before we describe how we compute ILFCR in our experiments, we introduce the definition of causal recourse and ILFCR in more detail. Given a model and a sample with feature $x$, the causal recourse $C(x)$ of the sample is the minimum cost of changing the features, in order to alter the decision of the model. Let $\tau^{(z)} \in \mathbb{R}$ be the acceptance threshold of group $z \in \{0, 1\}$ and $\mu(\Delta x) = |\Delta x|$ be the cost of improving the feature by $\Delta x$. Let $x'$ denote the improved feature. In our one-dimensional dynamic setting, the causal recourse is defined as

$$C(x(z, u)) = \min_{x' \geq \tau^{(z)}} \mu(x' - x(z, u)) = \max(\tau^{(z)} - x(z, u), 0),$$

for $z = 0, 1$. ILFCR requires different groups have the same causal recourse for all realizations of the latent variable $u$:

$$\max_u |C(x(0, u)) - C(x(1, u))| = 0.$$

Now we describe how we compute ILFCR disparity. Recall that for dynamic experiments, we assumed the feature $x$ follows Gaussian distribution for each group $z \in \{0, 1\}$, i.e., $x \mid z \sim \mathcal{N}(\mu_z, \sigma_z^2)$. This can be represented as a notation used in Von Kügelgen et al. (2022): the latent variable is $u \sim \mathcal{N}(0, 1)$ and the feature is represented as $x(z, u) = \mu_z + \sigma_z u$. We measure the ILFCR disparity of a decision boundary pair $(\tau_0, \tau_1)$ by

$$\text{ILFCR Disparity}(\tau^{(0)}, \tau^{(1)}) = \max_u |C(x(0, u)) - C(x(1, u))|$$

$$= \max_u \left| \max\left(\tau^{(0)} - x(0, u), 0\right) - \max\left(\tau^{(1)} - x(1, u), 0\right) \right|$$

$$= \max_u \underbrace{\left| \max\left(\tau^{(0)} - \mu_0 - \sigma_0 u, 0\right) - \max\left(\tau^{(1)} - \mu_1 - \sigma_1 u, 0\right) \right|}_{=(\tau^{(0)} - \tau^{(1)} - \mu_0 + \mu_1 - (\sigma_0 - \sigma_1)u) \to \infty \text{ when } u \to -\infty}.$$

Note that the ILFCR disparity is infinity if we take the maximum over all $u$. Therefore, we instead ignore the tail distribution and focus on the samples with $x \in [\mu_z - 3\sigma_z, \mu_z + 3\sigma_z]$ for group $z = 0, 1$. In order to find a classifier with small ILFCR disparity, we numerically solve the following constrained optimization problem

$$\min_{\tau^{(0)}, \tau^{(1)}} \quad \text{ILFCR Disparity}(\tau^{(0)}, \tau^{(1)}) \quad \text{s.t.} \quad \text{error rate} \leq \alpha/2.$$

Given the data distribution at each round $t$, we numerically solve all the constrained optimization problems above using a popular python module `scipy.optimize`.

### C.7 PERFORMANCE COMPARISON BETWEEN FAIRNESS NOTIONS UNDER OTHER DYMANIC MODELS

Here we provide one example in which EI improves long-term fairness, while some other methods (EO, BE, and ER) harm long-term fairness.

In this example, we modified the effort function from $\epsilon(x) = \frac{1}{(\tau_t^{(z)} - x + \beta)^2} \mathbf{1}\{x < \tau_t^{(z)}\}$ to $\epsilon(x) = \log\left(\max(\frac{1}{(\tau_t^{(z)} - x + \beta)^2}, 1)\right) \mathbf{1}\{x < \tau_t^{(z)}\}$ so that the rejected individuals make less improvement than the setting in the original manuscript. Here $\tau_t^{(z)}$ is the acceptance threshold for group $z$ at iteration $t$, and $\beta > 0$ is a small constant to avoid zero denominators. We set $\beta = 0.2$ and the true acceptance rate $\alpha = 0.5$. Figure 8 reports how long-term unfairness changes when the initial distribution is $x \mid z = 0 \sim \mathcal{N}(0.0, 3.0^2)$ for group 0 and $x \mid z = 1 \sim \mathcal{N}(1.0, 1.0^2)$ for group 1.

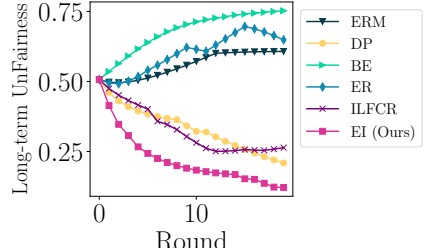

Figure 8: Long-term unfairness $d_{TV}(\mathcal{P}_t^{(0)}, \mathcal{P}_t^{(1)})$ at each round $t$ for various algorithms in the setup described in Sec. C.7.

# D    EXAMPLES: INVESTIGATING THE ONE-STEP IMPACT OF FAIRNESS NOTIONS VIA MATHEMATICAL ANALYSIS

In this section, we conduct mathematical analysis on two examples to better understand how EI improves long-term fairness compared to existing fairness notions. We first introduce a basic setup for the mathematical analysis and then apply it to two examples to compare EI with BE, ERM, and ER in terms of their one-step impact on data distributions.

## D.1    BASIC SETUP AND PRELIMINARIES

### D.1.1    CLASSIFIERS

We consider the $z-$aware classifier, having

$$f(x) = \begin{cases} \mathbf{1}\{x \geq \tau_0\}, & z = 0, \\ \mathbf{1}\{x \geq \tau_1\}, & z = 1, \end{cases}$$

which is parameterized by the threshold pair $(\tau_0, \tau_1)$. We assume the effort budget is $\delta = \frac{m}{2}$ where $m$ is defined in the dataset.

Recall the zero equal improvability (EI) disparity condition is:

$$\mathbb{P}\left(\max_{\mu(\Delta x) \leq \delta} f(x + \Delta x) \geq 0.5 \mid f(x) < 0.5, z = 0\right) = \mathbb{P}\left(\max_{\mu(\Delta x) \leq \delta} f(x + \Delta x) \geq 0.5 \mid f(x) < 0.5, z = 1\right)$$

where $\mu(\Delta x) = |\Delta x|$. This condition is equivalent to

$$\frac{\int_{\tau_0 - \delta}^{\tau_0} p_0(x)dx}{\int_{-\infty}^{\tau_0} p_0(x)dx} = \frac{\int_{\tau_1 - \delta}^{\tau_1} p_1(x)dx}{\int_{-\infty}^{\tau_1} p_1(x)dx}. \tag{15}$$

We denote the improvability ratio of each group as

$$r_0(\tau_0) = \frac{\int_{\tau_0 - \delta}^{\tau_0} p_0(x)dx}{\int_{-\infty}^{\tau_0} p_0(x)dx}, \tag{16}$$

$$r_1(\tau_1) = \frac{\int_{\tau_1 - \delta}^{\tau_1} p_1(x)dx}{\int_{-\infty}^{\tau_1} p_1(x)dx} \tag{17}$$

The classifier that satisfies this zero EI disparity condition and minimizes the error rate is denoted as the optimal EI classifier.

Recall that the bounded effort (BE) fairness constraint is:

$$\mathbb{P}\left(\max_{\mu(\Delta x) \leq \delta} f(x + \Delta x) \geq 0.5, f(x) < 0.5 \mid z = 0\right) = \mathbb{P}\left(\max_{\mu(\Delta x) \leq \delta} f(x + \Delta x) \geq 0.5, f(x) < 0.5 \mid z = 1\right) \tag{18}$$

where $\mu(\Delta x) = |\Delta x|$. Meanwhile, the Equal Recourse (ER) constraint is defined as

$$\mathbb{E}\left[\min_{f(x + \Delta x) \geq 0.5} \mu(\Delta x) \mid f(x) < 0.5, z = 0\right] = \mathbb{E}\left[\min_{f(x + \Delta x) \geq 0.5} \mu(\Delta x) \mid f(x) < 0.5, z = 1\right] \tag{19}$$

### D.1.2    DYNAMIC SCENARIO

Suppose each rejected sample improves its feature by

$$\varepsilon(x) = \begin{cases} \delta \cdot \mathbf{1}\{x \in [\tau_0 - \delta, \tau_0)\}, & \text{if } z = 0, \\ \delta \cdot \mathbf{1}\{x \in [\tau_1 - \delta, \tau_1)\}, & \text{if } z = 1 \end{cases}$$

Note that depending on the classifier we are using, the threshold pair $(\tau_0, \tau_1)$ changes, thus the formulation for $\varepsilon(x)$ also changes. Here, we use $\varepsilon^{\text{ERM}}(x)$ to denote the improvement of features given ERM classifier, and similarly define $\varepsilon^{\text{EI}}(x)$, $\varepsilon^{\text{BE}}(x)$ and $\varepsilon^{\text{ER}}(x)$.

Let $p_z^{\text{ERM}}(x) = p_z(x + \varepsilon^{\text{ERM}}(x))$ for $z \in \{0, 1\}$, which represent the data distribution after the features are improved based on ERM classifier. Similarly, we define $p_z^{\text{EI}}(x)$, $p_z^{\text{BE}}(x)$ and $p_z^{\text{ER}}(x)$ for EI/BE/ER classifiers, respectively. In the upcoming sections, we measure the total-variation (TV) distance

$$d_{TV}(p_0, p_1) = \frac{1}{2} \int_{\mathbb{R}} |p_0(x) - p_1(x)| dx$$

between two groups after a single step of feature improvement, and provide how this measurement differs for various classifiers.

## D.2 EI VERSUS BE & ERM

### D.2.1 FINDING THE OPTIMAL CLASSIFIER FOR EACH FAIRNESS CRITERION

**Setup** Let $p_z(x)$ be the data distribution of each group $z \in \{0, 1\}$, shown in Fig. 9. We consider the case of each sample having one feature $x$, and the label is assigned as

$$y = \begin{cases} \mathbf{1}\{x \geq m/2\}, & \text{if } z = 0 \\ \mathbf{1}\{x \geq 0\}, & \text{if } z = 1 \end{cases}$$

Note that we have $\mathbb{P}(y = 0|z = 0) = \frac{3}{4}$, $\mathbb{P}(y = 1|z = 0) = \frac{1}{4}$, $\mathbb{P}(y = 0|z = 1) = \frac{1}{2}$, and $\mathbb{P}(y = 1|z = 1) = \frac{1}{2}$. We set $\mathbb{P}(z = 0) = \frac{1}{4}$ and $\mathbb{P}(z = 1) = \frac{3}{4}$.

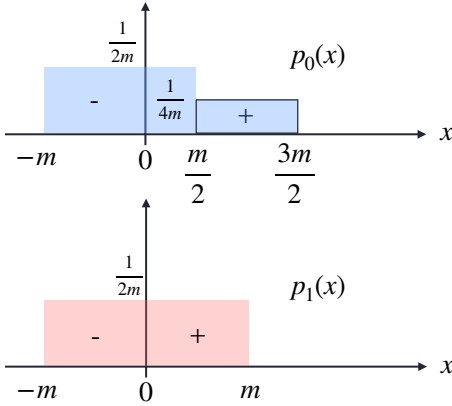

Figure 9: The data distribution $p_z(x)$ for each group $z \in \{0, 1\}$. Samples with $(+)$ sign have the true label $y = 1$, while samples with $(-)$ sign have the true label $y = 0$.

**ERM classifier** We have

$$(\tau_0^{\text{ERM}}, \tau_1^{\text{ERM}}) = \left(\frac{m}{2}, 0\right)$$

since this threshold pair has zero classification error.

**BE classifier** We have

$$(\tau_0^{\text{BE}}, \tau_1^{\text{BE}}) = \left(\frac{m}{2}, 0\right) \tag{20}$$

since this threshold pair has zero classification error and satisfy the BE condition in (18).

**EI classifier** The optimal EI classifier for dataset in Fig. 9 is

$$(\tau_0^{\text{EI}}, \tau_1^{\text{EI}}) = (0, 0). \tag{21}$$

*Proof.* Note that $(\tau_0^{\text{EI}}, \tau_1^{\text{EI}}) = (0, 0)$ has error rate $\mathbb{P}(\text{error}) = \mathbb{P}(z = 0)\mathbb{P}(\text{error}|z = 0) + \mathbb{P}(z = 1)\mathbb{P}(\text{error}|z = 1) = \frac{1}{4} \cdot \frac{1}{4} + \frac{3}{4} \cdot 0 = \frac{1}{16}$. We prove that no other classifier satisfing EI condition in (15) is having error rate less than $\frac{1}{16}$. Note that when $|\tau_1| > \frac{m}{6}$, the error rate is larger than $\frac{1}{16}$. Thus it is sufficient to consider cases when $|\tau_1| \leq \frac{m}{6}$. Combining this with the fact that

- $r_0(\tau_0)$ in (16) and $r_1(\tau_1)$ in (17) are monotonically decreasing,

- EI condition in (15) is satisfied when $r_0(\tau_0) = r_1(\tau_1)$ holds,

- $r_0(\tau_0) = r_1(\tau_1)$ holds for $\tau_0 = \tau_1 \leq \frac{m}{2}$,

we can see that the optimal EI classifier satisfies $\tau_0 = \tau_1$ and $|\tau_1| \leq \frac{m}{6}$. The error rate for these classifiers is represented as $\mathbb{P}(\text{error}) = \mathbb{P}(z = 0)\mathbb{P}(\text{error}|z = 0) + \mathbb{P}(z = 1)\mathbb{P}(\text{error}|z = 1) = \frac{1}{4} \cdot (\frac{m}{2} - \tau_0) \cdot \frac{1}{2m} + \frac{3}{4} \cdot |\tau_1| \cdot \frac{1}{2m}$. Plugging in $\tau_0 = \tau_1$ and optimizing the error probability over $|\tau_1| \leq \frac{m}{6}$ completes the proof. $\qquad\square$

### D.2.2 TOTAL-VARIATION DISTANCE BETWEEN TWO GROUPS

For the dataset given in Fig. 9, the total-variation distance between two groups for each classifier is:

$$d_{TV}(p_0^{\text{ERM}}, p_1^{\text{ERM}}) = 0.5,$$
$$d_{TV}(p_0^{\text{BE}}, p_1^{\text{BE}}) = 0.5,$$
$$d_{TV}(p_0^{\text{EI}}, p_1^{\text{EI}}) = 0.125.$$

*Proof.* Since ERM solution is identical to BE solution, proving the above equation for BE and EI is sufficient. Recall that the expression of BE/EI classifiers are in (20) and (21). Using this expression, we can derive the distribution of each group:

$$p_0^{\text{BE}}(x) = \begin{cases} \frac{3}{4m}, & \text{if } x \in [\frac{m}{2}, m] \\ \frac{1}{2m}, & \text{if } x \in [-m, 0] \\ \frac{1}{4m}, & \text{if } x \in [m, \frac{3m}{2}] \\ 0, & \text{if } x \in [0, \frac{m}{2}] \text{ or } x \notin [-m, \frac{3m}{2}] \end{cases},$$

$$p_1^{\text{BE}}(x) = \begin{cases} \frac{1}{m}, & \text{if } x \in [0, \frac{m}{2}] \\ \frac{1}{2m}, & \text{if } x \in [-m, -\frac{m}{2}] \cup [\frac{m}{2}, m] \\ 0, & \text{if } x \in [-\frac{m}{2}, 0] \text{ or } x \notin [-m, m] \end{cases},$$

$$p_0^{\text{EI}}(x) = \begin{cases} \frac{1}{m}, & \text{if } x \in [0, \frac{m}{2}] \\ \frac{1}{2m}, & \text{if } x \in [-m, -\frac{m}{2}] \\ \frac{1}{4m}, & \text{if } x \in [\frac{m}{2}, \frac{3m}{2}] \\ 0, & \text{if } x \in [-\frac{m}{2}, 0] \text{ or } x \notin [-m, \frac{3m}{2}] \end{cases},$$

$$p_1^{\text{EI}}(x) = p_1^{\text{BE}}(x) \quad \forall x$$

From this expression, we can derive the total-variation distance, which completes the proof. $\qquad\square$

### D.3 EI VERSUS ER

### D.3.1 FINDING THE OPTIMAL CLASSIFIER FOR EACH FAIRNESS CRITERION

**Setup** Let $p_z(x)$ be the data distribution of each group $z \in \{0, 1\}$, show in Fig. 10. We consider the case of each sample having one feature $x$, and the label is assigned as

$$y = \begin{cases} \mathbf{1}\{x \geq 0\}, & \text{if } z = 0 \\ \mathbf{1}\{x \geq 0\}, & \text{if } z = 1 \end{cases}.$$

Note that we have $\mathbb{P}(y = 0 \mid z = 0) = \mathbb{P}(y = 0 \mid z = 1) = \mathbb{P}(y = 1 \mid z = 0) = \mathbb{P}(y = 1 \mid z = 1) = \frac{1}{2}$. We set $\mathbb{P}(z = 0) = \mathbb{P}(z = 1) = \frac{1}{2}$.

**ER classifier** The optimal ER classifier for the dataset in Fig. 10 is

$$(\tau_0^{\text{ER}}, \tau_1^{\text{ER}}) = (-9m, 0).\qquad\qquad(22)$$

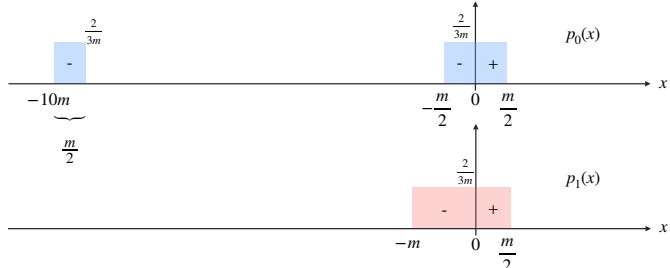

Figure 10: The data distribution $p_z(x)$ for each group $z \in \{0, 1\}$. Samples with $(+)$ sign have the true label $y = 1$, while samples with $(-)$ sign have the true label $y = 0$.

*Proof.* Note that the accepting thresholds $(-9m, 0)$ has error rate $\mathbb{P}(\text{error}) = \mathbb{P}(\text{error} \mid z = 0)\mathbb{P}(z = 0) + \mathbb{P}(\text{error} \mid z = 1)\mathbb{P}(z = 1) = \frac{1}{3} \cdot \frac{1}{2} + 0 \cdot \frac{1}{2} = \frac{1}{6}$. We prove that no other classifier satisfying ER constraint (19) is having error rate less than $\frac{1}{6}$.

The necessary condition for the classifier to have an error rate less than $\frac{1}{6}$ is $\tau_0, \tau_1 \in (-\frac{m}{2}, \frac{m}{2})$. Note that when $\tau_0 \in (-\frac{m}{2}, \frac{m}{2})$, we have the recourse of the group 0:

$$
\mathbb{E}\left[ \min_{f(x+\Delta x)\geq 0.5} \mu(\Delta x) \mid f(x) < 0.5, z = 0 \right]
$$
$$
= \mathbb{E}\left[ \tau_0 - x \mid x < \tau_0, z = 0 \right]
$$
$$
= \tau_0 - \underbrace{\mathbb{E}\left[ x \mid x < \tau_0, z = 0 \right]}_{\varphi(\tau_0)}, \tag{23}
$$

where

$$
\varphi(\tau_0) = \mathbb{E}\left[ x \mid x < \tau_0, z = 0 \right] \tag{24}
$$
$$
= \underbrace{\mathbb{P}\left( x \in \left[-10m, -\frac{19}{2}m\right] \mid x < \tau_0, z = 0 \right) \mathbb{E}\left[ x \mid x \in \left[-10m, -\frac{19}{2}m\right], z = 0 \right]}_{\triangleq \lambda}
$$
$$
+ \underbrace{\mathbb{P}\left( x \in \left[-\frac{m}{2}, \tau_0\right] \mid x < \tau_0, z = 0 \right)}_{1-\lambda} \mathbb{E}\left[ x \mid x \in \left[-\frac{m}{2}, \tau_0\right], z = 0 \right],
$$

for all $\tau_0 \in \left[-\frac{m}{2}, \frac{m}{2}\right]$, where $\lambda \in \left[\frac{1}{3}, 1\right]$. Since $\mathbb{E}\left[ x \mid x \in \left[-10m, -\frac{19}{2}m\right], z = 0 \right] < \mathbb{E}\left[ x \mid x \in \left[-\frac{m}{2}, \tau_0\right], z = 0 \right]$, (24) is a decreasing function. Consequently,

$$
\leq \frac{1}{3}\mathbb{E}\left[ x \mid x \in \left[-10m, -\frac{19}{2}m\right], z = 0 \right] + \frac{2}{3}\mathbb{E}\left[ x \mid x \in \left[-\frac{m}{2}, \tau_0\right], z = 0 \right]
$$
$$
= -\frac{39}{12}m + \frac{\tau_0}{3} - \frac{m}{6} = \frac{\tau_0}{3} - \frac{41}{12}m.
$$

Thus, the recourse of group 0 is

$$
(23) \geq \frac{2\tau_0}{3} + \frac{41}{12}m \geq \frac{37}{12}m \tag{25}
$$

Meanwhile, when $\tau_1 \in (-\frac{m}{2}, \frac{m}{2})$, we have the recourse of the group 1:

$$
\begin{aligned}
\mathbb{E} &\left[ \min_{f(x+\Delta x) \geq 0.5} \mu(\Delta x) \mid f(x) < 0.5, z = 1 \right] \\
&= \mathbb{E}\left[ \tau_1 - x \mid x < \tau_1, z = 0 \right] \\
&= \tau_1 - \mathbb{E}\left[ x \mid x < \tau_1, z = 0 \right] \\
&= \tau_1 - \frac{\tau_1 - m}{2} \\
&= \frac{\tau_1 + m}{2} \in \left( \frac{m}{4}, \frac{3}{4}m \right),
\end{aligned}
\tag{26}
$$

Therefore, combining (25) and (26) implies that when $\tau_0, \tau_1 \in (-\frac{m}{2}, \frac{m}{2})$, the ER constraint (19) cannot be satisfied. Thus, the way for achieving error rate $< \frac{1}{6}$ is $\tau_0 < -\frac{m}{2}$ and $\tau_1 = 0$. Note that the recourse of group 0 which is written in (23) is a strictly increasing function when $\tau_0 < -\frac{m}{2}$. Therefore, there exists a unique classifier that achieves both EI while maintaining error rate $< \frac{1}{6}$. One can easily verify that $(-9m, 0)$ is the optimal ER classifier. $\qquad\square$

**EI classifier** We have

$$
(\tau_0^{\text{EI}}, \tau_1^{\text{EI}}) = (0, 0) \tag{27}
$$

since this threshold pair has zero classification error and satisfies the EI constraint (15).

### D.3.2 TOTAL-VARIATION DISTANCE BETWEEN TWO GROUPS

For the dataset given in Fig. 10, the total variation distance between two groups for EI and ER classifier is

$$
d_{TV}(p_0^{\text{ER}}, p_1^{\text{ER}}) = \frac{2}{3},
$$
$$
d_{TV}(p_0^{\text{EI}}, p_1^{\text{EI}}) = \frac{1}{3}.
$$

*Proof.* By (27) and (22),

$$
p_0^{\text{ER}}(x) = \begin{cases} \frac{2}{3m}, & \text{if } x \in [-\frac{19m}{2}, -8m] \\ \frac{2}{3m}, & \text{if } x \in [-\frac{m}{2}, \frac{m}{2}] \\ 0, & \text{o.w.} \end{cases},
$$

$$
p_1^{\text{ER}}(x) = \begin{cases} \frac{2}{3m}, & \text{if } x \in [-m, -\frac{m}{2}] \\ \frac{4}{3m}, & \text{if } x \in [0, \frac{m}{2}] \\ 0, & \text{o.w.} \end{cases},
$$

$$
p_0^{\text{EI}}(x) = \begin{cases} \frac{2}{3m}, & \text{if } x \in [-10m, -\frac{19m}{2}] \\ \frac{4}{3m}, & \text{if } x \in [0, \frac{m}{2}] \\ 0, & \text{o.w.} \end{cases},
$$

$$
p_1^{\text{EI}}(x) = \begin{cases} \frac{2}{3m}, & \text{if } x \in [-m, -\frac{m}{2}] \\ \frac{4}{3m}, & \text{if } x \in [0, \frac{m}{2}] \\ 0, & \text{o.w.} \end{cases}.
$$

For this expression, we can derive the total-variation distance, which completes the proof. $\qquad\square$

