# OpenReview forum: "Equal Improvability: A New Fairness Notion Considering the Long-term Impact"
_ICLR.cc/2023/Conference — ICLR 2023 poster_

### Official Review · Reviewer_Eon1 · 2022-10-17

**Confidence:** 2
**Correctness:** 3
**Technical Novelty And Significance:** 2
**Empirical Novelty And Significance:** 2
**Recommendation:** 6

**Clarity, Quality, Novelty And Reproducibility:**

The paper is fairly well written.
 - The proposed methods are well explained with clear notations
 - Related works are thoroughly reviewed.

I haven't checked the details, but it seems there is no issue with reproducibility.



**Strength And Weaknesses:**

- Strength
  - The paper considers a timely important problem: training a fair machine learning model.

- Weaknesses
  - **Novelty:** It seems the novelty is weak. Specifically, the author proposed EI
$$
\mathbb{P}\left( \max_{  \mu( \Delta \mathbf{x}_1 ) \leq \delta} f(\mathbf{x} + \Delta\mathbf{x}) > 0.5 \mid f(\mathbf{x}) < 0.5, \mathbf{z}=z \right) = \mathbb{P}\left( \max_{  \mu( \Delta \mathbf{x}_1 ) \leq \delta} f(\mathbf{x} + \Delta\mathbf{x}) > 0.5 \mid f(\mathbf{x}) < 0.5 \right)
$$
which is equivalent to
$$ \mathbb{E} \left[  \mathbf{1} \set{ \max_{  \mu( \Delta \mathbf{x}_1 ) \leq \delta} f(\mathbf{x} + \Delta\mathbf{x}) > 0.5 }  \mid f(\mathbf{x}) < 0.5, \mathbf{z}=z \right] = \mathbb{E} \left[  \mathbf{1} \set{ \max_{  \mu( \Delta \mathbf{x}_1 ) \leq \delta} f(\mathbf{x} + \Delta\mathbf{x}) > 0.5 }  \mid f(\mathbf{x}) < 0.5 \right]. $$
Given that the same conditional distributions are considered in Equal Resource (ER) by Gupta et al. (2019), it seems the current idea might be a naive variant of the previous idea:
$$
\mathbb{E} \left[ \min_{f(\mathbf{x}^{\prime})>0.5} \mu( \mathbf{x}^\prime - \mathbf{x})  \mid f(\mathbf{x}) < 0.5, \mathbf{z}=z \right] = \mathbb{E} \left[  \min_{f(\mathbf{x}^{\prime})>0.5} \mu( \mathbf{x}^\prime - \mathbf{x})  \mid f(\mathbf{x}) < 0.5 \right].
$$
Also, using a fairness regularization function is not a new idea as the authors mentioned in section 3. To be fair, I believe the work can still have strengths even with these similarities. However, it is not clear why the proposed idea is powerful without comprehensive comparisons with ER. The authors explained the limitation of ER in section 2, but the behavior of EI on outliers is not explicitly examined.

  - **Mismatch:** Mismatch between the motivation and the proposed method: the authors discussed the long-term fairness in Abstract and Introduction, but the proposed method is not directly solving the dynamics of distribution shifts. In section 4.1, the dynamic setting is considered, but it is not clear how the objective function in section 4.1.1 is related to the equation (1) in section 3.

  - **Experiments:** Most of experiments consider fairly simple settings ($\mathbf{x}_t \in \mathbb{R}$, $\mathcal{Z}=\{0,1\}$, and $T$ is small) that are not realistic. The real problems should be higher dimensionality on $\mathbf{x}$ and have more sensitive classes. Moreover, it is not clear why achieving EI helps to improve long-term fairness.



**Summary Of The Paper:**

The paper proposes a new notion of fairness called Equal Improvability (EI) that requires a binary classifier to be equally difficult to change predictions for each sensitive attribution group. To achieve this property, three different penalizations are discussed: covariance-based EI penalty, KDE-based EI penalty, and loss-based EI Penalty. The authors empirically showed that the three penalization methods achieve EI disparity. In addition, they claimed that EI promotes long-term fairness in dynamic scenarios using synthetic data.

**Summary Of The Review:**

I think the main issue of the current submission is the lack of novelty and comprehensive experiments. I am happy to revise my reviews if there is anything trivial I miss something.

---

> ### Author Response · Authors · 2022-11-19
> **Response to Reviewer Eon1**
>
> > [R4-1] Novelty seems to be weak given the work by Gupta et al. (2019) suggesting Equal Resource (ER). It is not clear why the proposed idea (EI) is powerful without comprehensive comparisons with ER. Especially, the behaviors of EI and ER under outliers are not examined.
>
> It is true that the mathematical definitions of EI and ER are apparently similar:
>
> * EI: $\mathbb{P}\left(\max\limits_{\mu(\Delta x) \leq \delta} f(x + \Delta x) \ge  0.5 \mid f(x)  <   0.5, z \right)  =  \mathbb{P}\left(\max\limits_{\mu(\Delta x) \leq \delta} f(x + \Delta x)  \ge  0.5 \mid f(x) < 0.5 \right)$
> * ER: $\mathbb{E}\left[\min\limits_{f(x + \Delta x) \ge 0.5}\mu(\Delta  x)\mid f(x)<0.5, z\right]=\mathbb{E}\left[\min\limits_{f(x + \Delta x) \ge 0.5}\mu(\Delta  x)\mid f(x)<0.5\right]$
>
>
> However, the behaviors of EI/ER disparities are quite different especially when the dataset has outliers. Suppose an outlier, having feature-attribute pair $(x,z)$, is added to the dataset with $n$ samples. Let the outlier is misqualified $f(x) < 0.5$, and requires effort $\mu(\Delta x) = M$ to achieve $f(x+\Delta x) \ge 0.5$.
>
> In such case, the EI disparity increases at most $\frac{1}{n}$ since EI measures the *portion* of samples with improved outcomes after making efforts. On the other hand, the ER disparity increases by $\frac{M}{n}$, since ER measures the minimum required efforts *averaged* over all samples. Note that a single outlier with a large $M$ can significantly increase the ER disparity, which does not happen for EI. Thus, one can observe that EI is much more robust to outliers, compared with ER. To support the above claim, we added the empirical results in Section C.5.1 of Appendix.
>
> > [R4-2] Why does achieving EI improve long-term fairness?
>
> Thanks for the sharp question. In Section D of Appendix, we added theoretical results to demonstrate the benefit of EI in improving long-term fairness, compared with other fairness notions (BE and ER). To be specific, we consider the dynamic scenario where each rejected sample (that is close to the decision boundary) makes effort to improve its feature, in order to get accepted in the next round. For EI/BE/ER/ERM classifiers, we analyzed how the feature distributions of different groups behave in the next round. We proved that using the EI classifier is beneficial to reduce the total-variation distance $d_{TV}$ between different groups, compared with classifiers constrained with other first-order fairness notions as well as the ERM solution, i.e.,
> $$
> d_{TV}(EI) < d_{TV}(BE),
> d_{TV}(EI) < d_{TV}(ER),
> d_{TV}(EI) < d_{TV}(ERM).
> $$
> We expect these results provide insight on how EI has benefits in equalizing the distributions of different groups.
>
>
> > [R4-3] Most of experiments consider fairly simple settings ($x_t\in\mathbb{R}$, $Z=0,1$, and $T$ is small) that are not realistic.
>
> Thanks for your suggestion. In the conclusion section of the revised manuscript, we mentioned it as one of the limitations of our work.

---

> > ### Comment · Reviewer_Eon1 · 2022-12-04
> > **Thank you for the response!**
> >
> > I greatly thank the authors for their detailed responses. As for the EI vs ES, the example in the author's response makes a lot of sense and addresses my concern to some extent, but I still think it would be better to have numerical results using real datasets (or synthetic datasets). As for the realistic settings, I thank the authors for clearly identifying the limitation of this work. I understand that it would be an interesting future extension.
> >
> > I would like to keep the initial evaluation score because I think it is not yet convincing if the proposed idea correctly considers the dependency that arises by using the previous samples. For instance, as far I understand, the update rule in Section 4.2.1 is only taking the current $t$-th data point, i.e., $(x_t, y_t)$, not the previous data points. Also, a new theoretical result in Appendix D considers a **single step** of feature improvement. I am not completely sure if this proposed method is necessarily designed for long-term fairness.

---

> > > ### Author Response · Authors · 2022-12-08
> > > **Thank you for your feedback**
> > >
> > > Dear Reviewer Eon1,
> > >
> > > Thanks for your valuable feedback.
> > >
> > > > Q. I think it would be better to have numerical results using real datasets or synthetic datasets to compare EI and ER.
> > >
> > > A. We will take your advice and add numerical results.
> > >
> > > > Q: I am not completely sure if this proposed method is necessarily designed for long-term fairness.
> > >
> > > A. We will modify the flow of our paper to emphasize the definition of EI itself, and tone down our claim that EI improves long-term fairness.

---

> > > > ### Comment · Reviewer_Eon1 · 2022-12-08
> > > > **Response from Reviewer Eon1**
> > > >
> > > > Thank the authors for the responses. Throughout the discussions with authors and reviewers, I notice this paper has some limits, but it still shows nontrivial benefits in studying long-term fairness. I updated my evaluation.

---

> > > > > ### Author Response · Authors · 2022-12-12
> > > > > **Reply to Reviewer Eon1**
> > > > >
> > > > > Thanks for reading our response carefully and updating your evaluation! Based on your suggestions, we have added a new numerical experiment and modified the flow of our manuscript. Our responses are detailed below.
> > > > >
> > > > > > I think it would be better to have numerical results using real datasets or synthetic datasets to compare EI and ER.
> > > > >
> > > > > | Gaussian Dataset (in Fig.6) | Metric | EI Classifier | ER Classifier |
> > > > > | -------- | -------- | -------- |--------|
> > > > > | Dataset Without Outlier    | Error Rate$(\downarrow)$   | $.001\pm.001$     |$.001\pm.001$
> > > > > ||EI Disp.$(\downarrow)$|$.001\pm.001$|$.005\pm.001$|
> > > > > ||ER Disp$(\downarrow)$|$.004\pm.001$|$.001\pm.001$|
> > > > > | Dataset With Outlier    | Error Rate$(\downarrow)$   | $.001\pm.001$     |$.020\pm.001$
> > > > > ||EI Disp.$(\downarrow)$|$.017\pm.001$|$.348\pm.001$|
> > > > > ||ER Disp$(\downarrow)$|$.020\pm.001$|$.291\pm.001$|
> > > > >
> > > > > The above table compares EI and ER classifiers, for the Gaussian dataset shown in Fig.6 of Appendix C.5.1. The first row shows the results when training data does not have outliers, while the second row shows the results when there are outliers in the training data. We report the performance measured on the test dataset which does not have any outliers. One can see that the performance of ER classifier drops drastically when trained on the dataset with outliers, while EI classifier is much more robust to outliers.
> > > > >
> > > > >
> > > > > > I am not completely sure if this proposed method is necessarily designed for long-term fairness.
> > > > >
> > > > > In the revised manuscript, we will tone down our claim on long-term fairness. Specifically, we will clarify that our EI fairness notion is designed for taking the improvability of rejected samples into consideration, and we additionally found that EI improves long-term fairness. For example, below we share our new abstract:
> > > > >
> > > > > **[Revised abstract]**
> > > > > Devising a fair classifier that does not discriminate against different groups is an important problem in machine learning. Recently, effort-based fairness notions are getting attention, which considers the scenarios of each individual making effort to improve its feature over time. Such scenarios happen in the real world, e.g., college admission and credit loaning, where each rejected sample makes effort to change its features to get accepted afterward. In this paper, we propose a new effort-based fairness notion called Equal Improvability (EI), which equalizes the potential acceptance rate of the rejected samples across different groups assuming a bounded level of effort will be spent by each rejected sample. We also propose and study three different approaches for finding a classifier that satisfies the EI requirement. Through experiments on both synthetic and real datasets, we demonstrate that the proposed EI-regularized algorithms encourage us to find a fair classifier in terms of EI. Additionally, we ran experiments on dynamic scenarios which highlight the advantages of our EI metric in equalizing the distribution of features across different groups, after the rejected samples make some effort to improve. Finally, we provide mathematical analyses of several aspects of EI: the relationship between EI and existing fairness notions, and the effect of EI in dynamic scenarios. Codes are available in an anonymous GitHub repository.

---

### Official Review · Reviewer_q7oe · 2022-10-23

**Confidence:** 4
**Correctness:** 4
**Technical Novelty And Significance:** 3
**Empirical Novelty And Significance:** 3
**Recommendation:** 6

**Clarity, Quality, Novelty And Reproducibility:**

The paper is clear and it is easy to follow the high-level ideas of the paper. The paper is relevant to the community. The definitions are novel though the connections to prior work should be described more clearly (see above). The code is provided.

**Strength And Weaknesses:**

------------------------------------------
Strengths:
------------------------------------------
-- The delayed impact of fairness in machine learning has not received much attention in the community (mainly due to the hardness of modeling this impact) so in that sense, it is great to see papers on this topic.

--The paper is pretty well written and it is clear to follow the high-level ideas of the paper. The delayed impact notion of fairness is natural and sensible.

------------------------------------------
Weaknesses:
------------------------------------------
-- The main technical contribution of the paper is the definition. Hence, the technical novelty (when enforcing the equal improvability constraint) is limited.

-- Section 4.2 contains the most interesting part of the paper, though some of the details are hard to follow/justify. It would be great to justify the choice of $\epsilon(x)$ as well as describe in more detail how the updates to the classifier are done.

------------------------------------------
Minor Comments:
------------------------------------------
-- Theorem 2.5 and Corollary 2.6 show that equal improvability and bounded effort are mathematically different. It would be nice to
describe this in more detail as it is not clear to me how to set the reward function in Heidari et al., 2019 to connect the two works.

-- Are there deeper connections between equal recourse and equal improvability that are overlooked? If not, is not it surprising that these two methods work exactly the same in the experiments in section 4.2?

-- Please stop using the Adult dataset: https://proceedings.neurips.cc/paper/2021/hash/32e54441e6382a7fbacbbbaf3c450059-Abstract.html

-- The Y value in Figure 4 should be bounded by 1. Why is it above one in the left two sub-figures?


**Summary Of The Paper:**

The paper studies the delayed impact of fairness. The authors introduce a notion of fairness called \emph{equal improvability}, which, subject to a bounded amount of improvement, equalizes the probability of acceptance for the rejected members across all populations. The authors aim to solve the typical loss minimization problem of classification subject to the equal improbability constraint. Since the constraint is non-convex, the paper proposes 3 different approaches to solve the constrained optimization. The paper concludes with empirical analysis both in the one-shot game as well as the repeated setting.

**Summary Of The Review:**

Overall, I enjoyed reading the paper. I think it is a nice addition to the growing literature on the delayed impact of fairness. Though the technical contributions are not strong, it can be a welcome addition to the ICLR's program.

------------------------
Post rebuttal:
------------------------
I want to sincerely thank the authors for answering my questions thoroughly. I have also read the other reviews and author responses. I think we all feel that this work can go either way. I am still (slightly) in favor of acceptance though.

---

> ### Author Response · Authors · 2022-11-19
> **Response to Reviewer q7oe**
>
> > [R3-1] The main technical contribution of the paper is the definition. Hence, the technical novelty (when enforcing the equal improvability constraint) is limited.
>
> We agree with the reviewer that we enforced the EI constraint using a standard regularization-based approach. However, we want to emphasize that we still have novel contributions of
> * suggesting a new first-order fairness notion (called EI) that encourages the long-term fairness
> * providing mathematical analysis on the solution of EI-regularized optimization problem
>
>
> Especially, regarding the second contribution, Sec.B.2 provides the closed-form solution of inner maximization for generalized linear models as well as the adversarial training-based approach for a more general setup. Sec.B.3 also derives the optimal linear EI classifier for Gaussian mixtures.
>
> > [R3-2] Section 4.2 contains the most interesting part of the paper, though some of the details are hard to follow/justify. It would be great to justify the choice of $\epsilon(x)$ as well as describe in more detail how the updates to the classifier are done.
>
> Thanks for appreciating our Section 4.2 discussing the dynamic scenarios, and thanks for the sharp question. Below we first justify the choice of $\epsilon(x)$, the amount of effort a sample (having feature $x$) takes in the dynamic setting.
>
> * Updating individual's features (choice of $\epsilon(x)$)
>
> At first, we designed a model by assuming that each individual makes an effort, where the amount of the effort he/she makes is (1) proportional to the reward (the improvement on the outcome) it will get by making such effort, and (2) inversely proportional to the required amount of efforts to improve the outcome, as shown in the below equation:
>     $$
>     \begin{align}
>     \text{Realized effort $\epsilon(x)$} &\triangleq \frac{\text{improvement on the outcome}}{\text{required efforts for improving the outcome}^2} \\ &= \frac{ \mathbf{1}\{x < \tau_t^{(z)}\} }{(\tau_t^{(z)} - x)^2},
>     \end{align}
>     $$
> where $\tau_t^{(z)}$ is the accepting threshold for the individuals from group $z$ at round $t$, and $x$ is the feature. This equation can be interpreted as follows: an individual that is unqualified $(x < \tau_t^{(z)})$ is willing to make positive effort, but he/she is less motivated if the distance $\tau_t^{(z)} - x$ to the decision boundary is too large.
>
> In order to upper bound the amount of reward, we added a small constant $\beta > 0$ in the numerator, thus having the final form:
>     $$\epsilon(x) = \frac{\mathbf{1}\{x < \tau_t^{(z)}\}}{(\tau_t^{(z)}-x+\beta)^2}.$$
>
> * Updating the classifier
> Note that at each round $t$, each individual's features are updated. Accordingly, we also update the classifier (having parameters $\tau_t^{(0)}$ and $\tau_t^{(1)}$) in the same round. We update the EI classifier based on the following constrained optimization problem:
>     $$
>     \min \text{(EI Disparity)} \quad \text{s.t.}\quad \text{error rate} \leq \text{threshold},
>     $$
>     which implies that we aim to find a classifier that guarantees good accuracy and small EI disparity. This optimization can be rewritten as
>     $$
>     \begin{align}
>     & \min_{\tau_t^{(0)}, \tau_t^{(1)}}\overbrace{\left| \mathbb{P}(\underbrace{x_t + \delta_t > \tau_t^{(0)}}_{\text{the outcome can be improved}} \mid z = 0, \underbrace{x_t <\tau_t^{(0)}}_{\text{rejected}})  - \mathbb{P}(x_t + \delta_t > \tau_t^{(1)} \mid z = 1, x_t <\tau_t^{(1)})\right|}^{\text{EI Disparity}} \\ & \quad \text{s.t.}~\overbrace{\mathbb{P}(\hat{y}_t \neq y_t)}^{\text{error rate}}\leq c.
>     \end{align}
>     $$
>
> Given the data distribution at each round $t$, we numerically solve the constrained optimization problems using a popular python module `scipy.optimize`.
>
> > [R3-3] Theorem 2.5 and Corollary 2.6 show that equal improvability and bounded effort are mathematically different.  It would be nice to describe how to set the reward function in Heidari et al. (2019) to connect the two works.
>
> Thanks for the clarification question. Below we wrote how we set the reward function so that we can connect bounded effort with the equal improvability.

---

> > ### Author Response · Authors · 2022-11-19
> > **Continued: Response to Reviewer q7oe**
> >
> > Recall that the reward function in `Heidari et al. (2019)` is defined as the benefit gained by changing an individual's characteristics from $w=(x,y)$ to $w'=(x',y')$:
> > $$\mathcal{R}(w,w')=b(f(x'),y')-b(f(x),y)$$
> > where $b$ is the benefit function, and $x'=x+\Delta x$ is the updated feature. If we set the benefit function as $b(f(x),y)=\mathbf 1_{f(x)\geq0.5}$, then the reward function becomes:
> > $$\mathcal{R}(w,w')=\mathbf 1_{f(x+\Delta x)\geq0.5}-\mathbf 1_{f(x)\geq0.5}$$
> > Then, the bounded effort fairness is defined as:
> > $$\mathbb{E}\left[\max_{\Delta x}\mathcal{R}(w,w') \text{ s.t. } \mu(\Delta x)<\delta|\text{z}=z\right]=\mathbb{E}\left[\max_{\Delta x}\mathcal{R}(w,w') \text{ s.t. } \mu(\Delta x)<\delta\right] \text{ for all } z$$
> > As you can see,
> > $$
> > \left(\max_{\Delta x}\mathcal{R}(w,w') \text{ s.t. } \mu(\Delta x)<\delta\right) = \begin{cases} 1 \text{ if } \max_{\mu(\Delta x)<\delta}f(x+\Delta x)\geq0.5 \text{ and } f(x)<0.5 \\ 0 \text{ otherwise}\end{cases}
> > $$
> > Therefore, we can write expectation as probability:
> > $$
> > P\left(\max\limits_{\mu(\Delta x) \leq \delta}f(x+\Delta x)\geq0.5, f(x) < 0.5 \mid  \text{z} = z\right) = P\left(\max\limits_{\mu(\Delta x) \leq \delta}f(x+\Delta x)\geq0.5, f(x) < 0.5 \right),
> > $$
> >
> > which is in Table 1 of the manuscript. We will add these details in the revised manuscript.
> >
> >
> > > [R3-4] ER and EI behave quite similarly in the experiment in Sec.4.2. Does this mean there are some deep connections between them that are overlooked by the authors?
> >
> > We thank the reviewer for the great question. In fact, we found that under certain cases, EI criterion implies ER criterion (see Lemma A.1 in Appendix A.2 in the revised manuscript). Despite of the similarity shared by EI and ER, our EI is still impactful because of its robust to outliers. Figure 6 in the revised manuscript demonstrates the vulnerability of ER to outliers.
> >
> > > [R3-5] Please stop using the Adult dataset: https://proceedings.neurips.cc/paper/2021/hash/32e54441e6382a7fbacbbbaf3c450059-Abstract.html
> >
> > Thanks for your suggestion. Below we add experimental results on the new Adult dataset, only for a subset due to the time limitation. To be specific, we tested on the dataset `ACSIncome-CA` in the new Adult dataset.
> >
> > | Metric | ERM | Covariance-based | KDE-based | Loss-based |
> > |:---:|:---:|:---:|:---:|:---:|
> > | Error Rate ($\downarrow$) | $.184 \pm .000$ | $.200 \pm .000$ | $.196 \pm .000$ | $.193 \pm .000$ |
> > | EI Disp ($\downarrow$) | $.031 \pm .001$ | $.008 \pm .001$ | $.005 \pm .001$ | $.006 \pm .001$ |
> >
> > We observe that our proposed approaches perform well on the new dataset.
> >
> > > [R3-6] The $y$-axis value of Figure 4 should be bounded by 1. Why is it above one in the left two sub-figures?
> >
> > Thanks for pointing this out. Instead of reporting the total variation distance $d_{\operatorname{TV}}(P,Q)=\frac{1}{2} \lVert P-Q \rVert_1$, we reported the norm $\lVert P-Q \rVert_1$ which is bounded above by 2. We corrected this in our revised manuscript.

---

> > > ### Comment · Reviewer_q7oe · 2022-12-09
> > > **Re: Comment**
> > >
> > > I want to sincerely thank the authors for their response. Similar to others, I think the claims of long-term improvement should be toned down. I do like the paper though and I think it would be a nice addition.
> > >
> > > In the experiments, EI and all other previous notions cause improvement. Are there cases where only EI causes improvement but other methods such as DP and EO cause harm?

---

> > > > ### Author Response · Authors · 2022-12-12
> > > > **Reply to Reviewer q7oe: Part I**
> > > >
> > > > First, we want to thank the reviewer for carefully reading our response. Our answer to your question is detailed below.
> > > >
> > > > > ... In the experiments, EI and all other previous notions cause improvement. Are there cases where only EI causes improvement but other methods such as DP and EO cause harm?
> > > >
> > > > Yes. As per the reviewer's comment, below we provide one example in which EI improves long-term fairness, while some other methods (EO, BE, and ER) harm the long-term fairness. Below the table, we also provide the intuition behind why such a situation happens, by visualizing the probability density function of each group and the acceptance threshold of each method.
> > > >
> > > > In this example, we modified the effort function from $\epsilon(x) = \frac{1}{(\tau_t^{(z)} - x +\beta)^2}\mathbf{1}_{x<\tau_t^{(z)}}$ to $\epsilon(x) = \log\left(\max(\frac{1}{(\tau_t^{(z)} - x +\beta)^2},1)\right)\mathbf{1}_{x<\tau_t^{(z)}}$ so that the rejected individuals make less improvement than the setting in the original manuscript. Here $\tau_t^{(z)}$ is the acceptance threshold for group $z$ at iteration $t$, and $\beta>0$ is a small constant to avoid zero denominators. We set $\beta = 0.2$ and the true acceptance rate $\alpha = 0.5$. The table below reports how long-term unfairness changes when the initial distribution is $x \mid z=0 \sim \mathcal{N}(0.0, 3.0^2)$ for group 0 and $x \mid z=1 \sim \mathcal{N}(1.0, 1.0^2)$ for group 1. In the revised manuscript, we will visualize this table to better show the advantage of EI in improving long-term fairness.
> > > >
> > > > | | Long-term Unfairness ($\downarrow$) | Long-term Unfairness ($\downarrow$)| Long-term Unfairness ($\downarrow$)| Long-term Unfairness ($\downarrow$)| Long-term Unfairness ($\downarrow$)|
> > > > |:--:|:--:|:--:|:--:|:--:|:--:|
> > > > |  |  DP | EO | BE| ER| EI (Ours) |
> > > > |Initial Distribution| .507| .507| .507| .507| .507|
> > > > |Iteration 2| .431| .496| .564| .492| .347|
> > > > |Iteration 4| .395| .504| .616| .518| .267|
> > > > |Iteration 6| .375| .522| .660| .558| .225|
> > > > |Iteration 8| .363| .545| .692| .599| .200|
> > > > |Iteration 10| .322| .572| .713| .615| .184|
> > > >
> > > > The following string graph provides the intuition of why EI performs the best compared to DP and EO. We illustrate the probability density function of each group, and the acceptance thresholds of EI, DP, and EO, at iteration 0.
> > > >
> > > > Note that the true acceptance thresholds satisfy 100% accuracy and thus achieve EO, i.e., $P(\hat{Y}=1 | Y=1, Z=0) = P(\hat{Y}=1 | Y=1, Z=1)$. As shown below, EO yields the result that most of the rejected individuals from group 1 (advantaged group) can make much more effort than the rejected individuals from group 0 (disadvantaged group), thereby increasing the gap between the two groups and harming long-term fairness. In contrast, EI lowers the acceptance threshold for the disadvantaged group, and increases the acceptance threshold for the advantaged group, thereby encouraging rejected individuals from the disadvantaged group to make more effort and improving long-term fairness. DP constraint results in acceptance thresholds between the acceptance thresholds of EI and EO, which helps improve long-term unfairness, but not as much as EI.

---

> > > > > ### Author Response · Authors · 2022-12-12
> > > > > **Reply to Reviewer q7oe: Part II**
> > > > >
> > > > > ```
> > > > > PDF of Group 0 (disadvantaged group) x|z=0 ~ N(0.0, 3.0^2)
> > > > >                                            EI   DP  True Acceptance threshold (EO)
> > > > >                                             |***|***|
> > > > >                                        *****|   |   |*****
> > > > >                                    ****     |   |   |     ****
> > > > >                                 ***         |   |   |         ***
> > > > >                             ****            |   |   |            ****
> > > > >                        *****                |   |   |                *****
> > > > >                ********                     |   |   |                     *********
> > > > > ***************                             |   |   |                              ***************
> > > > >
> > > > > PDF of Group 1 (advantaged group) x|z=1 ~ N(1.0, 1.0^2)
> > > > >                        True Acceptance threshold (EO)   DP  EI
> > > > >                                                     |   |   |
> > > > >                                                     | * |*  |
> > > > >                                                     |*  | * |
> > > > >                                                     |   |   |
> > > > >                                                     |   |   |
> > > > >                                                    *|   |  *|
> > > > >                                                     |   |   |
> > > > >                                                   * |   |   *
> > > > >                                                     |   |   |
> > > > >                                                  *  |   |   |*
> > > > >                                                     |   |   |
> > > > >                                                     |   |   |
> > > > >                                                 *   |   |   | *
> > > > >                                                *    |   |   |  *
> > > > >                                                     |   |   |
> > > > >                                               *     |   |   |   *
> > > > >                                              *      |   |   |    *
> > > > >                                             *       |   |   |     *
> > > > >                                           **        |   |   |      **
> > > > > ******************************************          |   |   |        *******************************
> > > > > ```

---

### Official Review · Reviewer_Zw9H · 2022-10-25

**Confidence:** 3
**Correctness:** 2
**Technical Novelty And Significance:** 3
**Empirical Novelty And Significance:** 3
**Recommendation:** 5

**Clarity, Quality, Novelty And Reproducibility:**

Overall, the paper is not hard to follow. The results are relatively clearly presented. The paper can benefit from some additional illustrations/discussions of the claimed advantages of EI. The implementation details are provided in the paper for reproducibility.

**Strength And Weaknesses:**

## Strength

The proposed EI notion combines advantages of previous notions of a similar flavor (as summarized in Table 1), and makes an effort to avoid (some) problems of previous group-level fairness notions. Three different ways to (approximately) enforce EI are also provided. The authors demonstrate the benefit of considering EI in the dynamic setting in the experiments.

## Weakness

### 1. w.r.t. the distinction between zero-order vs. first-order fairness

I am wondering what is the "order" over here. Roughly, I think the paper is pointing out the fact that EI (among others, e.g., Equal Recourse, Bounded Effort) cares about the ability to change the status quo. However, since it is a shared characteristics among some previous notions, I am not sure what specific benefit is unique to EI. Also, this distinction is only made in the beginning of the paper, and was not mentioned after Section 2. It would be very helpful if authors can share some insight over here, especially for the purpose of long-term fairness, as claimed in the paper.

### 2. the unmentioned connection between individual-level recourse fairness and EI

If I understand it correctly, the central claim w.r.t. the advantage of EI compared to other notions is the potential to "equally" improve decision outcome for those who are currently rejected. This intuition is shared by, as pointed out in the paper, effort-based fairness notions. In my opinion, the intuition is actually more aligned with individual-level recourse fairness, e.g., von Kügelgen et al. (2022) "On the Fairness of Causal Algorithmic Recourse". EI claims to be more focused on those who are potentially able to recourse. However, the fact that EI remains a group-level fairness notion indicates that individual-level recourse fairness is more fine-grained and can better serve the intended purpose laid out by the authors. Additional discussion on the connection and difference would be very helpful.

### 3. w.r.t Definition 2.1 EI

I am wondering why we should limit our consideration to $\Delta x$ such that $\Delta_M x = 0$. According to the notion, by definition mutable features do not contribute to the cost of effort but can still alter the final prediction outcome. Then, I think it is more reasonable to encode such detail in the measurement of cost instead of forcing $\Delta_M x = 0$, since this will implicitly limit the possibility to recourse for certain individuals and in turn affect the estimation of recourse probability.


**Summary Of The Paper:**

The paper proposes a new notion of group-level fairness, i.e., Equal Improvability (EI). The intuition behind the notion is that, by considering those who are rejected but are not too far away from the decision border, the effort taken by those to recourse should not vary among groups. The paper provides three empirical ways to (approximately) enforce EI in classification tasks. Empirical results in the simulated dynamics are also provided.

**Summary Of The Review:**

The paper proposes a new group-level fairness notion and three ways to empirically enforce it in classification. There are some worries about the claimed advantages of EI, as well as the formulation of the definition of EI itself (as detailed in "Strength and Weakness"). It would be greatly appreciated if authors can kindly provide additional clarifications.

====== Post rebuttal =========

Thank authors for the response. I have updated my evaluation.

---

> ### Author Response · Authors · 2022-11-19
> **Response to Reviewer Zw9H**
>
> > [R2-1] What is the distinction between zero-order and first-order fairness notions?
>
> In the table below, we provide the difference between the two categories. In short, zero-order notions consider immediate fairness, while first-order notions consider fairness measured one step after each individual improves its feature. Note that EI (suggested in our paper) as well as BE and ER are categorized into first-order notions. This distinction can be also found in the equation of each definition in Table 1 of the revised manuscript; the first-order fairness notion contains $\Delta x$ term (the improvement of feature), while the zero-order notion does not.
>
> To avoid readers' confusion, in the Introduction section of our revised manuscript, we put the definition of the first-order fairness notion.
>
> | | Zero-order Fairness Notions | First-order Fairness Notions |
> |:--:|:--:|:--:|
> | Definition | Fairness notions that do NOT consider the improvability of individuals | Fairness notions that consider the improvability of individuals
> | Examples | Demographic Parity (DP), Equal Opportunity (EO), Equalized Odds (EOD) | Equal Improvability (EI), Bounded Effort (BE), Equal Recourse (ER)|
>
> > [R2-2] What specific benefit is unique to EI, compared with ER and BE? Can authors provide more insight on how EI improves long-term fairness compared with ER and BE?
>
> As per the reviewer's questions, we have systematically investigated the specific advantages of EI over BE (`Heidari et al. 2019`) and ER (`Gupta et al., 2019`) via numerical experiments and mathematical analysis.
>
> First, our results show that EI is robust in general data sets, while ER and BE are vulnerable to cases of having outliers or imbalanced group negative rates.
>
> **(EI v.s. ER)** We start by sharing a simple analysis. Suppose an outlier, having feature-attribute pair $(x,z)$, is added to the dataset with $n$ samples. Let the outlier is misqualified $f(x) < 0.5$, and requires effort $\mu(\Delta x) = M$ to achieve $f(x+\Delta x) \ge 0.5$.
>
> In such case, the EI disparity increases at most $\frac{1}{n}$ since EI measures the *portion* of samples with improved outcomes after making efforts. On the other hand, the ER disparity increases by $\frac{M}{n}$, since ER measures the minimum required efforts *averaged* over all samples. Note that a single outlier with a large $M$ can significantly increase the ER disparity, which does not happen for EI. Thus, one can observe that EI is much more robust to outliers, compared with ER. To support the above claim, we added the empirical results in Section C.5.1 of the Appendix.
>
> **(EI v.s. BE)** We conducted a numerical experiment for highlighting the benefit of EI over BE in terms of robustness to an imbalanced group negative rate. This is important since it is common to have an imbalanced group population in cases with fairness issues. As shown in Figure 7 in Section C.5.2 of the Appendix, if we increase the difference between group negative rates $|\mathbb{P}(\mathrm{y} = 0\mid \mathrm{z} = 0)-\mathbb{P}(\mathrm{y} = 0\mid \mathrm{z} = 1)|$ while keeping the same feature distributions, then the optimal BE classifier leads to completely different decision boundaries, while the decision boundary of EI classifier is almost unchanged. This phenomenon showcases the benefit of EI over BE.
>
> Second, achieving EI is guiding us to have better long-term fairness compared with ER and BE as empirically shown in Figure 4. Following the reviewer's comment, we added a mathematical analysis of why EI improves long-term fairness compared with ER and BE, in Section D of the Appendix in the revised manuscript.
>
> To be specific, we consider the dynamic scenario where each rejected sample (that is close to the decision boundary) makes effort to improve its feature, in order to get accepted in the next round. For EI/BE/ER/ERM classifiers, we analyzed how the feature distributions of different groups behave in the next round. We proved that using the EI classifier is beneficial to reduce the total-variation distance $d_{TV}$ between different groups, compared with classifiers constrained with other first-order fairness notions as well as the ERM solution, i.e.,
> $$
> d_{TV}(EI) < d_{TV}(BE),
> d_{TV}(EI) < d_{TV}(ER),
> d_{TV}(EI) < d_{TV}(ERM).
> $$
> We expect these results provide insight in how EI has benefits in equalizing the distributions of different groups.

---

> > ### Author Response · Authors · 2022-11-19
> > **Continued: Response to Reviewer Zw9H**
> >
> > > [R2-3] The connection between individual-level recourse fairness and EI is not mentioned.
> >
> >
> > We thank the reviewer for introducing the related work `Kügelgen et al. (2022)`. The individual-level recourse fairness notion suggested by `Kügelgen et al. (2022)` aims at finding a classifier $f$ satisfying
> >
> > $$
> > \min_{x': f(x') \geq 0.5} \mu_z(x', x) = \min_{x': f(x') \geq 0.5} \mu_{z'}(x', x), \quad \text{for all rejected individuals and } z, z' \in [Z],
> > $$
> > where $\mu_z(x', x)$ denotes the cost of improving feature from $x$ to $x'$ within a causal model when the individual has sensitive attribute $z \in [Z]$. This means that the minimum effort needed to improve the decision outcome is identical irrespective of the sensitive attribute, for all rejected samples. As pointed out by the reviewer, the individual-level recourse fairness summarized above is similar to EI (ours) defined as
> >
> >
> > $\mathbb{P}\left(\max\limits_{\mu(\Delta x) \leq \delta} f(x + \Delta x) \ge  0.5 \mid f(x)  <   0.5, z \right)  =  \mathbb{P}\left(\max\limits_{\mu(\Delta x) \leq \delta} f(x + \Delta x)  \ge  0.5 \mid f(x) < 0.5, z' \right),$
> >
> >
> > in the sense that both are taking care of equalizing the potential to improve the decision outcome for the rejected samples. It is also true that the fairness notion suggested by `Kügelgen et al. (2022)` is individual-level fairness, which has its own benefit compared with the group-level fairness notion considered in our work.
> >
> > However, at the same time, introducing individual-level fairness with respect to different groups inherently requires counterfactual fairness, which has its own limitation, as described by `Wu et al. (2019)`.
> >
> >
> > We added this comparison in the revised manuscript (see Appendix B.4).
> >
> > *References*:
> > * `Kügelgen et al. (2022)`: Von Kügelgen, Julius, Amir-Hossein Karimi, Umang Bhatt, Isabel Valera, Adrian Weller, and Bernhard Schölkopf. On the fairness of causal algorithmic recourse. In Proceedings of the AAAI Conference on Artificial Intelligence, vol. 36, no. 9, pp. 9584-9594. 2022.
> > * `Wu et al. (2019)`: Wu, Yongkai, Lu Zhang, and Xintao Wu. "Counterfactual fairness: Unidentification, bound and algorithm." Proceedings of the Twenty-Eighth International Joint Conference on Artificial Intelligence. 2019.
> >
> > > [R2-4] Why do not consider mutable features when defining the effort?
> >
> > Thanks for the good question. Following `Chen et al. (2020)`, we categorized the features into improvable features $x_{\text{I}}$, manipulable (mutable) features $x_{\text{M}}$, and immutable features ${x}_{\text{IM}}$ and only considered improving the improvable features. We can also think of your proposed way of defining effort, which is also reasonable and will not change the main point of our work.
> >
> > *References*:
> >
> > * `Chen et al. (2020)`: Chen, Yatong, Jialu Wang, and Yang Liu. "Linear Classifiers that Encourage Constructive Adaptation." arXiv preprint arXiv:2011.00355 (2020).

---

> > > ### Comment · Reviewer_Zw9H · 2022-12-06
> > > **Thank authors for responses**
> > >
> > > Thank authors for responses.
> > >
> > > In comparison between EI and Individual Recourse Fairness (Kügelgen et al.), thank authors for acknowledging that the fairness idea has been proposed previously in individual-level, and the current work is on group-level. If the goal is to analyze long-term fairness on the group level, previous works have provide some answer; if the goal is to give fine-grained improvability analysis for long-term fairness, I am not sure if the group-level EI is a better choice compared to more fine-grained notion with a very similar flavor (e.g., by Kügelgen et al.). I think the contribution of the paper can be strengthen if the benefit over their notion (e.g., a better rate of getting improved) can be illustrated via additional experiments (which may not be trivial, but would be very helpful).
> > >
> > > Regarding categorization of first-/zero- order fairness, I understand the fact that authors would like to emphasize the improvability considerations (i.e., "first-order fairness" in the paper) of EI. However, I am not convinced by the claim that zero-order fairness notions cannot tackle improvability. The dynamic fairness literature has demonstrated the downstream effect (one kind of improvability) of DP, EO, both are classified as zero-order fairness by the paper. The categorization should not only depend on what information is explicitly captured, but also how one interprets the audit result in the context. It might worth considering how to make sure the takeaway msg is informative and at the same time not (potentially) misleading.
> > >
> > > It would be great if those points can be addressed (when preparing the latest version, even if the revision cannot be uploaded later on). I have updated my evaluation accordingly. Thanks.

---

> > > > ### Author Response · Authors · 2022-12-08
> > > > **Thank you for your feedback**
> > > >
> > > > Dear Reviewer Zw9H,
> > > >
> > > > Thanks for reading our response carefully and for your valuable comments. We will take your suggestions for future modification.

---

> > > > ### Author Response · Authors · 2022-12-12
> > > > **Reply to Reviewer Zw9H: Part 1**
> > > >
> > > > > In comparison between EI and Individual Recourse Fairness (Kügelgen et al.), thank authors for acknowledging that the fairness idea has been proposed previously in individual-level, and the current work is on group-level. If the goal is to analyze long-term fairness on the group level, previous works have provide some answer; if the goal is to give fine-grained improvability analysis for long-term fairness, I am not sure if the group-level EI is a better choice compared to more fine-grained notion with a very similar flavor (e.g., by Kügelgen et al.). I think the contribution of the paper can be strengthen if the benefit over their notion (e.g., a better rate of getting improved) can be illustrated via additional experiments (which may not be trivial, but would be very helpful).
> > > >
> > > > Thanks for the great suggestion. Our experimental results below show that EI indeed outperforms Individual-Level Fair Causal Recourse (ILFCR) proposed by `Kügelgen et al. (2022)` in terms of improving long-term fairness.
> > > >
> > > > We first describe the setting considered in `Kügelgen et al. (2022)`. This paper assumes that each observed variable $x_i$ is determined by (i) its direct causes (causal parents) which include the sensitive attribute $z$ and other observed variables $x_j$, and (ii) an unobserved variable $u_i$. Since our dynamics experiment (results shown in Fig.4) considers only one-dimensional variable $x$, it is determined by the sensitive attribute $z$ and latent variable $u$. Therefore, we write $x$ as a function of $z$ and $u$, i.e., $x(z,u).$
> > > >
> > > > Before we describe how we compute ILFCR in our experiments, we introduce the definition of causal recourse and ILFCR. Given a model and a sample with feature $x$, the causal recourse $C(x)$ of the sample is the minimum cost of changing the features, in order to alter the decision of the model.
> > > >
> > > > Let $\tau^{(z)} \in \mathbb{R}$ be the acceptance threshold of group $z\in\{0,1\}$ and $\mu(\Delta x) = |\Delta x|$ be the cost of improving the feature by $\Delta x$. Let $x'$ denote the improved feature. In our one-dimensional dynamic setting, the causal recourse is defined as
> > > >
> > > > $$
> > > > C(x(z,u)) = \min_{x' \geq \tau^{(z)}} \mu(x'-x(z,u)) = \max(\tau^{(z)} - x(z, u), 0),$$
> > > > for $z=0,1$. ILFCR requires different groups have the same causal recourse for all realizations of the latent variable $u$:
> > > > $$
> > > > \max_{u} |C(x(0,u))  - C(x(1,u)) | = 0.
> > > > $$
> > > >
> > > > Now we describe how we compute ILFCR disparity. Recall that for dynamic experiments, we assumed the feature $x$ follows Gaussian distribution for each group $z \in \{0,1\}$, i.e., $x \mid z \sim \mathcal{N}(\mu_z, \sigma_z^2)$. This can be represented as a notation used in `Kügelgen et al. (2022)`: the latent variable is $u \sim \mathcal{N}(0,1)$ and the feature is represented as $x(z,u) = \mu_z + \sigma_z u$.
> > > > We measure the ILFCR disparity of a decision boundary pair ($\tau_0, \tau_1$)  by
> > > >
> > > > $$
> > > > \text{ILFCR Disparity}(\tau^{(0)}, \tau^{(1)})
> > > > $$
> > > > $$
> > > > = \max_{u}|C (x(0,u)) - C (x(1,u))|
> > > > $$
> > > > $$
> > > > = \max_{u}\left|\max\left(\tau^{(0)} - x(0,u), 0\right) - \max\left(\tau^{(1)} - x(1,u), 0\right)\right|
> > > > $$
> > > > $$
> > > > = \max_{u}\underbrace{\left|\max\left(\tau^{(0)} - \mu_0-\sigma_0 u, 0\right) - \max\left(\tau^{(1)} - \mu_1-\sigma_1 u, 0\right)\right|}_{ = (\tau^{(0)}-\tau^{(1)} -\mu_0 +\mu_1-(\sigma_0-\sigma_1)u) \to \infty \text{ when }u \to -\infty}.
> > > > $$
> > > >
> > > > Note that the ILFCR disparity is infinity if we take the maximum over all $u$. Therefore, we instead ignore the tail distribution and focus on the samples with $x \in [\mu_z - 3\sigma_z, \mu_z + 3\sigma_z]$ for group $z = 0,1.$ In order to find a classifier with a small ILFCR disparity, we numerically solve the following constrained optimization problem
> > > > $$
> > > > \min\limits_{\tau^{(0)}, \tau^{(1)}} \quad  \text{ILFCR Disparity}(\tau^{(0)}, \tau^{(1)})\quad \text{s.t.}\quad \text{error rate} \leq \alpha/2,
> > > > $$
> > > > using `scipy.optimize`, where $\alpha$ is the acceptance ratio.
> > > >
> > > > In Table R1 and R2, we provide experimental results comparing the long-term unfairness of ILFCR and EI classifiers under the dynamic scenarios. The $\mu_z, \sigma_z$ values are set to $\mu_0=0, \sigma_0=0.5, \mu_1 = 1,\sigma_1=1$ (same as the parameter used for Figure 4(ii)) for Table R1, and $\mu_0=0, \sigma_0=0.5, \mu_1 = 1,\sigma_1=0.5$ (same as the parameter used for Figure 4(iv)) for Table R2. From the table below, one can confirm that EI has better performance than ILFCR. We will include this result in the revised manuscript, by updating the plots of Figure 4.
> > > >
> > > > Table R1: Long-term unfairness at various iterations. Initial distribution is set to $x\mid z=0\sim\mathcal{N}(0,0.5^2)$ and $x\mid z=1\sim \mathcal{N}(1,1)$.
> > > > | | Long-term unfairness ($\downarrow$) | Long-term unfairness ($\downarrow$) |
> > > > |:--:|:--:|:--:|
> > > > |  | ILFCR `(Kügelgen et al., 2022)` | EI (Ours) |
> > > > |Initial Distribution| .547| .547|
> > > > |Iteration 2| .526| .119|
> > > > |Iteration 4| .534| .088|
> > > > |Iteration 6| .542| .073|
> > > > |Iteration 8| .549| .065|
> > > > |Iteration 10| .556| .059|

---

> > > > > ### Author Response · Authors · 2022-12-12
> > > > > **Continued: Reply to Reviewer Zw9H: Part 2**
> > > > >
> > > > > Table R2: Long-term unfairness at various iterations. Initial distribution is set to $x\mid z=0\sim\mathcal{N}(0,0.5)$ and $x\mid z=1\sim \mathcal{N}(1,0.5)$.
> > > > > | | Long-term unfairness ($\downarrow$) | Long-term unfairness ($\downarrow$) |
> > > > > |:--:|:--:|:--:|
> > > > > | | ILFCR `(Kügelgen et al., 2022)` | EI (Ours) |
> > > > > |Initial Distribution| .683| .683|
> > > > > |Iteration 2| .280| .115|
> > > > > |Iteration 4| .320| .094|
> > > > > |Iteration 6| .358| .083|
> > > > > |Iteration 8| .392| .075|
> > > > > |Iteration 10| .418| .069|
> > > > >
> > > > > We now explain why EI outperforms ILFCR in terms of improving long-term fairness, as in Table R1. Recall that the initial feature distributions are set to $x \mid z = 0 \sim \mathcal{N}(0,0.5^2)$ and $x\mid z = 1 \sim \mathcal{N}(1,1)$, and the ILFCR classifier aims to minimize $|C(x(0,u)) - C(x(1,u))|$ for any $u$.
> > > > >
> > > > > We visualize the ILFCR classifier as below. Note that the ILFCR classifier accepts more individuals from group 1 (the advantaged group), compared to the ground-truth classifier. In contrast, EI classifier has a lower acceptance threshold for the disadvantaged group, which encourages the rejected individuals from the disadvantaged group to make more effort, thereby outperforming ILFCR in terms of improving long-term fairness.
> > > > >
> > > > > ```
> > > > > PDF of Group 0 (disadvantaged group) x|z=0 ~ N(0,0.5^2)
> > > > >
> > > > >                  True acceptance threshold & ILFCR
> > > > >                               |
> > > > >                         EI |  |
> > > > >                   *   *    |  |
> > > > >                            |  |
> > > > >                  *     *   |  |
> > > > >                 *       *  |  |
> > > > >                            |  |
> > > > >                *         * |  |
> > > > >                            |  |
> > > > >               *           *|  |
> > > > >                            |  |
> > > > >              *             *  |
> > > > >                            |  |
> > > > >             *              |* |
> > > > >            *               | *|
> > > > >           *                |  *
> > > > >        ***                    |***
> > > > > ***(*)*                       |    **************************
> > > > >  mu_0-3*sigma_0               |
> > > > >     |---> Causal Recourse <---|
> > > > >
> > > > >
> > > > > PDF of Group 1 (advantaged group) x|z ~ N(1,1)
> > > > >
> > > > >                   True acceptance threshold
> > > > >                               |
> > > > >                       ILFCR |*|*| EI
> > > > >                         *** | | | ***
> > > > >                       **    | | |    **
> > > > >                     **      | | |      **
> > > > >                   **        | | |        **
> > > > >                ***          | | |          ***
> > > > >             ***             | | |             ***
> > > > >        *****                | | |                *****
> > > > > (*)****                     | | |                     ******
> > > > > mu_1-3*sigma_1              |
> > > > >  |---> Causal Recourse <----|
> > > > >
> > > > > ```
> > > > > Note: the figure above visualizes the causal recourse of the ILFCR classifier when $u=-3$.
> > > > >
> > > > > > Regarding categorization of first-/zero- order fairness, I understand the fact that authors would like to emphasize the improvability considerations (i.e., "first-order fairness" in the paper) of EI. However, I am not convinced by the claim that zero-order fairness notions cannot tackle improvability. The dynamic fairness literature has demonstrated the downstream effect (one kind of improvability) of DP, EO, both are classified as zero-order fairness by the paper. The categorization should not only depend on what information is explicitly captured, but also how one interprets the audit result in the context. It might worth considering how to make sure the takeaway msg is informative and at the same time not (potentially) misleading.
> > > > >
> > > > > We appreciate the sharp comments. We totally agree that claiming the "zero-order fairness notion cannot tackle improvability" is incorrect. In the revised manuscript, we will remove misleading sentences and clarify that the categorization is only based on "whether the definition has the effort term", and it does not mean that the zero-order notion cannot be used for dynamic scenarios.

---

### Official Review · Reviewer_dcYV · 2022-10-25

**Confidence:** 3
**Correctness:** 4
**Technical Novelty And Significance:** 2
**Empirical Novelty And Significance:** 2
**Recommendation:** 6

**Clarity, Quality, Novelty And Reproducibility:**

Clarity: Overall the paper is straightforward to read. The proofs I skimmed (app A) seem clear.
- Page 6 in experimental setting: Where are the "statistics for five trials"? Are the errors standard error or standard deviation?

Reproducibility: Information needed for reproducibility is located in the appendix. I did not look at the code but the authors also assert that hyperparameters used in the experiments are located there as well. I feel confident that I could re-construct the experiments from the paper.

Quality/Novelty: The authors show that the notion of fairness considered is similar to current long-term metrics but distinct, and provide three algorithms, two of which are small changes from previous works that use soft constraints, i.e., create a fairness-based regularization penalty. These methods do not come with guarantees on the solution found by the optimization process, but can be considered stepping stones for future work.

Small typos / issues and suggested improvements:
- Appendix B first sentence: "we explain what each term means in EI definition means" -> "we explain what each term in the EI definition means"
- I did not realize until the end of the paper that there was a second related work section---I suggest mentioning this in the first related work section!
- "First-order" is never truly defined, and the reader is left to infer from the text. Does first-order imply that these metrics "take potential follow-up inequity risk" one "time-step" into the future, i.e., one decision into the future? The experimental section shows a one-step scenario (the first set of experiments), and a multi-step scenario (three rounds). Are the long-term metrics mentioned actually n-step fairness notions?
- EI disparity can be inferred from the main text but I suggest that to reduce confusion it is defined in the main work. ATM the authors refer to appendix B.3 and then use this term later numerous times in the main body.

**Strength And Weaknesses:**

Strengths / comments:
- I enjoyed reading the introduction and the (first) related work section, where the authors pinpoint the importance of their metric and show how it is distinct from current known metrics in the field.
- Concepts are elaborated on in the appendix or figures are provided to help readers better understand material. The example of $\mu(\Delta \mathbf{x_1})$ in the para underneath Defn 2.1 was very helpful.
- The authors did not over-state the results of their fairness-based regularization algorithms, e.g., acknowledged that these methods can mitigate EI unfairness and do not provide further guarantees on solutions found.
- The organization of section 3 was straightforward and well-written.

Weaknesses / comments:

- Why don't the authors compare the other long-term fairness metrics to the long-term fairness-unaware metrics in the first set of experiments? ERM is a very simple baseline, and arguably the three datasets provided are three more datasets over which other long-term fairness baselines can be used as a comparison. We could consider this a single-step episode, which is a subset of an n-step episode. The paper that presents ER also presents a fairness-based regularization method to find solutions that mitigate ER.

- How easy is it to find policies that mitigate unfairness when there are multiple groups, as is the case in many real-world applications? (For example, in applications where one might consider race, gender, or relevant proxies.)  One of the issues with just adding a (fairness-based) penalty to the objective is that the alg is only given incentive to satisfy fairness, but has the freedom to violate fairness if that results in a more improvement to the original objective. Qualitatively, it seems that scaling the importance of each of the additional pair-wise penalties wrt the primary objective would not be simple or insightful for users.

- Adding on to the previous point, the Zafar paper that inspired the covariance-based EI penalty creates proxies to the original fairness constraint, but does not show (theoretically) that solutions found by the new proxy-objective solve the original objective. The covariance-based EI method also has no guarantees on the solution found by the algorithms (perhaps the KDE-based penalty as well though I am unfamiliar with the work which inspired it, and did not take the time to read through the work). Ultimately this is a weakness of using fairness-based regularization to mitigate unfairness. I suggest that the users address this as a weakness in the main body.


**Summary Of The Paper:**

This work proposes a notion of long-term impact called equal improvability (EI) that equalizes the "effort required to improve" of rejected individuals belonging to different sensitive groups. This metric is theoretically compared to three other long-term metrics: Bounded Effort and Equal Recourse---conditions for meeting each of these notions of fairness (in relation to each other) are derived. Three fairness-based regularization methods are proposed to find policies that mitigate unfairness wrt EI, each with a unique differentiable penalty term. These methods are empirically tested against synthetic and real-world data. In the first set of experiments, the three mitigation strategies are tested against a fairness-unaware algorithm over two real-world datasets (non-dynamic). In the second set of experiments, results for a simple dynamical setting are shown.

**Summary Of The Review:**

I make a recommendation to accept.

- The approach seems well-motivated, and the authors take care to show that their metric is distinct from other current methods in the longterm fairness space.
- The paper also does not over-state its claims, e.g., the three optimization algorithms introduced offer no guarantees on the solutions found will actually mitigate EI unfairness. This limitation is well-known in regards to soft constraints but is still important to address in the main body (see weaknesses for an elaboration).
- There are also a few clarifying points from the experiments I would like addressed, such as the choice of baselines for the non-dynamic experiments, as well as the statistics being presented in the experiments.

--- post rebuttal ---
I thank the authors for the detailed response! The authors agreed to address many of my concerns in their next iteration, which I think will make the work more straightforward. After discussion with other reviewers, I still lean towards acceptance.

---

> ### Author Response · Authors · 2022-11-19
> **Response to Reviewer dcYV**
>
> > [R1-1] Compare EI with other long-term fairness baselines in the first set of experiments.
>
> As per the reviewer's comment, we added ER (`Gupta et al., 2019`) and BE (`Heidari et al., 2019`) as new baselines in our experiment in Section 4.1. We leverage the algorithm suggested by `Gupta et al., 2019` for mitigating ER disparity. We extended our loss-based approach (designed for reducing EI disparity) to reduce BE disparity, by redefining the BE loss of group $z$ as
> $$
> L_z^{\text{BE}} \triangleq \frac{1}{\text{number of samples in group }z} \sum_{i \in I_{-,z}} \ell (1, \max_{\lVert \Delta x_{\text{I} i} \rVert \leq \delta} f(x_i + \Delta x_i)),
> $$
> where
> $$
> L_z^{\text{EI}} \triangleq \frac{1}{\text{number of rejected samples in group }z} \sum_{i \in I_{-,z}} \ell (1, \max_{\lVert \Delta x_{\text{I} i} \rVert \leq \delta} f(x_i + \Delta x_i)),
> $$
> and $I_{-,z}$ is the set of rejected samples in group $z$ for $z\in [Z]$. The table presents our experiment results on synthetic dataset. We observe that our proposed approach designed for reducing EI indeed achieves the lowest EI disparity.
> | Metric | ERM | ER (`Gupta et al., 2019`) | BE (Loss-based Approach) |  EI (Loss-based Approach) |
> |:---:|:---:|:---:|:---:|:---:|
> |Error Rate $(\downarrow)$ | $.221 \pm .001$ | $.235 \pm .009$ | $.252 \pm .006$ | $.246 \pm .001$ |
> | EI Disp $(\downarrow)$| $.117 \pm .007$ | $.036 \pm .018$ | $.006 \pm .004$ | $.002 \pm .001$ |
>
> *References*:
> * Vivek Gupta, Pegah Nokhiz, Chitradeep Dutta Roy, and Suresh Venkatasubramanian. Equalizing recourse across groups. arXiv preprint arXiv:1909.03166, 2019.
> * Hoda Heidari, Vedant Nanda, and Krishna P. Gummadi. On the long-term impact of algorithmic decision policies: Effort unfairness and feature segregation through social learning, 2019.
>
> > [R1-2]  Can we find policies that mitigate unfairness on multiple sensitive attributes (race and gender), as is the case in many real-world applications? It seems like scaling the importance of each of the additional pair-wise penalties with respect to the primary objective would not be simple or insightful for users.
>
> Thanks for the sharp question. It is true that handling multiple groups at the same time is not trivial for our method, and it is an interesting future research direction. We stated this point in the conclusion section of the revised manuscript.
>
> > [R1-3] Zafar's work creates the covariance-based penalty as a proxy to the original fairness constraint but does not theoretically show that this solves the original objective. The proposed schemes also suffer from such limitations.
>
> Yes, the reviewer is true that similar to Zafar's work, the proposed proxy-objective-based regularization methods have no unfairness guarantees on the solution. We will add this as a weakness in the camera-ready version.
>
> > [R1-4] Where are the "statistics for five trials"? Are the errors standard error or standard deviation?
>
> In Tables 2 and 3, we reported the mean and standard deviation for 5 trials. The standard deviation is given after the $\pm$ sign.
>
> > [R1-5] Appendix B first sentence: "we explain what each term means in EI definition means" -> "we explain what each term in the EI definition means"
>
> Thanks for thoroughly reading our manuscript including the appendix! We have made the corresponding fix in the updated version, highlighted in blue.
>
> > [R1-6] I did not realize until the end of the paper that there was a second related work section---I suggest mentioning this in the first related work section!
>
> Thanks for your suggestion. We added a pointer to the related work section on page 4 of the revised manuscript.

---

> > ### Author Response · Authors · 2022-11-19
> > **Continued: Response to Reviewer dcYV**
> >
> > > [R1-7] "First-order" is never truly defined, and the reader is left to infer from the text. It is better to clarify "first-order" and "long-term fairness" metrics.
> >
> > It is our fault that we did not explicitly explain what first-order means. The below table shows the distinction between zero-order, first-order and long-term fairness notions. We will add this to the revised manuscript.
> >
> > | | Zero-order Fairness | First-order Fairness | Long-term Fairness |
> > |:--:|:--:|:--:|:--:|
> > | Definition | Fairness notions that do NOT consider the improvability of individuals | Fairness notions that consider the improvability of individuals | Fairness notions that equalize the feature distributions of different groups after multiple steps of improvement of individuals
> > | Examples | Demographic Parity (DP), Equal Opportunity (EO), Equalized Odds (EOD) | Equal Improvability (EI), Bounded Effort (BE), Equal Recourse (ER)| -
> > | Long-term Motivations | - | Consider the effort needed for unqualified samples (in different groups) to improve their decision outcome | Equalizes the quality of different groups
> >
> > > [R1-8] The definition of EI disparity is in Appendix B.3, but I suggest defining it in the main manuscript to reduce confusion.
> >
> >
> > Thanks for your suggestion. We have added the definition of EI disparity to the beginning of the experiment section as per the reviewer's comment.

---

### Author Response · Authors · 2022-11-19
**General Comments to AC and All Reviewers**

We thank the Reviewers for their insightful feedback and constructive comments and for providing suggestions that would improve our paper.

First of all, we are encouraged that the reviewers found that: (i) the topic is well-motivated (R-dcYV), timely and interesting (R-q7oe, R-Eon1), and a nice addition to the growing literature on the delayed impact of fairness (R-q7oe); (ii) the proposed EI notion combines advantages of previous notions and avoids problems of existing notions (R-Zw9H) and is theoretically analyzed (R-dcYV); (iii) the proposed methods for achieving EI showed their benefits in experimental results (R-Zw9H, R-Eon1); (iv) our paper is well-written and clear (R-dcYV, R-Zw9H, R-97oe, R-Eon1) and enjoyable to read R-dcYV, R-q7oe, R-Eon1), figures help better understand the material (R-dcYV), and the benefits of the proposed method are not over-claimed (R-dcYV).


As for the concerns/questions raised, we believe that we successfully addressed every single one, as replied to each reviewer. We integrated most of the answers and new results in the newly updated version (attached).

In particular, we found that there are three major questions raised by the reviewers. To answer them, we provide additional mathematical analysis and additional experimental results, and summarized below are the three most important questions and our responses to them.


> Q1. Why EI improves long-term fairness?

In Section D of the Appendix, we added theoretical results to demonstrate the benefit of EI in improving long-term fairness, compared with other fairness notions (BE and ER). To be specific, we consider the dynamic scenario where each rejected sample (that are close to the decision boundary) makes effort to improve its feature, in order to get accepted in the next round. For EI/BE/ER/ERM classifiers, we analyzed how the feature distributions of different groups behave in the next round. We proved that using the EI classifier is beneficial to reduce the total-variation distance $d_{TV}$ between different groups, compared with classifiers constrained with other first-order fairness notions as well as the ERM solution, i.e.,
$$
d_{TV}(EI) < d_{TV}(BE),
d_{TV}(EI) < d_{TV}(ER),
d_{TV}(EI) < d_{TV}(ERM).
$$
We expect these results provide insight in how EI has benefits in equalizing the distributions of different groups.

> Q2. Comparison with related works?

As per the reviewers' questions, we have systematically investigated the specific advantages of EI over BE (`Heidari et al. 2019`) and ER (`Gupta et al., 2019`) via numerical experiments and mathematical analysis. The results show that the main unique benefit of EI compared with ER and BE is that EI is robust in general data sets, while ER and BE are vulnerable to cases of having outliers or imbalanced group negative rates.

**(EI v.s. ER)** We start by sharing a simple analysis. Suppose an outlier, having feature-attribute pair $(x,z)$, is added to the dataset with $n$ samples. Let the outlier is misqualified $f(x) < 0.5$, and requires effort $\mu(\Delta x) = M$ to achieve $f(x+\Delta x) \ge 0.5$.

In such case, the EI disparity increases at most $\frac{1}{n}$ since EI measures the *portion* of samples with improved outcomes after making efforts. On the other hand, the ER disparity increases by $\frac{M}{n}$, since ER measures the minimum required efforts *averaged* over all samples. Note that a single outlier with a large $M$ can significantly increase the ER disparity, which does not happen for EI. Thus, one can observe that EI is much more robust to outliers, compared with ER. To support the above claim, we added the empirical results in Section C.5.1 of the Appendix.

**(EI v.s. BE)** We conducted a numerical experiment for highlighting the benefit of EI over BE in terms of robustness to imbalanced group negative rate. This is important since it is common to have an imbalanced group population in cases with fairness issues. As shown in Figure 7 in Section C.5.2 of the Appendix, if we increase the difference between group negative rates $|\mathbb{P}(\mathrm{y} = 0\mid \mathrm{z} = 0)-\mathbb{P}(\mathrm{y} = 0\mid \mathrm{z} = 1)|$ while keeping the same feature distributions, then the optimal BE classifier leads to completely different decision boundaries, while the decision boundary of EI classifier is almost unchanged. This phenomenon showcases the benefit of EI over BE.

**(EI v.s. Individual-level Equal Recourse)** We perform a comparison between EI and Individual-level Equal Recourse(`Kügelgen et al. (2022)`) here. The individual-level recourse fairness notion suggested by `Kügelgen et al. (2022)` aims at finding a classifier $f$ satisfying
$$
\min_{x': f(x') \geq 0.5} \mu_z(x', x) = \min_{x': f(x') \geq 0.5} \mu_{z'}(x', x), \quad \text{for all rejected individuals and } z, z' \in [Z],
$$

---

> ### Author Response · Authors · 2022-11-19
> **Continued: [Q2] Comparison with related works and [Q3] Comprehensive Experiments**
>
> where $\mu_z(x', x)$ denotes the cost of improving feature from $x$ to $x'$ within a causal model when the individual has sensitive attribute $z \in [Z]$. This means that the minimum effort needed to improve the decision outcome is identical irrespective of the sensitive attribute, for all rejected samples. The individual-level recourse fairness summarized above is similar to EI (ours) defined as
>
> $\mathbb{P}\left(\max\limits_{\mu(\Delta x) \leq \delta} f(x + \Delta x) \ge  0.5 \mid f(x)  <   0.5, z \right)  =  \mathbb{P}\left(\max\limits_{\mu(\Delta x) \leq \delta} f(x + \Delta x)  \ge  0.5 \mid f(x) < 0.5, z' \right),$
>
> in the sense that both are taking care of equalizing the potential to improve the decision outcome for the rejected samples. It is also true that the fairness notion suggested by `Kügelgen et al. (2022)` is individual-level fairness, which has its own benefit compared with the group-level fairness notion considered in our work.
>
> However, at the same time, introducing individual-level fairness with respect to different groups inherently requires counterfactual fairness, which has its own limitation, as described by `Wu et al. (2019)`.
>
> > Q3. Comprehensive Experiments
>
> As per the reviewer's comment, we added ER (`Gupta et al., 2019`) and BE (`Heidari et al., 2019`) as new baselines in our experiment in Section 4.1. We leverage the algorithm suggested by `Gupta et al., 2019` for mitigating ER disparity. We extended our loss-based approach (designed for reducing EI disparity) to reduce BE disparity, by redefining the BE loss of group $z$ as
> $$
> L_z^{\text{BE}} \triangleq \frac{1}{\text{number of samples in group }z} \sum_{i \in I_{-,z}} \ell (1, \max_{\lVert \Delta x_{\text{I} i} \rVert \leq \delta} f(x_i + \Delta x_i)),
> $$
> where
> $$
> L_z^{\text{EI}} \triangleq \frac{1}{\text{number of rejected samples in group }z} \sum_{i \in I_{-,z}} \ell (1, \max_{\lVert \Delta x_{\text{I} i} \rVert \leq \delta} f(x_i + \Delta x_i)),
> $$
> and $I_{-,z}$ is the set of rejected samples in group $z$ for $z\in [Z]$. The table presents our experiment results on synthetic dataset. We observe that our proposed approach designed for reducing EI indeed achieves the lowest EI disparity.
>
> | Metric | ERM | ER (`Gupta et al., 2019`) | BE (Loss-based Approach) |  EI (Loss-based Approach) |
> |:---:|:---:|:---:|:---:|:---:|
> |Error Rate $(\downarrow)$ | $.221 \pm .001$ | $.235 \pm .009$ | $.252 \pm .006$ | $.246 \pm .001$ |
> | EI Disp $(\downarrow)$| $.117 \pm .007$ | $.036 \pm .018$ | $.006 \pm .004$ | $.002 \pm .001$ |
>
> *References*:
> * Vivek Gupta, Pegah Nokhiz, Chitradeep Dutta Roy, and Suresh Venkatasubramanian. Equalizing recourse across groups. arXiv preprint arXiv:1909.03166, 2019.
> * Hoda Heidari, Vedant Nanda, and Krishna P. Gummadi. On the long-term impact of algorithmic decision policies: Effort unfairness and feature segregation through social learning, 2019.
> * Von Kügelgen, Julius, Amir-Hossein Karimi, Umang Bhatt, Isabel Valera, Adrian Weller, and Bernhard Schölkopf. On the fairness of causal algorithmic recourse. In Proceedings of the AAAI Conference on Artificial Intelligence, vol. 36, no. 9, pp. 9584-9594. 2022.

---

### Comment · Area_Chair_nx3o · 2022-12-09
**Summary of Reviewer Virtual Meeting**

Most of the reviewers participated in a virtual call to discuss the paper. Overall, the reviewers agreed that the paper tackles an important problem. While most of their initially raised questions have been addressed by the edits to the paper or author response, there are a few remaining concerns:

1) lack of comparison in the *main text* of the current paper to "On the Fairness of Causal Algorithmic Recourse" (Kügelgen et al., AAAI). I see that the authors added a discussion to Appendix B.4, but the reviewers believe that such a discussion belongs in the main text. If no such comparison exists, the paper may be misrepresenting the novelty of their contributions to the literature
2) hard categorization of notions of fairness to zero vs first shot. From the reviewers point of view, such a categorization is somewhat misrepresentative, and may confuse the literature, or be incorrectly copied to work building on this one. Even so-called zero-shot fairness notions could be applied again as the world changes, and so they are not restricted to being zero-shot, even if historically they were used in a static way. Therefore, the reviewers believe that the language around these previously introduced notions of fairness should be toned down not to cause confusion.
3) lack of discussion of limitations: the presented analysis is a one-step analysis of long term fairness.

Please also see individual responses from each of the reviewers, and suggested experiments that could further improve the work.

While the authors cannot upload new versions of the paper, it would be great to hear the exact edits they would propose to address the remaining concerns.

---

> ### Author Response · Authors · 2022-12-12
> **Reply to AC & All Reviewers**
>
> > Overall, the reviewers agreed that the paper tackles an important problem. While most of their initially raised questions have been addressed by the edits to the paper or author's response, there are a few remaining concerns. ... While the authors cannot upload new versions of the paper, it would be great to hear the exact edits they would propose to address the remaining concerns.
>
> We thank the area chair and reviewers for reviewing our updated manuscript as well as the author's response. Below we provide the exact edit plan to handle the remaining concerns and we also provide the revised abstract in the response to reviewer Eon1.
>
>
> > 1. lack of comparison in the main text of the current paper to "On the Fairness of Causal Algorithmic Recourse" (Kügelgen et al., AAAI). I see that the authors added a discussion to Appendix B.4, but the reviewers believe that such a discussion belongs in the main text. If no such comparison exists, the paper may be misrepresenting the novelty of their contributions to the literature
>
> We agree that the comparison with the AAAI paper (`Kügelgen et al., 2022`) should be in the main manuscript. In the revised manuscript, we will move the comparison (currently in Sec.B.4) to Sec.2 of the main manuscript, specifically in the paragraph `Comparison of EI with other fairness notions`. To match the page limit of the main manuscript, we will revise figures and sentences in a more compact format.
>
>
>
> > 2. hard categorization of notions of fairness to zero vs first shot. From the reviewers' point of view, such a categorization is somewhat misrepresentative and may confuse the literature, or be incorrectly copied to work building on this one. Even so-called zero-shot fairness notions could be applied again as the world changes, and so they are not restricted to being zero-shot, even if historically they were used in a static way. Therefore, the reviewers believe that the language around these previously introduced notions of fairness should be toned down not to cause confusion.
>
>
>
> We agree that the terminology of `zeroth-order` and `first-order` fairness is misrepresentative and makes confusion since DP, and EO fairness notions can be also applied when each sample updates its features. In the revised manuscript, we will remove the terminology (zero/one-shot or zeroth/first-order). Instead, when introducing BE (`Heidari et al. 2019`), ER (`Gupta et al., 2019`), individual-level ER (`Kügelgen et al. (2022)`) and EI (ours), we will just explain that these notions consider fairness after each individual makes effort to improve its features. This revision will be made throughout the paper.
>
>
>
> > 3. lack of discussion of limitations: the presented analysis is a one-step analysis of long-term fairness.
>
> In Sec.4.2 of the revised manuscript, we will give pointers for the one-step analysis of long-term fairness (in Appendix D), and clearly mention that this analysis has a limitation of only handling the one-step feature update setting, not the multi-step setting.
>
> > Please also see individual responses from each of the reviewers and suggested experiments that could further improve the work.
>
>
> We ran additional experiments suggested by reviewers and added the results in the response. The summary of the three main experiments is as follows:
>
> * We added examples when some of the existing fairness notions (EO, BE, and ER) harm the long-term fairness, while EI improves. See our response to Reviewer q7oe.
> * We showed that EI outperforms individual-level fair causal recourse (ILFCR) in terms of improving long-term fairness. See our response to Reviewer Zw9H.
> * We provided numerical results showing that EI is more robust to outliers than ER. See our response to Reviewer Eon1.

---

### Decision · Program_Chairs · 2023-01-20

**Decision:**

Accept: poster

**Justification For Why Not Higher Score:**

While the paper studies an important problem and proposes a new notion of fairness, the existence of Kügelgen et al. paper diminishes the novelty and contributions to the literature. In particular, the current submission can be seen as a group version of the fairness notions proposed in Kügelgen et al. Further, according to the reviewers, the proposed solutions to optimizing for EI are also no new and closely match those existing in previous work.

**Justification For Why Not Lower Score:**

After posting the meeting summary on openreview, the authors responded to each of the points proposing concrete edits to the paper. They also added additional experiments, which should strengthen the paper further. Overall, the paper is studying an interesting and important problem.

**Metareview: Summary, Strengths And Weaknesses:**

The paper proposed a new notion of fairness at a group level that captures a dynamic system. In particular, the motivating goal is to model a world where each group can spend "effort" to change their features to maximize for some desired outcome. Based on this new notion of fairness, which the authors term "equal improvability" (EI), several optimization objectives are proposed. In their empirical work, the authors show how enforcing EI can be advantageous for long term fairness. The initial empirical results were somewhat lacking in the dynamic setting and high dimensional setting. The other key weaknesses identified by the reviewers are summarized  in the AC-reviewer meeting notes below.

**Note From Pc:**

if the above contains the word "oral" or "spotlight" please see: "oral" presentation means -> notable-top-5% and "spotlight" means -> notable-top-25%. As stated in our emails, we are disassociating presentation type from AC recommendations

**Summary Of Ac-Reviewer Meeting:**

3 out of 4 reviewers participated in a virtual call to discuss the paper (reviewer dcYV was missing but provided a justification for not being able to participate). Overall, the reviewers agreed that the paper tackles an important problem. While most of their initially raised questions have been addressed by the edits to the paper or author response, there were a few remaining concerns:

1) lack of comparison in the *main text* of the current paper to "On the Fairness of Causal Algorithmic Recourse" (Kügelgen et al., AAAI). I see that the authors added a discussion to Appendix B.4, but the reviewers believe that such a discussion belongs in the main text. If no such comparison exists, the paper may be misrepresenting the novelty of their contributions to the literature
2) hard categorization of notions of fairness to zero vs first shot. From the reviewers point of view, such a categorization is somewhat misrepresentative, and may confuse the literature, or be incorrectly copied to work building on this one. Even so-called zero-shot fairness notions could be applied again as the world changes, and so they are not restricted to being zero-shot, even if historically they were used in a static way. Therefore, the reviewers believe that the language around these previously introduced notions of fairness should be toned down not to cause confusion.
3) lack of discussion of limitations: the presented analysis is a one-step analysis of long term fairness.
4) additional experiments could further improve the work (see individual responses from each of the reviewers).